# Oscillators are all you need: Irregular Time Series Modelling via Damped Harmonic Oscillators with Closed-form Solutions

## Abstract

Transformers excel at time series modelling through attention mechanisms that capture long-term temporal patterns. However, they assume uniform time intervals and therefore struggle with irregular time series. Neural Ordinary Differential Equations (NODEs) effectively handle irregular time series by modelling hidden states as continuously evolving trajectories. ContiFormers (Chen et al., 2023) combine NODEs with Transformers, but inherit the computational bottleneck of the former by using heavy numerical solvers. This bottleneck can be removed by using a closed-form solution for the given dynamical system - but this is known to be intractable in general! We obviate this by replacing NODEs with a novel linear damped harmonic oscillator analogy - which has a known closed-form solution. We model keys and values as damped, driven oscillators and expand the query in a sinusoidal basis up to a suitable number of modes. This analogy naturally captures the query-key coupling that is fundamental to any transformer architecture by modelling attention as a resonance phenomenon. Our closed-form solution eliminates the computational overhead of numerical ODE solvers while preserving expressivity. We prove that this oscillator-based parameterisation maintains the universal approximation property of continuous-time attention; specifically, any discrete attention matrix realisable by ContiFormer's continuous keys can be approximated arbitrarily well by our fixed oscillator modes. Our approach delivers both theoretical guarantees and scalability, achieving state-of-the-art performance on irregular time series benchmarks while being orders of magnitude faster.

## 1 Introduction

Transformers are widely used for modelling time series data (Zeng et al., 2022). However, they assume uniform sampling (Zeng et al., 2022), whereas many real world datasets, such as finance, astronomy, healthcare, and magnetic navigation, are often based on irregular time series (Rubanova et al., 2019). This data exhibits continuous behaviour with intricate relationships across continuously evolving observations (Lipton et al., 2016). Dividing the timeline into intervals of equal size can hamper the continuity of data. Neural Ordinary Differential Equations (NODEs) (Chen et al., 2019) address irregular time series by abandoning the fixed-layer stack and instead letting a neural network dictate how the hidden state moves through time. This keeps the representation on the exact observation times and preserves the natural topology of the input space (Dupont et al., 2019). The bottleneck of using NODEs is the high computational cost due to the use of numerical solvers (Oh et al., 2025). While there have been closed-form solutions for continuous RNNs (Hasani et al., 2022) that have addressed computational bottlenecks in continuous-time RNNs, these approaches still fall short of the efficiency that attention mechanisms provide for capturing both long-range dependencies (Niu et al., 2024).

This challenge of finding a closed-form solution for the ContiFormer motivated us to explore neural networks through the lens of physical systems (Hopfield, 1982), where efficient solutions can often emerge from exploiting underlying physical principles. Many neural architectures are inherently based on physical systems – Boltzmann Machines and Hopfield Networks are derived from statistical mechanics (Smart & Zilman, 2021). In fact, training of neural networks can be recast as

a control problem where Hamiltonian dynamics emerge from the Pontryagin maximum principle; transformers have been modelled as interacting particle systems (Evens et al., 2021).

Instead of trying to find analytical solutions to complex differential equations, which is intractable in general, we model the dynamics of the ContiFormer architecture using forced damped harmonic oscillators (Flores-Hidalgo & Barone, 2011) because these systems provide closed-form solutions (Dutta et al., 2020). Furthermore, oscillators are a rich system which can be used to model dynamical systems (Herrero et al., 2012) – they have been used to solve Boolean SAT problems (Bashar et al., 2022), and have also been the inspiration for neural networks (Rohan et al., 2024) as well as state-of-the-art state-space models (Rusch & Rus, 2025).

We model attention as resonance behavior of a forced harmonic oscillator, where query-key similarity creates high attention when frequencies align and low attention when they are misaligned. This mapping works because attention in ContiFormer is fundamentally a time-windowed inner product between query and key trajectories. When we model keys using a damped oscillator, the subsequent integral becomes a resonance detector that measures spectral overlap weighted by the oscillator's transfer function $H(\omega) = \beta/(\omega_i^2 - \omega^2 + 2i\gamma_i\omega)$.

Overall, our work makes the following main contributions:

Firstly, we formulate a novel linear damped, driven harmonic oscillator analogy (with a closed-form solution) to replace the Neural ODE of the original ContiFormer. This helps us surmount the computational overhead of numerical solvers. We call our architecture "OsciFormer".

Secondly, we demonstrate that our Harmonic Oscillators can universally approximate any discrete attention matrix realizable by ContiFormer's continuous keys thus maintaining the expressivity of original architecture. In fact, we show that any continuous query function and any collection of continuous key functions defined on compact intervals, can be approximated arbitrarily well using a shared bank of harmonic oscillators with different initial conditions.

Thirdly, we discuss how the oscillator-based modeling would preserve equivariance properties of physical systems, which can be useful in spatiotemporal applications such as weather modelling. A detailed description of E(3)-equivariance is given in appendix C.

Finally, we provide the following detailed results: On event prediction, OsciFormer matches ContiFormer across six datasets in both accuracy and log-likelihood. On long-context UCR tasks it achieves top average accuracy (64.5%) with large margins on MI (91.8 ± 0.2), and on the clinical HR benchmark it obtains the lowest RMSE (2.56 ± 0.18) while ContiFormer runs out of memory. On synthetic irregular sequences, OsciFormer reaches 99.83 ± 0.32 accuracy with the fastest per-epoch time (0.56 min) among compared models.

Code: `https://anonymous.4open.science/anonymize/contiformer-2-C8EB`
Note: We have used LLMs to help reformat equations and text for LaTeX.

## 2 Preliminaries

Consider an irregular time series $\Gamma = \{(X_i, t_i)\}_{i=1}^N$ with ordered sampling times $0 \leq t_1 < t_2 < \cdots < t_N \leq T$, which represents observations from an underlying continuous-time process. This time series arises from sampling a continuous-time path $X \in \mathcal{C}(\mathbb{R}_+; \mathbb{R}^d)$, where $\mathcal{C}(\mathbb{R}_+; \mathbb{R}^d) = \{g : \mathbb{R}_+ \to \mathbb{R}^d \mid g \text{ continuous}\}$ denotes the space of continuous functions mapping non-negative reals to $d$-dimensional vectors. (Schirmer et al., 2022)

To model this using a standard Transformer (Vaswani et al., 2017), let $Q = [Q_1; \ldots; Q_N]$, $K = [K_1; \ldots; K_N]$, $V = [V_1; \ldots; V_N]$ denote the query, key, and value embeddings in the Transformer. However, simply dividing the time steps into equally sized intervals can damage the continuity of the data which is necessary for irregular time series modelling. To overcome the loss of temporal continuity caused by uniform time discretisation, ContiFormer (Chen et al., 2023) lets every observation $(X_i, t_i)$ initiate a continuous key/value trajectory governed by a NODE.

$$\mathbf{k}_i(t_i) = \mathbf{K}_i, \qquad \mathbf{k}_i(t) = \mathbf{k}_i(t_i) + \int_{t_i}^{t} f(\tau, \mathbf{k}_i(\tau); \boldsymbol{\theta}_k)\, d\tau,$$

$$\mathbf{v}_i(t_i) = \mathbf{V}_i, \qquad \mathbf{v}_i(t) = \mathbf{v}_i(t_i) + \int_{t_i}^{t} f(\tau, \mathbf{v}_i(\tau); \boldsymbol{\theta}_v)\, d\tau. \tag{1}$$

Subsequently, the discrete self-attention computed via the query–key dot-product is extended to its continuous-time counterpart by integrating the time-varying query and key trajectories: $\alpha_i(t) = \frac{1}{t-t_i} \int_{t_i}^{t} q(\tau)\, k_i(\tau)^\top\, d\tau$.

Herein, each layer computes attention between *all* $N$ queries and $N$ keys. For each of the $N^2$ pairs, the integral is approximated with a numerical solver like RK4, where each step evaluates two $d$-dimensional NODE vector fields, giving an $\mathcal{O}(d^2)$ cost per step. Thus one layer runs in $T_{\text{layer}} = \mathcal{O}(N^2 S d^2)$.

## 3 HARMONIC OSCILLATOR BASED MODELLING

Due to page limits, we provide our detailed derivation and model in appendix A. What follows here is a brief sketch.

We model the NODEs that govern keys and values in ContiFormer as *linear damped driven harmonic oscillators*. Keys are the solution of $\ddot{k}(t) + 2\gamma\dot{k}(t) + \omega^2 k(t) = F^k(t)$ where $\gamma \geq 0$ is the learnable damping coefficient, $\omega > 0$ is the learnable natural frequency, and $F^k(t)$ is the driving force. Likewise, values obey the same structure: $\ddot{v}(t) + 2\gamma_v \dot{v}(t) + \omega_v^2 v(t) = F^v(t)$ with independent learnable parameters $\gamma_v, \omega_v$ and value-intrinsic drive $F^v(t)$.

Our damped driven oscillators are governed by the second-order ODE $\ddot{x} + 2\gamma\dot{x} + \omega^2 x = F(t)$.

We first convert this into a first-order ODE like the ones governing the keys and values; to do this, we introduce the augmented state vector $z = \begin{bmatrix} x \\ p \end{bmatrix}, p = \frac{dx}{dt}$ and then write the second-order ODE in matrix form as

$$\frac{dz}{dt} = \underbrace{\begin{bmatrix} 0 & 1 \\ -\omega^2 & -2\gamma \end{bmatrix}}_{A} z + \underbrace{\begin{bmatrix} 0 \\ F(t) \end{bmatrix}}_{B(t)}. \tag{2}$$

We derive the general solution for any $t_0$, $z(t) = e^{A(t-t_0)}z(t_0) + \int_{t_0}^{t} e^{A(t-s)}B(s)\, ds$ with the first term $z_h(t)$ (homogeneous) and the second term $z_p(t)$ (particular). We subsequently handle $z_h(t)$ and $z_p(t)$ separately.

We first derive our homogeneous solution for $z_h(t) = e^{At}z_0$ by cases. Consider three cases: (1) Underdamped, $\gamma^2 < \omega^2 \quad (\gamma < \omega)$; (2) Critically damped: $\gamma^2 = \omega^2 \quad (\gamma = \omega)$; and, (3) Overdamped: $\gamma^2 > \omega^2 \quad (\gamma > \omega)$. This derivation is rather involved; we provide the details in appendix A.1.

We handle the particular solution $z_p(t) = \int_{t_0}^{t} e^{A(t-s)}B(s)\, ds$ similarly (appendix A.2), and then provide a steady state solution for the damped, driven oscillator (appendix A.3).

**Query:** We expand the interpolation function in the oscillator basis up to a suitable number of modes and obtain the coefficients $A_k$, $B_k$ by a least-squares fit. This circumvents the absence of a closed-form solution for the integral of the original cubic spline. $q(t) = \sum_{k=1}^{N}\Big(A_k \cos(\omega_k t) + B_k \sin(\omega_k t)\Big)$.

**Attention integral:** The complete derivation is available in appendix A.5. We compute the averaged attention coefficient $\alpha_i(t) = \frac{1}{\Delta} \int_{t_i}^{t} \langle q(\tau), k_i(\tau) \rangle\, d\tau, \Delta := t - t_i > 0$ when the (vector) key coordinates obey a *driven* damped oscillator, anchored at $t_i$ with zero particular state. The total key is $k_i = k_{i,\text{hom}} + k_{i,\text{part}} + c_i$, where the homogeneous part $k_{i,\text{hom}}$ was derived in section A.1, and here we add the driven part $k_{i,\text{part}}$. We then derive the steady-state solution for the driven

oscillator, for underdamped, critical, and overdamped driven keys, combining the steady-state and transient contributions to find the driven contribution to the averaged attention (equation 74) and the complete logit.

**Averaged attention: decomposition.** Using equation 51, equation 68, and equation 70 with $s \in [0, \Delta]$:

$$\int_{t_i}^{t} \langle q(\tau), k_{i,\mathrm{part}}(\tau) \rangle \, d\tau = \underbrace{\int_0^{\Delta} \langle q(t_i + s), x_{\mathrm{ss},i}(t_i + s) \rangle \, ds}_{\mathcal{I}_i^{(\mathrm{ss})}} +$$

$$\underbrace{\int_0^{\Delta} e^{-\gamma s} \langle q(t_i + s), E_i \cos(\omega_d s) + F_i \sin(\omega_d s) \rangle \, ds}_{\mathcal{I}_i^{(\mathrm{tr})}}. \qquad (3)$$

**Steady-state contribution $\mathcal{I}_i^{(\mathrm{ss})}$:** Using the undamped kernels equation 58–equation 61:

$$\mathcal{I}_i^{(\mathrm{ss})} = \sum_{j=1}^{J} \sum_{m=1}^{M_f} \Big[ \langle \tilde{A}_j, \widehat{C}_{i,m} \rangle \, I_{cc}(\Delta; 0, \omega_j, \varpi_m) + \langle \tilde{A}_j, \widehat{D}_{i,m} \rangle \, I_{cs}(\Delta; 0, \omega_j, \varpi_m)$$

$$+ \langle \tilde{B}_j, \widehat{C}_{i,m} \rangle \, I_{sc}(\Delta; 0, \omega_j, \varpi_m) + \langle \tilde{B}_j, \widehat{D}_{i,m} \rangle \, I_{ss}(\Delta; 0, \omega_j, \varpi_m) \Big]. \qquad (4)$$

**Transient contribution $\mathcal{I}_i^{(\mathrm{tr})}$.** Using the damped kernels equation 54–equation 57 with $\lambda_1 \in \{\omega_d\}$ and $\lambda_2 \in \{\omega_j\}$:

$$\mathcal{I}_i^{(\mathrm{tr})} = \sum_{j=1}^{J} \Big[ \langle E_i, \tilde{A}_j \rangle \, I_{cc}(\Delta; \gamma, \omega_d, \omega_j) + \langle E_i, \tilde{B}_j \rangle \, I_{cs}(\Delta; \gamma, \omega_d, \omega_j)$$

$$+ \langle F_i, \tilde{A}_j \rangle \, I_{sc}(\Delta; \gamma, \omega_d, \omega_j) + \langle F_i, \tilde{B}_j \rangle \, I_{ss}(\Delta; \gamma, \omega_d, \omega_j) \Big]. \qquad (5)$$

**Final result.** The driven contribution to the averaged attention is $\boxed{\alpha_i^{(\mathrm{driven})}(t) = \dfrac{1}{\Delta} \left( \mathcal{I}_i^{(\mathrm{ss})} + \mathcal{I}_i^{(\mathrm{tr})} \right)}$ where $\mathcal{I}_i^{(\mathrm{ss})}$ and $\mathcal{I}_i^{(\mathrm{tr})}$ are given by equation 4 and equation 5, respectively.

## 4 HARMONIC APPROXIMATION THEOREM

Due to page limits, we provide a detailed derivation in appendix B. What follows here is a brief sketch.

Start with a continuous function $f$ on $[a, b] \to$ Approximate it with trigonometric polynomials using Fejér $\to$ Shift the basis from $(t - a)$ to $(t - t_i)$ for each key. $\to$ Realize each term of the polynomial with an oscillator $\to$ Sum the oscillators to reconstruct the polynomial $\to$ Finally, show that the approximation error in keys leads to bounded error in attention weights using the Lipschitz property of softmax.

**Theorem 1.** *Let $q \in C([a, b]; \mathbb{R}^{d_k})$ and continuous keys $\{k_i\}_{i=1}^N$ with $k_i : [t_i, b] \to \mathbb{R}^{d_k}$. For any $\varepsilon > 0$ there exists an integer $M$ (depending on $\varepsilon$ and the keys) and a single shared oscillator bank on the fixed grid $\{\omega_n\}_{n=0}^M$ with $\gamma_n = 0$ such that one can choose initial states $\{z_{i,0}\}_{i=1}^N$ with the property*

$$\sup_{t \in [t_i, b]} \|k_i(t) - \tilde{k}_i(t)\|_2 < \varepsilon \quad \text{for all } i,$$

*where $\tilde{k}_i(t) := C e^{A(t - t_i)} z_{i,0}$ is the bank-generated key. Consequently, for all $j \geq i$,*

$$\left| \alpha_i(t_j; q, k_i) - \alpha_i(t_j; q, \tilde{k}_i) \right| \leq \|q\|_{\infty} \varepsilon, \qquad \|w(t_j) - \tilde{w}(t_j)\|_1 \leq \frac{\|q\|_{\infty}}{\sqrt{d_k}} \varepsilon.$$

*Proof.* Fix $\varepsilon > 0$. For each $i$, extend $k_i$ continuously from $[t_i, b]$ to $[a, b]$ (e.g., set $k_i(t) = k_i(t_i)$ for $t \in [a, t_i]$). Apply Lemma 2 to this extension to obtain a vector trigonometric polynomial

$$P_i(t) = c_{i,0} + \sum_{n=1}^{N_i} \left( c_{i,n} \cos \omega_n (t - a) + s_{i,n} \sin \omega_n (t - a) \right)$$

with $\sup_{t \in [a,b]} \|k_i(t) - P_i(t)\|_2 < \varepsilon/2$. Use Lemma 3 to rewrite $P_i$ as

$$P_i(t) = c_{i,0} + \sum_{n=1}^{N_i} \left( \tilde{c}_{i,n} \cos \omega_n (t - t_i) + \tilde{s}_{i,n} \sin \omega_n (t - t_i) \right).$$

Let $N := \max_i N_i$ and take $M \geq N$. By Lemma 4 (with $\gamma_n = 0$), choose $z_{i,0}$ so that the shared bank realizes $P_i$ *exactly*: $\tilde{k}_i(t) \equiv P_i(t)$ on $[t_i, b]$. Therefore $\sup_{t \in [t_i, b]} \|k_i(t) - \tilde{k}_i(t)\|_2 < \varepsilon/2 < \varepsilon$. For $t > t_i$,

$$|\alpha_i(t) - \tilde{\alpha}_i(t)| \leq \frac{1}{t - t_i} \int_{t_i}^{t} \|q(\tau)\|_2 \, \|k_i(\tau) - \tilde{k}_i(\tau)\|_2 \, d\tau \leq \|q\|_\infty \, \varepsilon.$$

At $t = t_i$ the bound $\left| \left\langle q(t_i), k_i(t_i) - \tilde{k}_i(t_i) \right\rangle \right| \leq \|q\|_\infty \, \varepsilon$ is immediate. Applying the softmax Lipschitz Lemma 6 to the logits scaled by $1/\sqrt{d_k}$ yields the stated $\ell_1$ bound. $\square$

**Corollary 1.** *Under the hypotheses of Theorem 1, fix $\varepsilon > 0$ and construct the undamped realization above. Then there exists $\bar{\gamma} > 0$ such that, for any damped bank with $0 \leq \gamma_n \leq \bar{\gamma}$, one can reuse the same initial states $\{z_{i,0}\}$ and obtain*

$$\sup_{t \in [t_i, b]} \|k_i(t) - \tilde{k}_i^{(\gamma)}(t)\|_2 < \varepsilon, \qquad \|w^{(\gamma)}(t_j) - w(t_j)\|_1 \leq \frac{\|q\|_\infty}{\sqrt{d_k}} \, \varepsilon,$$

*where the superscript $(\gamma)$ denotes readouts from the damped bank. In particular, a small amount of damping does not affect universality.*

## 5 COMPUTATIONAL COMPLEXITY

We analyze (i) arithmetic operations, (ii) sequential depth , and (iii) activation memory for one layer. All complexity bounds are per attention head.

### SETUP AND NOTATION

- $N$: sequence length.
- $d$: per-head feature dimension.
- $S$: number of vector-field/quadrature evaluations of the ODE solver on the normalized interval $[-1, 1]$ in one forward pass.
- $C_f(d)$: cost of one evaluation of the ODE vector field on a $d$-dimensional state; with dense linear maps, $C_f(d) = \Theta(d^2)$.
- The standard $Q, K, V$ projections cost $O(Nd^2)$ per head and are listed explicitly.

### 5.1 NUMERICAL CONTINUOUS-TIME REALIZATION (BASELINE)

Each position $i \in \{1, \ldots, N\}$ induces continuous key/value trajectories by solving an ODE on $[-1, 1]$. For every query–key pair $(i, j)$, the attention score is an integral of $\langle q_i(t), k_j(t) \rangle$ over $t \in [-1, 1]$, approximated by evaluating the ODE state and inner product at $S$ nodes. The total work across all pairs and steps is:

$$T_{\text{num}} = \Theta(N^2 S C_f(d)) + O(Nd^2) = \Theta(N^2 S d^2) + O(Nd^2),$$

$$D_{\text{num}} = \Theta(S),$$

$$M_{\text{num}} = \Theta(N^2 S d).$$

The first term in $T_{\text{num}}$ accounts for $N^2$ pairs, $S$ solver/quadrature nodes, and per-node cost $C_f(d) = \Theta(d^2)$. Depth is determined by the $S$ time steps on the critical path. Activation memory stores $d$-dimensional states for $S$ nodes per pair.

## 5.2 CLOSED-FORM REALIZATION

When the key/value ODEs admit closed forms, each query trajectory can be represented by a $J$-term trigonometric expansion so that the attention integral decomposes into $J$ modewise expressions, all evaluable in closed form. This yields:

$$T_{\text{cf}} = \Theta(N^2 Jd) + O(Nd^2),$$

$$D_{\text{cf}} = \Theta(1),$$

$$M_{\text{cf}} = \Theta(N^2 d).$$

Each query key pair involves computing $J$ mode coefficients with $d$-dimensional features, contributing $O(Jd)$ operations. The closed form eliminates time-stepping, yielding constant depth. Activation memory stores only $O(d)$ values per pair for backpropagation.

## 5.3 COMPARISON

Ignoring the shared projection term $O(Nd^2)$ and constants, the dominant cost ratio is

$$\frac{T_{\text{cf}}}{T_{\text{num}}} \asymp \frac{N^2 Jd}{N^2 Sd^2} = \frac{J}{Sd}.$$

The closed-form layer is asymptotically faster when $J \ll Sd$. It also achieves lower sequential depth by a factor $\Theta(S)$ and requires $\Theta(S)$-times less activation memory:

$$\frac{M_{\text{cf}}}{M_{\text{num}}} \asymp \frac{1}{S}.$$

## 5.4 REPRESENTATIVE INSTANCE

With $S = 80$ (e.g., fixed-step RK4 on $[-1, 1]$), $d = 64$, and $J = 8$,

$$\frac{T_{\text{cf}}}{T_{\text{num}}} = \frac{8}{80 \cdot 64} = \frac{1}{640},$$

meaning the dominant $N^2$ term is reduced by approximately three orders of magnitude, while sequential depth and activation memory decrease by a factor of $S$.

## 6 ARCHITECTURE

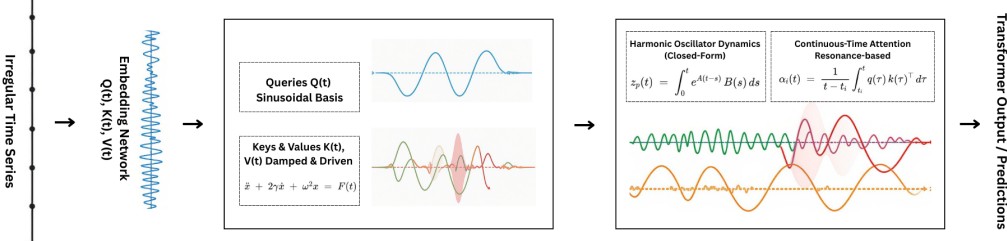

Figure 1: Architecture Pipeline

Each input generates an *oscillator* for its key and another for its value. Those oscillators evolve in continuous time with closed-form solutions. The projections for each key and value per head $h \in [H]$ with $d_h = d/H$ are given by $Q_i = W_Q X_i + b_Q$, $K_i = W_K X_i + b_K$, and $V_i = W_V X_i + b_V$ where $Q_i, K_i, V_i \in \mathbb{R}^{d_h}$.

Following this, the learnable parameters are: projection matrices and biases $W_Q, W_K, W_V, W_O, b_Q, b_K, b_V$; oscillator spectra (per head and channel, i.e. one learnable frequency $\omega$ and damping $\zeta$ for every coordinate c $= 1 \cdots$ d$_h$ inside each head) $\omega_h^k, \zeta_h^k \in \mathbb{R}_{>0}^{d_h}, \mathbb{R}_{\geq 0}^{d_h}$

for keys and $\omega_h^v, \zeta_h^v$ for values; initial-velocity maps $U_h^k, U_h^v \in \mathbb{R}^{d_h \times d_h}$; and, when intrinsic drives $F_h^{k/v}(t)$ are used, their matrices $A_h^{k/v}, B_h^{k/v} \in \mathbb{R}^{d_h \times d_h}$

The forward pass follows a plain Transformer (Vaswani et al., 2017) where for each head $h$ at time $t_j$ we project tokens to $Q_i, K_i, V_i$, compute the closed-form key and value trajectories $k_{i,h}(\tau), v_{i,h}(\tau)$ on $[t_i, t_j]$ for every $i \leq j$, fit the query expansion coefficients $(A_{j,\cdot}, B_{j,\cdot})$, evaluate the unnormalised scores $\alpha_{i,h}(t_j)$ in closed form or with a short integral average, softmax over $i \leq j$ to get weights $w_{i,h}(t_j)$, form the weighted value $\bar{v}_{i,h}(t_j)$ and emit $y_h(t_j) = \sum_{i \leq j} w_{i,h}(t_j)\bar{v}_{i,h}(t_j)$, then merge heads with $W_O$ and add residual plus layer-norm.

# 7 EXPERIMENTS

We evaluate on all irregular time-series benchmarks used across ContiFormer (Chen et al., 2023), Rough Transformers (Moreno-Pino et al., 2025), and Closed-Form Liquid Time-Constant Networks (Hasani et al., 2022) continuous models, spanning health, finance, event, sequential prediction, and synthetic (sine/spirals/XOR) settings. We adopt the UEA multivariate classification setting where irregularity is created by randomly dropping observations at ratios of 30%, 50%, and 70% per sample (Bagnall et al., 2018).

| Model | Metric | Synthetic | Neonate | Traffic | MIMIC | StackOverflow | BookOrder |
|---|---|---|---|---|---|---|---|
| HP (Laub et al., 2024) | LL (↑) | -3.084 ± .005 | -4.618 ± .005 | -1.482 ± .005 | -4.618 ± .005 | -5.794 ± .005 | -1.036 ± .000 |
| | Accuracy (↑) | 0.756 ± .000 | — | 0.570 ± .000 | 0.795 ± .000 | 0.441 ± .000 | 0.604 ± .000 |
| | RMSE (↓) | 0.953 ± .000 | 10.957 ± .012 | 0.407 ± .000 | 1.021 ± .000 | 1.341 ± .000 | 3.781 ± .000 |
| RMTPP (Du et al., 2016) | LL (↑) | -1.025 ± .030 | -2.817 ± .023 | -0.546 ± .012 | -1.184 ± .023 | -2.374 ± .001 | -0.952 ± .007 |
| | Accuracy (↑) | 0.841 ± .000 | — | 0.805 ± .002 | 0.823 ± .004 | 0.461 ± .000 | 0.624 ± .000 |
| | RMSE (↓) | 0.369 ± .014 | 9.517 ± .023 | 0.337 ± .001 | 0.864 ± .017 | 0.955 ± .000 | 3.647 ± .003 |
| NeuralHP (Shen & Cheng, 2025) | LL (↑) | -1.371 ± .004 | -2.795 ± .012 | -0.643 ± .004 | -1.239 ± .027 | -2.608 ± .000 | -1.104 ± .005 |
| | Accuracy (↑) | 0.841 ± .000 | — | 0.759 ± .000 | 0.814 ± .001 | 0.450 ± .000 | 0.621 ± .000 |
| | RMSE (↓) | 0.631 ± .002 | 9.614 ± .013 | 0.358 ± .001 | 0.846 ± .007 | 1.022 ± .000 | 3.734 ± .003 |
| GRU-Δt (Chung et al., 2014) | LL (↑) | -0.871 ± .050 | -2.736 ± .031 | -0.613 ± .062 | -1.164 ± .026 | -2.389 ± .002 | -0.915 ± .006 |
| | Accuracy (↑) | 0.841 ± .000 | — | 0.800 ± .004 | 0.832 ± .007 | 0.466 ± .000 | 0.627 ± .000 |
| | RMSE (↓) | 0.249 ± .013 | 9.421 ± .050 | 0.335 ± .001 | 0.850 ± .010 | 0.950 ± .000 | 3.666 ± .016 |
| ODE-RNN (Rubanova et al., 2019) | LL (↑) | -1.032 ± .102 | -2.732 ± .080 | -0.491 ± .011 | -1.183 ± .028 | -2.395 ± .001 | -0.988 ± .006 |
| | Accuracy (↑) | 0.841 ± .000 | — | 0.812 ± .006 | 0.827 ± .006 | 0.467 ± .000 | 0.624 ± .000 |
| | RMSE (↓) | 0.342 ± .030 | 9.289 ± .048 | 0.334 ± .000 | 0.865 ± .021 | 0.952 ± .000 | **3.605 ± .004** |
| mTAN Shukla & Marlin (2021) | LL (↑) | -0.920 ± .036 | -2.722 ± .026 | -0.548 ± .023 | -1.149 ± .029 | -2.391 ± .002 | -0.980 ± .004 |
| | Accuracy (↑) | 0.842 ± .000 | — | 0.811 ± .002 | 0.832 ± .009 | 0.466 ± .000 | 0.620 ± .000 |
| | RMSE (↓) | 0.286 ± .008 | 9.363 ± .042 | 0.334 ± .001 | 0.848 ± .006 | 0.950 ± .000 | 3.680 ± .015 |
| **ContiFormer**(Chen et al., 2023) | LL (↑) | **-0.535 ± .028**[+] | **-2.550 ± .026** | **0.635 ± .019**[+] | -1.135 ± .023 | **-2.332 ± .001**[+] | **-0.270 ± .010**[+] |
| | Accuracy (↑) | **0.842 ± .000** | — | **0.822 ± .001**[+] | **0.836 ± .006** | **0.473 ± .000**[+] | **0.628 ± .001**[+] |
| | RMSE (↓) | **0.192 ± .005** | 9.233 ± .033 | **0.328 ± .001**[+] | **0.837 ± .007** | **0.948 ± .000**[+] | 3.614 ± .020 |
| **OsciFormer (Ours)** | LL (↑) | -0.558 ± .025[+] | -2.573 ± .028 | 0.612 ± .022[+] | -1.142 ± .021 | -2.315 ± .002[+] | -0.288 ± .009[+] |
| | Accuracy (↑) | 0.841 ± .000 | — | 0.819 ± .001[+] | 0.834 ± .007 | 0.471 ± .000[+] | 0.626 ± .001[+] |
| | RMSE (↓) | 0.198 ± .006 | 9.187 ± .031 | 0.331 ± .001[+] | 0.841 ± .008 | 0.951 ± .000[+] | 3.621 ± .017 |

Table 1: Performance comparison of different models on event prediction tasks. Results shown for log-likelihood (LL) and accuracy (ACC) metrics. Arrow symbols ↑ and ↓ denote whether higher or lower values represent superior performance, respectively. For comparison, other values in Table are sourced from (Chen et al., 2023) reported benchmarks.

| Dataset | LRU | S5 | S6 | Mamba | NCDE | NRDE | LogNCDE | Transformer | RFormer | **OsciFormer** |
|---|---|---|---|---|---|---|---|---|---|---|
| SCP1 | 82.6 ± 3.4 | **89.9 ± 4.6** | 82.8 ± 2.7 | 80.7 ± 1.4 | 79.8 ± 5.6 | 80.9 ± 2.5 | 83.1 ± 2.8 | 84.3 ± 6.3 | 81.2 ± 2.8 | 84.1 ± 3.0 |
| SCP2 | 51.2 ± 3.6 | 50.5 ± 2.6 | 49.9 ± 9.5 | 48.2 ± 3.9 | 53.0 ± 2.8 | **53.7 ± 6.9** | 53.7 ± 4.1 | 49.1 ± 2.5 | 52.3 ± 3.7 | 58.7 ± 6.8 |
| MI | 48.4 ± 5.0 | 47.7 ± 5.5 | 51.3 ± 4.7 | 47.7 ± 4.5 | 49.5 ± 2.8 | 47.0 ± 5.7 | 53.7 ± 5.3 | 50.5 ± 3.0 | **55.8 ± 6.6** | 91.8 ± 0.2 |
| EW | 87.8 ± 2.8 | 81.1 ± 3.7 | 85.0 ± 16.1 | 70.9 ± 15.8 | 75.0 ± 3.9 | 83.9 ± 7.3 | 85.6 ± 5.1 | OOM | **90.3 ± 0.1** | 48.9 ± 3.4 |
| ETC | 21.5 ± 2.1 | 24.1 ± 4.3 | 26.4 ± 6.4 | 27.9 ± 4.5 | 29.9 ± 6.5 | 25.3 ± 1.8 | 34.4 ± 6.4 | **40.5 ± 6.3** | 34.7 ± 4.1 | 31.5 ± 4.6 |
| HB | **78.4 ± 6.7** | 77.7 ± 5.5 | 76.5 ± 8.3 | 76.2 ± 3.8 | 73.9 ± 2.6 | 72.9 ± 4.8 | 75.2 ± 4.6 | 70.5 ± 0.1 | 72.5 ± 0.1 | 71.8 ± 0.1 |
| Av. | 61.7 | 61.8 | 62.0 | 58.6 | 60.2 | 60.6 | 64.3 | 59.0 | **64.5** | **64.5** |

Table 2: Classification performance on various long context temporal datasets from UCR TS archive (Tan et al., 2020). For comparison, other values in Table are sourced from (Moreno-Pino et al., 2025) reported benchmarks.

We also evaluate on next-event type and time prediction across different datasets (see Table 1): Neonate (clinical seizures), Traffic (PeMS events), MIMIC (ICU visits), BookOrder (financial limit order book transactions for "buy/sell"), and StackOverflow (badge events). Following Hasani et al. (2022), we run experiments on irregularly sampled clinical time series over the first 48 hours in ICU with missing features across 37 channels (see Table 2).

| Model | HR (RMSE ↓) |
|---|---|
| ODE-RNN$^{\diamond}$ | $13.06 \pm 0.00$ |
| Neural-CDE$^{\diamond}$ | $9.82 \pm 0.34$ |
| Neural-RDE$^{\diamond}$ | $2.97 \pm 0.45$ |
| GRU$^{\dagger}$ | $13.06 \pm 0.00$ |
| ODE-RNN$^{\dagger}$ | $13.06 \pm 0.00$ |
| Neural-RDE$^{\dagger}$ | $4.04 \pm 0.11$ |
| Transformer | $8.24 \pm 2.24$ |
| ContiFormer | Out of memory |
| RFormer | $2.66 \pm 0.21$ |
| **OsciFormer** | $2.56 \pm 0.18$ |

Table 3: Evaluation on HR dataset (lower RMSE is better). For comparison, other values in Table are sourced from (Moreno-Pino et al., 2025) reported benchmarks.

| Model | Equidistant encoding | Event-based (irregular) encoding | Epoch Time (min) | ODE-based? |
|---|---|---|---|---|
| †Augmented LSTM (20) | $100.00\% \pm 0.00$ | $89.71\% \pm 3.48$ | 0.62 | No |
| † CT-GRU (34) | $100.00\% \pm 0.00$ | $61.36\% \pm 4.87$ | 0.80 | No |
| † RNN Decay (7) | $60.28\% \pm 19.87$ | $75.53\% \pm 5.28$ | 0.90 | No |
| † Bi-directional RNN (38) | $100.00\% \pm 0.00$ | $90.17\% \pm 0.69$ | 1.82 | No |
| † GRU-D (36) | $100.00\% \pm 0.00$ | $97.90\% \pm 1.71$ | 0.58 | No |
| † CT-LSTM (35) | $97.73\% \pm 0.08$ | $95.09\% \pm 0.30$ | 0.86 | No |
| † ODE-RNN (7) | $50.47\% \pm 0.06$ | $51.21\% \pm 0.37$ | 4.11 | Yes |
| † CT-RNN (33) | $50.42\% \pm 0.12$ | $50.79\% \pm 0.34$ | 4.83 | Yes |
| † GRU-ODE (7) | $50.41\% \pm 0.40$ | $52.52\% \pm 0.35$ | 1.55 | Yes |
| † ODE-LSTM (9) | $100.00\% \pm 0.00$ | $98.89\% \pm 0.26$ | 1.18 | Yes |
| LTC (1) | $100.00\% \pm 0.00$ | $49.11\% \pm 0.00$ | 2.67 | Yes |
| ContiFormer | $100.00\% \pm 0.00$ | $99.93\% \pm 0.12$ | 3.83 | Yes |
| **OsciFormer** | $100.00\% \pm 0.00$ | $99.83\% \pm 0.32$ | **0.56** | No |

Table 4: Detailed accuracy and time comparison including encoding types

Finally, we evaluate on synthetic datasets with binary sequence classification in two encodings: equidistant (regular) and event-based (irregular, only bit-change events). We also test interpolation and extrapolation on 2-D spiral trajectories with irregular time points- refer to figures 2c and 2d.

Across the irregular-benchmark suite (health/HR, finance/LOB-style streams, and synthetic sine – see Table 3), we observe that setting $J = 8$ (i.e., the number of oscillator modes) yields essentially *identical predictive performance* to larger settings. In the tasks where $J$ indexes the oscillator modes in our module, accuracy saturates around $J \in [6, 8]$ with no meaningful gains beyond that range. At the same time, we obtain consistent computational benefits relative to the ODE-based Contiformer, with the largest speedups on the longest or most irregular sequences. **These gains vary from 3x to 20x** depending on benchmarks and value of the Oscillator mode – see Table 4 for these results.

Furthermore, we establish the following hyperparameter configuration. For optional driven dynamics we use a collocation–matched, causal sinusoidal drive per head $h$ and token $i$: $F^{k/v}i, h(t) = \sum m = 1^M \left( g^{k/v}h, m \odot E^{k/v}i, h \right) \cos\left(\varpi_{h,m}(t - t_i)\right) + \left( h^{k/v}h, m \odot E^{k/v}i, h \right) \sin\left(\varpi_{h,m}(t - t_i)\right)$ for $t \geq t_i$, where $E^k i, h = Ki, h$ and $E^v i, h = Vi, h$ are the per–head projections, $g^{k/v}h, m, h^{k/v}h, m \in \mathbb{R}^{d_h}$ are learnable element-wise gains, and $\varpi_{h,m}$ are drive frequencies drawn from the collocation bank $\{\omega_1, \ldots, \omega_J\}$ (we use $J = 8$). This choice admits closed–form solutions via the transfer function $H(\omega)$ and aligns the forcing spectrum with the query basis.

For model architecture, we use width $d = 256$ and $H = 8$ attention heads, where $d_h = d/H$. For training, we apply ridge regularization for the query fit, dropout in projections and feed-forward

networks, and weight decay through the optimizer. We use AdamW with learning rate $1 \times 10^{-3}$ and weight decay $0.01$, employing cosine decay with 5% warmup. Parameters are initialized with $\omega$ log-uniform on $[10^{-2}, 10^1]$ (normalized time), damping $\zeta \in [0.05, 0.4]$, and $U_h^{k/v} = 0$. This configuration consistently delivers optimal performance across our benchmark suite.

We have conducted a detailed set of ablations over (i) the number of oscillator modes (J) (ii) different damping ranges (iii) several frequency grid parameterizations (see Tables in Appendix E.1). To visualize the resonance view of attention, we have conducted simple irregular time-series based classification and regression experiments, given in E.2 and E.3.

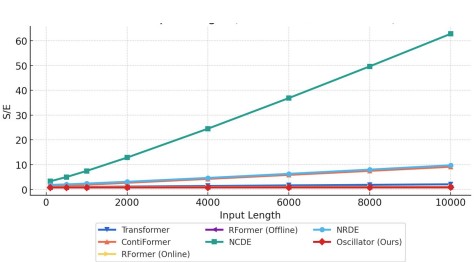

(a) Per-epoch Training Time vs. Input Length by Model Type

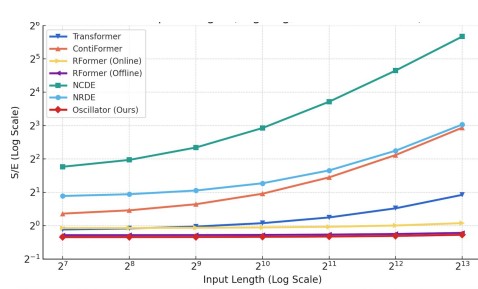

(b) Per-epoch Training Time vs. Input Length by Model Type (log scale)

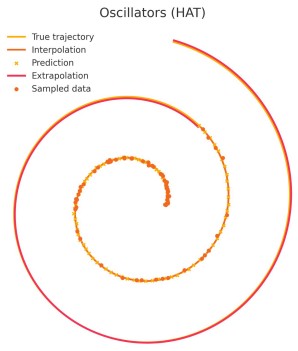

(c) Osciformer samples and predictions

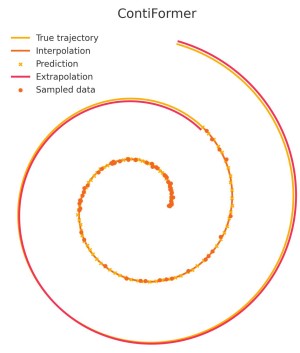

(d) ContiFormer samples and predictions

Figure 2: Trajectories and Training Time Visualisations

## 8 DISCUSSION

We replaced the continuous-time dynamics of Contiformer with a linear, damped, driven oscillator. This keeps the continuous-time property intact while requiring only a handful of closed-form operations per step, eliminating the memory blow-up that plagues the standard Contiformer and delivering accuracy on par with structured state-space models. We proved that a bank of damped oscillators reproduces key-value signals exactly and faithfully approximates discrete attention. The generalization bounds we provide are only a first step and can be tightened further, which could lead to an even richer and more accurate model family. Furthermore, stacking multiple oscillators provides a principled way to recreate every primitive of a standard transformer, opening a concrete pathway toward a universal approximation theorem for transformers while simultaneously revealing the class of functions that such oscillators and transformers more broadly cannot approximate. This can help us understand the bounds of current transformers and help us develop better architectures for more efficient representation. We also think oscillators provide a viewpoint beyond time series. The same physical viewpoint allows us to embed oscillators inside large language models, using frequency, damping, and forcing terms to model how meaning vibrates across semantic dimensions and providing a new class of physically grounded representations for LLMs.

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

## A  HARMONIC OSCILLATOR BASED MODELLING

As discussed earlier, we model the NODEs that govern keys and values in ContiFormer as *linear damped driven harmonic oscillators*.

Keys are the solution of $\ddot{k}(t) + 2\gamma\dot{k}(t) + \omega^2 k(t) = F^k(t)$ where $\gamma \geq 0$ is the learnable damping coefficient, $\omega > 0$ the learnable natural frequency, and $F^k(t)$ is the driving force. Likewise, values obey the same structure: $\ddot{v}(t) + 2\gamma_v\dot{v}(t) + \omega_v^2 v(t) = F^v(t)$ with independent learnable parameters $\gamma_v, \omega_v$ and value-intrinsic drive $F^v(t)$.

The following damped driven oscillators are governed by the second-order ODE

$$\ddot{x} + 2\gamma\dot{x} + \omega^2 x \;=\; F(t). \tag{6}$$

To convert this to a first-order ODE like the ones above governing the keys and values, introduce the augmented state vector

$$z = \begin{bmatrix} x \\ p \end{bmatrix}, \qquad p = \frac{dx}{dt}.$$

Using this, the second-order ODE can be written in matrix form as

$$\frac{dz}{dt} \;=\; \underbrace{\begin{bmatrix} 0 & 1 \\ -\omega^2 & -2\gamma \end{bmatrix}}_{A} z \;+\; \underbrace{\begin{bmatrix} 0 \\ F(t) \end{bmatrix}}_{B(t)}. \tag{7}$$

The solution to this can be found using the variation of parameters method. We start with the following.

$$\frac{dz}{dt} = Az + B(t). \tag{8}$$

The homogeneous version is

$$\frac{dz_h}{dt} = Az_h \;\Rightarrow\; z_h(t) = Ce^{At} \quad \text{for some constant vector } C.$$

To find a particular solution, try

$$z_p(t) = u(t)\,e^{At} \qquad \text{(variation of parameters; let the constant become a function } u(t)).$$

Then

$$\frac{dz_p}{dt} = \frac{d}{dt}\left(u(t)e^{At}\right) = Ae^{At}u(t) + e^{At}\frac{du}{dt}.$$

Plugging into the original ODE,

$$\frac{dz_p}{dt} = Az_p + B(t) \;\Rightarrow\; Ae^{At}u(t) + e^{At}\frac{du}{dt} = Ae^{At}u(t) + B(t),$$

hence

$$e^{At}\frac{du}{dt} = B(t) \;\Rightarrow\; \frac{du}{dt} = e^{-At}B(t).$$

Therefore

$$u(t) = \int_0^t e^{-A\tau}B(\tau)\,d\tau + u(0),$$

and

$$z_p(t) = e^{At}u(t) = e^{At}\left(\int_0^t e^{-A\tau}B(\tau)\,d\tau + u(0)\right) = e^{At}\int_0^t e^{-A\tau}B(\tau)\,d\tau + e^{At}u(0).$$

We can set $u(0) = 0$ without loss of generality, giving

$$z_p(t) = e^{At}\int_0^t e^{-A\tau}B(\tau)\,d\tau. \tag{9}$$

To solve this further we change variables: Let $\tau = t - s \Rightarrow d\tau = -ds$. When $\tau = 0 \Rightarrow s = t$, and when $\tau = t \Rightarrow s = 0$. Then

$$\int_0^t e^{-A\tau}B(\tau)\,d\tau = \int_{s=t}^{s=0} e^{-A(t-s)}B(t-s)\,(-ds) = \int_{s=0}^{s=t} e^{-A(t-s)}B(t-s)\,ds$$

$$= \int_0^t e^{-At}\,e^{As}B(t-s)\,ds.$$

Hence

$$z_p(t) \;=\; e^{At}e^{-At}\int_0^t e^{As}B(t-s)\,ds \;=\; \int_0^t e^{A(t-s)}B(s)\,ds,$$

where in the last step we renamed the dummy variable. Thus, the general solution for any $t_0$,

$$z(t) \;=\; e^{A(t-t_0)}z(t_0) \;+\; \int_{t_0}^t e^{A(t-s)}B(s)\,ds, \tag{10}$$

with the first term $z_h(t)$ (homogeneous) and the second term $z_p(t)$ (particular).

## A.1 HOMOGENEOUS SOLUTION $z_h(t) = e^{At}z_0$ BY CASES

We will find $z_h(t)$ and $z_p(t)$ separately. Consider three cases:

1. (1) Underdamped: $\gamma^2 < \omega^2 \quad (\gamma < \omega)$
2. (2) Critically damped: $\gamma^2 = \omega^2 \quad (\gamma = \omega)$
3. (3) Overdamped: $\gamma^2 > \omega^2 \quad (\gamma > \omega)$

**Eigenvalues of $A$:**

$$\det(A - \lambda I) = \begin{vmatrix} -\lambda & 1 \\ -\omega^2 & -2\gamma - \lambda \end{vmatrix} = (-\lambda)(-2\gamma - \lambda) + \omega^2 = \lambda^2 + 2\gamma\lambda + \omega^2 = 0,$$

so

$$\lambda_{1,2} = -\gamma \pm \sqrt{\gamma^2 - \omega^2}.$$

### A.1.1 CASE I: $\gamma < \omega$ (UNDERDAMPED)

Let $\omega_d = \sqrt{\omega^2 - \gamma^2}$, then $\lambda_{1,2} = -\gamma \pm i\omega_d$.

**Eigenvectors.** For $\lambda_1 = -\gamma + i\omega_d$,

$$(A - \lambda_1 I) = \begin{bmatrix} \gamma - i\omega_d & 1 \\ -\omega^2 & -\gamma - i\omega_d \end{bmatrix}$$

$$\implies (\gamma - i\omega_d)x + y = 0, \quad -\omega^2 x + (-\gamma - i\omega_d)y = 0$$

so one eigenvector is

$$v_1 = \begin{bmatrix} 1 \\ -\gamma + i\omega_d \end{bmatrix}.$$

For $\lambda_2 = -\gamma - i\omega_d$,

$$v_2 = \begin{bmatrix} 1 \\ -\gamma - i\omega_d \end{bmatrix}.$$

Collect the eigenvectors in

$$V = \begin{bmatrix} 1 & 1 \\ -\gamma + i\omega_d & -\gamma - i\omega_d \end{bmatrix}.$$

*The matrix $V$ is complex but the state is real; since $v_2 = \overline{v_1}$ we can form a real basis from $\Re(v_1)$ and $\Im(v_1)$:*

$$\Re(v_1) = \begin{bmatrix} 1 \\ -\gamma \end{bmatrix}, \quad \Im(v_1) = \begin{bmatrix} 0 \\ \omega_d \end{bmatrix} \Rightarrow V_{\mathbb{R}} = \begin{bmatrix} 1 & 0 \\ -\gamma & \omega_d \end{bmatrix}, \quad V_{\mathbb{R}}^{-1} = \frac{1}{\omega_d} \begin{bmatrix} \omega_d & 0 \\ \gamma & 1 \end{bmatrix}.$$

In this real basis,

$$A \sim V_{\mathbb{R}}^{-1} A V_{\mathbb{R}} = \begin{bmatrix} -\gamma & \omega_d \\ -\omega_d & -\gamma \end{bmatrix} = -\gamma I + B, \qquad B = \begin{bmatrix} 0 & \omega_d \\ -\omega_d & 0 \end{bmatrix}.$$

Since $I$ and $B$ commute,

$$\exp\big((-\gamma I + B)t\big) = e^{-\gamma t}\exp(Bt).$$

To find $\exp(Bt)$, we compute successive powers of $B$:

$$B^2 = -\omega_d^2 I, \quad B^3 = -\omega_d^2 B, \quad B^4 = \omega_d^4 I, \quad \Rightarrow \quad B^{2k} = (-1)^k \omega_d^{2k} I, \ \ B^{2k+1} = (-1)^k \omega_d^{2k} B.$$

Therefore the matrix exponential series is:

$$\exp(Bt) = \sum_{n=0}^{\infty} \frac{(Bt)^n}{n!} = \sum_{k=0}^{\infty} \frac{B^{2k} t^{2k}}{(2k)!} + \sum_{k=0}^{\infty} \frac{B^{2k+1} t^{2k+1}}{(2k+1)!}$$

$$= I \sum_{k=0}^{\infty} \frac{(-1)^k (\omega_d t)^{2k}}{(2k)!} + \frac{B}{\omega_d} \sum_{k=0}^{\infty} \frac{(-1)^k (\omega_d t)^{2k+1}}{(2k+1)!}$$

$$= I \cos(\omega_d t) + \frac{B}{\omega_d} \sin(\omega_d t) = \begin{bmatrix} \cos(\omega_d t) & \sin(\omega_d t) \\ -\sin(\omega_d t) & \cos(\omega_d t) \end{bmatrix}.$$

Thus

$$e^{At} = V_{\mathbb{R}} \, e^{-\gamma t} \begin{bmatrix} \cos(\omega_d t) & \sin(\omega_d t) \\ -\sin(\omega_d t) & \cos(\omega_d t) \end{bmatrix} V_{\mathbb{R}}^{-1}.$$

Multiplying out gives the standard real form

$$e^{At} = e^{-\gamma t} \begin{bmatrix} \cos(\omega_d t) + \dfrac{\gamma}{\omega_d} \sin(\omega_d t) & \dfrac{\sin(\omega_d t)}{\omega_d} \\ -\dfrac{\omega^2}{\omega_d} \sin(\omega_d t) & \cos(\omega_d t) - \dfrac{\gamma}{\omega_d} \sin(\omega_d t) \end{bmatrix}. \tag{11}$$

Hence, for the homogeneous motion,

$$z_h(t) = e^{At} z_0 = e^{-\gamma t} \begin{bmatrix} \cos(\omega_d t) + \dfrac{\gamma}{\omega_d} \sin(\omega_d t) & \dfrac{\sin(\omega_d t)}{\omega_d} \\ -\dfrac{\omega^2}{\omega_d} \sin(\omega_d t) & \cos(\omega_d t) - \dfrac{\gamma}{\omega_d} \sin(\omega_d t) \end{bmatrix} z_0.$$

### A.1.2  CASE II: $\gamma = \omega$ (CRITICAL DAMPING) — JORDAN FORM

Here $\lambda_{1,2} = -\gamma$ (repeated eigenvalue). Algebraic multiplicity 2, geometric multiplicity 1, so we need a Jordan block.

Eigenvector $v_1$ satisfies

$$(A - \lambda I)v_1 = (A + \gamma I)v_1 = 0, \quad (A + \gamma I) = \begin{bmatrix} \gamma & 1 \\ -\omega^2 & -\gamma \end{bmatrix} = \begin{bmatrix} \gamma & 1 \\ -\gamma^2 & -\gamma \end{bmatrix} \Rightarrow v_1 = \begin{bmatrix} 1 \\ -\gamma \end{bmatrix}.$$

For the generalized eigenvector $v_2$, we solve

$$(A - \lambda I)v_2 = v_1 \iff (A + \gamma I)v_2 = v_1 \Rightarrow \begin{bmatrix} \gamma & 1 \\ -\gamma^2 & -\gamma \end{bmatrix} \begin{bmatrix} v_{2,1} \\ v_{2,2} \end{bmatrix} = \begin{bmatrix} 1 \\ -\gamma \end{bmatrix}.$$

From the first equation, $\gamma v_{2,1} + v_{2,2} = 1$. Choose $v_{2,1} = 0 \Rightarrow v_{2,2} = 1$; hence

$$v_2 = \begin{bmatrix} 0 \\ 1 \end{bmatrix}.$$

Let

$$P = [v_1 \; v_2] = \begin{bmatrix} 1 & 0 \\ -\gamma & 1 \end{bmatrix}, \qquad P^{-1} = \begin{bmatrix} 1 & 0 \\ \gamma & 1 \end{bmatrix}.$$

Jordan normal form:

$$J = P^{-1} A P = \begin{bmatrix} -\gamma & 1 \\ 0 & -\gamma \end{bmatrix} \quad \text{(a } 2 \times 2 \text{ Jordan block with } \lambda = -\gamma).$$

For a Jordan block,

$$e^{Jt} = e^{\lambda t} \begin{bmatrix} 1 & t \\ 0 & 1 \end{bmatrix} = e^{-\gamma t} \begin{bmatrix} 1 & t \\ 0 & 1 \end{bmatrix}.$$

Therefore

$$e^{At} = P\,e^{Jt}\,P^{-1} = e^{-\gamma t} \begin{bmatrix} 1 & 0 \\ -\gamma & 1 \end{bmatrix} \begin{bmatrix} 1 & t \\ 0 & 1 \end{bmatrix} \begin{bmatrix} 1 & 0 \\ \gamma & 1 \end{bmatrix}$$

$$= e^{-\gamma t} \begin{bmatrix} 1+\gamma t & t \\ -\gamma^2 t & 1-\gamma t \end{bmatrix}.$$

Thus, for the homogeneous motion in the critically–damped case,

$$z_h(t) = e^{At} z_0 = e^{-\gamma t} \begin{bmatrix} 1+\gamma t & t \\ -\gamma^2 t & 1-\gamma t \end{bmatrix} z_0.$$

### A.1.3 CASE III: $\gamma > \omega$ (OVERDAMPED)

Real, distinct eigenvalues $\lambda_{1,2} = -\gamma \pm \sqrt{\gamma^2 - \omega^2} = -\gamma \pm \sigma$, where $\sigma = \sqrt{\gamma^2 - \omega^2}$
Let us find the two eigenvectors.
For $\lambda_1 = -\gamma + \sigma$:

$$(A - \lambda_1 I)v_1 = \begin{bmatrix} \gamma - \sigma & 1 \\ -\omega^2 & -\gamma - \sigma \end{bmatrix} \begin{bmatrix} v_{1,1} \\ v_{1,2} \end{bmatrix} = 0$$

From the first equation, $(\gamma - \sigma)v_{1,1} + v_{1,2} = 0$:

$$\implies v_1 = \begin{bmatrix} 1 \\ -\gamma + \sigma \end{bmatrix}$$

Similarly, for $\lambda_2 = -\gamma - \sigma$:

$$v_2 = \begin{bmatrix} 1 \\ -\gamma - \sigma \end{bmatrix}$$

Let

$$P = [v_1 \quad v_2] = \begin{bmatrix} 1 & 1 \\ -\gamma + \sigma & -\gamma - \sigma \end{bmatrix}, \qquad P^{-1} = \frac{-1}{2\sigma} \begin{bmatrix} -\gamma - \sigma & -1 \\ \gamma - \sigma & 1 \end{bmatrix}$$

Finally,

$$e^{At} = P \begin{bmatrix} e^{\lambda_1 t} & 0 \\ 0 & e^{\lambda_2 t} \end{bmatrix} P^{-1}$$

$$\implies e^{At} = \begin{bmatrix} 1 & 1 \\ -\gamma + \sigma & -\gamma - \sigma \end{bmatrix} \begin{bmatrix} e^{(-\gamma+\sigma)t} & 0 \\ 0 & e^{(-\gamma-\sigma)t} \end{bmatrix} \begin{bmatrix} \frac{\gamma+\sigma}{2\sigma} & \frac{1}{2\sigma} \\ \frac{-\gamma+\sigma}{2\sigma} & \frac{-1}{2\sigma} \end{bmatrix}$$

Using,

$$\cosh(\sigma t) = \frac{e^{\sigma t} + e^{-\sigma t}}{2}$$

$$\sinh(\sigma t) = \frac{e^{\sigma t} - e^{-\sigma t}}{2}$$

We get the homogeneous motion in the overdamped case,

$$z_h(t) = e^{At} z_0 = e^{-\gamma t} \begin{bmatrix} \cosh(\sigma t) + \frac{\gamma}{\sigma}\sinh(\sigma t) & \frac{\sinh(\sigma t)}{\sigma} \\ -\frac{\omega^2}{\sigma}\sinh(\sigma t) & \cosh(\sigma t) - \frac{\gamma}{\sigma}\sinh(\sigma t) \end{bmatrix} z_0.$$

### A.2 PARTICULAR SOLUTION $z_p(t) = \int_{t_0}^t e^{A(t-s)} B(s)\,ds$ BY CASES

Now we calculate the particular solution for the three damping cases. For the forced system

$$\dot{z}(t) = Az(t) + Bf(t),$$

the solution is

$$z(t) = e^{At} z_0 + \int_0^t e^{A(t-s)} B f(s)\,\mathrm{d}s. \tag{12}$$

We define the (matrix) Green's function

$$G(t, s) = e^{A(t-s)} B. \tag{13}$$

The particular solution is then the convolution

$$z_p(t) = \int_0^t G(t, s) f(s) \, \mathrm{d}s. \tag{14}$$

For our system

$$A = \begin{bmatrix} 0 & 1 \\ -\omega^2 & -2\gamma \end{bmatrix}, \qquad B = \begin{bmatrix} 0 \\ 1 \end{bmatrix},$$

we have

$$G(t, s) = e^{A(t-s)} \begin{bmatrix} 0 \\ 1 \end{bmatrix}. \tag{15}$$

Let

$$z(t) = \begin{bmatrix} x(t) \\ \dot{x}(t) \end{bmatrix}, \qquad \dot{z}(t) = Az(t) + Bf(t).$$

From equation 12,

$$z_p(t) = \int_0^t e^{A(t-s)} \begin{bmatrix} 0 \\ F(s) \end{bmatrix} \mathrm{d}s = \int_0^t e^{A(t-s)} \begin{bmatrix} 0 \\ \alpha \, f(s) \end{bmatrix} \mathrm{d}s, \tag{16}$$

where the driving force is given by $F(s) = \alpha \, f(s)$, with

$$f(s) = \sum_{j=1}^J \Big( A_j \cos(\omega_j s) + B_j \sin(\omega_j s) \Big). \tag{17}$$

By linearity, we can compute the response to each mode separately and then sum.

Starting from equation 16 and letting $\tau = t - s$ (so $s = t - \tau$, $\mathrm{d}s = -\mathrm{d}\tau$),

$$z_p(t) = \int_t^0 e^{A\tau} \begin{bmatrix} 0 \\ \alpha \, f(t - \tau) \end{bmatrix} (-\mathrm{d}\tau) = \int_0^t e^{A\tau} \begin{bmatrix} 0 \\ \alpha \, f(t - \tau) \end{bmatrix} \mathrm{d}\tau. \tag{18}$$

Write

$$e^{A\tau} = \begin{bmatrix} g_{11}(\tau) & g_{12}(\tau) \\ g_{21}(\tau) & g_{22}(\tau) \end{bmatrix}. \tag{19}$$

Since the forcing appears only in the second component,

$$z_p(t) = \int_0^t \begin{bmatrix} g_{12}(\tau) \, \alpha \, f(t - \tau) \\ g_{22}(\tau) \, \alpha \, f(t - \tau) \end{bmatrix} \mathrm{d}\tau. \tag{20}$$

Because $z_p(t) = \begin{bmatrix} x_p(t) \\ \dot{x}_p(t) \end{bmatrix}$ and we are only concerned with $x(t)$,

$$x_p(t) = \int_0^t g_{12}(\tau) \, \alpha \, f(t - \tau) \, \mathrm{d}\tau. \tag{21}$$

Take $f(s) = \cos(\omega_j s) \Rightarrow f(t - \tau) = \cos\big(\omega_j (t - \tau)\big)$.

### A.2.1  CASE I: $\gamma < \omega$ (UNDERDAMPED)

From the homogeneous analysis,

$$g_{12}(\tau) = e^{-\gamma\tau}\,\frac{\sin(\omega_d\tau)}{\omega_d}, \qquad \omega_d := \sqrt{\omega^2 - \gamma^2}. \tag{22}$$

Hence

$$x_p(t) = \alpha \int_0^t e^{-\gamma\tau}\frac{\sin(\omega_d\tau)}{\omega_d}\,\cos(\omega_j(t-\tau))\,\mathrm{d}\tau. \tag{23}$$

Using $\cos(a-b) = \cos a \cos b + \sin a \sin b$,

$$x_p(t) = \frac{\alpha}{\omega_d}\Big[\cos(\omega_j t)\,I_1 + \sin(\omega_j t)\,I_2\Big], \tag{24}$$

where

$$I_1 = \int_0^t e^{-\gamma\tau}\sin(\omega_d\tau)\cos(\omega_j\tau)\,\mathrm{d}\tau, \tag{25}$$

$$I_2 = \int_0^t e^{-\gamma\tau}\sin(\omega_d\tau)\sin(\omega_j\tau)\,\mathrm{d}\tau. \tag{26}$$

Using

$$\sin a \cos b = \tfrac{1}{2}\big[\sin(a+b) + \sin(a-b)\big], \qquad \sin a \sin b = \tfrac{1}{2}\big[\cos(a-b) - \cos(a+b)\big],$$

we obtain

$$I_1 = \frac{1}{2}\int_0^t e^{-\gamma\tau}\Big[\sin\big((\omega_d + \omega_j)\tau\big) + \sin\big((\omega_d - \omega_j)\tau\big)\Big]\mathrm{d}\tau, \tag{27}$$

$$I_2 = \frac{1}{2}\int_0^t e^{-\gamma\tau}\Big[\cos\big((\omega_d - \omega_j)\tau\big) - \cos\big((\omega_d + \omega_j)\tau\big)\Big]\mathrm{d}\tau. \tag{28}$$

Let $\lambda_+ := \omega_d + \omega_j$ and $\lambda_- := \omega_d - \omega_j$. Using

$$\int e^{-\gamma\tau}\sin(\lambda\tau)\,\mathrm{d}\tau = \frac{e^{-\gamma\tau}}{\gamma^2 + \lambda^2}\Big(-\gamma\sin(\lambda\tau) - \lambda\cos(\lambda\tau)\Big),$$

and evaluating from $0$ to $t$ gives

$$I_1 = \frac{1}{2}\sum_{\lambda\in\{\lambda_+,\lambda_-\}}\frac{1}{\gamma^2 + \lambda^2}\Big[-\gamma\big(e^{-\gamma t}\sin(\lambda t) - 0\big) - \lambda\big(e^{-\gamma t}\cos(\lambda t) - 1\big)\Big]. \tag{29}$$

Similarly, using

$$\int e^{-\gamma\tau}\cos(\lambda\tau)\,\mathrm{d}\tau = \frac{e^{-\gamma\tau}}{\gamma^2 + \lambda^2}\Big(-\gamma\cos(\lambda\tau) + \lambda\sin(\lambda\tau)\Big),$$

we obtain

$$I_2 = \frac{1}{2}\sum_{\lambda\in\{\lambda_+,\lambda_-\}}\frac{1}{\gamma^2 + \lambda^2}\Big[\big(-\gamma e^{-\gamma t}\cos(\lambda t) + \lambda e^{-\gamma t}\sin(\lambda t) + \gamma\big)\Big]. \tag{30}$$

Therefore the particular solution for Case I may be written compactly as

$$x_p(t) = \frac{\alpha}{\omega_d}\Big[\cos(\omega_j t)\,I_1 + \sin(\omega_j t)\,I_2\Big], \qquad I_1 \text{ as in equation 29, } I_2 \text{ as in equation 30.} \tag{31}$$

### A.2.2  CASE II: $\gamma = \omega$ (CRITICALLY DAMPED)

Here

$$g_{12}(\tau) = e^{-\gamma\tau}\,\tau, \qquad f(t-\tau) = \cos(\omega_j(t-\tau)), \tag{32}$$

so

$$x_p(t) = \alpha \int_0^t e^{-\gamma\tau}\,\tau\,\cos(\omega_j(t-\tau))\,\mathrm{d}\tau, \tag{33}$$

which can also be evaluated in closed form.

### A.2.3 CASE III: $\gamma > \omega$ (OVERDAMPED)

Write $\sigma = \sqrt{\gamma^2 - \omega^2}$. Then

$$g_{12}(\tau) = e^{-\gamma\tau}\, \frac{\sinh(\sigma\tau)}{\sigma}, \qquad f(t-\tau) = \cos\big(\omega_j(t-\tau)\big), \tag{34}$$

and

$$x_p(t) = \alpha \int_0^t e^{-\gamma\tau}\, \frac{\sinh(\sigma\tau)}{\sigma}\, \cos\big(\omega_j(t-\tau)\big)\, \mathrm{d}\tau, \tag{35}$$

which likewise admits a closed form.

### A.3 STEADY-STATE SOLUTION FOR THE DRIVEN, DAMPED OSCILLATOR

Consider the scalar ODE

$$\ddot{x} + 2\gamma\dot{x} + \omega_0^2 x = \alpha \sum_{j=1}^J \Big(A_j \cos(\omega_j t) + B_j \sin(\omega_j t)\Big). \tag{36}$$

We seek the steady-state particular solution $x_{p,\mathrm{ss}}(t)$. For a single forcing component $\alpha\,[A_j \cos(\omega_j t) + B_j \sin(\omega_j t)]$, assume

$$x_{pj}(t) = C_j \cos(\omega_j t) + D_j \sin(\omega_j t). \tag{37}$$

Then

$$\dot{x}_{pj}(t) = -C_j \omega_j \sin(\omega_j t) + D_j \omega_j \cos(\omega_j t),$$
$$\ddot{x}_{pj}(t) = -C_j \omega_j^2 \cos(\omega_j t) - D_j \omega_j^2 \sin(\omega_j t).$$

Substituting gives

$$\big[-C_j \omega_j^2 \cos(\omega_j t) - D_j \omega_j^2 \sin(\omega_j t)\big] + 2\gamma\big[-C_j \omega_j \sin(\omega_j t) + D_j \omega_j \cos(\omega_j t)\big] +$$
$$\omega_0^2 \big[C_j \cos(\omega_j t) + D_j \sin(\omega_j t)\big] = \alpha\big[A_j \cos(\omega_j t) + B_j \sin(\omega_j t)\big].$$

Collecting coefficients of $\cos(\omega_j t)$ and $\sin(\omega_j t)$ yields the linear system

$$\begin{bmatrix} \omega_0^2 - \omega_j^2 & 2\gamma\omega_j \\ -2\gamma\omega_j & \omega_0^2 - \omega_j^2 \end{bmatrix} \begin{bmatrix} C_j \\ D_j \end{bmatrix} = \alpha \begin{bmatrix} A_j \\ B_j \end{bmatrix}. \tag{38}$$

(One can solve for $C_j, D_j$ in closed form if desired.)

Collecting the $\cos(\omega_j t)$ terms gives

$$C_j\big(\omega_0^2 - \omega_j^2\big) + 2\gamma\omega_j D_j = \alpha A_j. \tag{39}$$

Collecting the $\sin(\omega_j t)$ terms gives

$$-D_j \omega_j^2 - 2\gamma C_j \omega_j + \omega_0^2 D_j = \alpha B_j \quad\Longrightarrow\quad D_j\big(\omega_0^2 - \omega_j^2\big) - 2\gamma\omega_j C_j = \alpha B_j. \tag{40}$$

Therefore, we have the linear system

$$\begin{bmatrix} \omega_0^2 - \omega_j^2 & 2\gamma\omega_j \\ -2\gamma\omega_j & \omega_0^2 - \omega_j^2 \end{bmatrix} \begin{bmatrix} C_j \\ D_j \end{bmatrix} = \alpha \begin{bmatrix} A_j \\ B_j \end{bmatrix}. \tag{41}$$

Its determinant is

$$\det = \big(\omega_0^2 - \omega_j^2\big)^2 + \big(2\gamma\omega_j\big)^2. \tag{42}$$

Using Cramer's rule,

$$C_j = \alpha\, \frac{A_j\big(\omega_0^2 - \omega_j^2\big) - B_j\big(2\gamma\omega_j\big)}{\big(\omega_0^2 - \omega_j^2\big)^2 + \big(2\gamma\omega_j\big)^2}, \tag{43}$$

$$D_j = \alpha\, \frac{B_j\big(\omega_0^2 - \omega_j^2\big) + A_j\big(2\gamma\omega_j\big)}{\big(\omega_0^2 - \omega_j^2\big)^2 + \big(2\gamma\omega_j\big)^2}. \tag{44}$$

Hence

$$x_{p,j}(t) = \frac{\alpha}{\left(\omega_0^2 - \omega_j^2\right)^2 + \left(2\gamma\omega_j\right)^2} \Big( \left[A_j(\omega_0^2 - \omega_j^2) - B_j(2\gamma\omega_j)\right] \cos(\omega_j t) +$$

$$\left[B_j(\omega_0^2 - \omega_j^2) + A_j(2\gamma\omega_j)\right] \sin(\omega_j t) \Big). \quad (45)$$

By superposition, the complete steady–state solution is

$$x_{p,\mathrm{ss}}(t) = \sum_{j=1}^{J} x_{p,j}(t). \quad (46)$$

Equivalently, written out explicitly,

$$x_{p,\mathrm{ss}}(t) = \alpha \sum_{j=1}^{J} \frac{1}{\left(\omega_0^2 - \omega_j^2\right)^2 + \left(2\gamma\omega_j\right)^2} \Big( \left[A_j(\omega_0^2 - \omega_j^2) - B_j(2\gamma\omega_j)\right] \cos(\omega_j t) +$$

$$\left[B_j(\omega_0^2 - \omega_j^2) + A_j(2\gamma\omega_j)\right] \sin(\omega_j t) \Big). \quad (47)$$

A.4    QUERY FUNCTION

For the query, we expand the interpolation function in the oscillator basis up to a suitable number of modes and obtain the coefficients $A_k$, $B_k$ by a least-squares fit. This circumvents the absence of a closed-form solution for the integral of the original cubic spline.

$$q(t) = \sum_{k=1}^{N} \Big( A_k \cos(\omega_k t) + B_k \sin(\omega_k t) \Big). \quad (48)$$

A.5    ATTENTION INTEGRAL

We compute the averaged attention coefficient

$$\alpha_i(t) = \frac{1}{\Delta} \int_{t_i}^{t} \langle q(\tau), k_i(\tau) \rangle \, d\tau, \qquad \Delta := t - t_i > 0,$$

when the (vector) key coordinates obey a *driven* damped oscillator, anchored at $t_i$ with zero particular state. The total key is $k_i = k_{i,\mathrm{hom}} + k_{i,\mathrm{part}} + c_i$, where the homogeneous part $k_{i,\mathrm{hom}}$ was derived in section A.1, and here we add the driven part $k_{i,\mathrm{part}}$. All expressions act coordinate-wise and we keep vector inner products to avoid clutter.

A.5.1    MATHEMATICAL FRAMEWORK

**Query expansion and rotation to anchor.**    We fix $i$ and expand the $d_k$-vector query:

$$q(\tau) = \sum_{j=1}^{J} \Big( A_j \cos(\omega_j \tau) + B_j \sin(\omega_j \tau) \Big), \qquad A_j, B_j \in \mathbb{R}^{d_k}, \ \omega_j > 0. \quad (49)$$

With $s := \tau - t_i \in [0, \Delta]$, the rotated coefficients

$$\tilde{A}_j := A_j \cos(\omega_j t_i) + B_j \sin(\omega_j t_i), \qquad \tilde{B}_j := -A_j \sin(\omega_j t_i) + B_j \cos(\omega_j t_i), \quad (50)$$

give the anchor-shifted query

$$q(t_i + s) = \sum_{j=1}^{J} \Big( \tilde{A}_j \cos(\omega_j s) + \tilde{B}_j \sin(\omega_j s) \Big). \quad (51)$$

**Exponential-trigonometric kernels.** For $\gamma \geq 0$, $\lambda \in \mathbb{R}$, $\Delta > 0$, define

$$C_\gamma(\Delta, \lambda) := \int_0^\Delta e^{-\gamma s} \cos(\lambda s) \, ds = \frac{e^{-\gamma\Delta}\big(-\gamma \cos(\lambda\Delta) + \lambda \sin(\lambda\Delta)\big) + \gamma}{\gamma^2 + \lambda^2}, \tag{52}$$

$$S_\gamma(\Delta, \lambda) := \int_0^\Delta e^{-\gamma s} \sin(\lambda s) \, ds = \frac{e^{-\gamma\Delta}\big(-\gamma \sin(\lambda\Delta) - \lambda \cos(\lambda\Delta)\big) + \lambda}{\gamma^2 + \lambda^2}. \tag{53}$$

Their $\lambda \to 0$ limits are $C_\gamma(\Delta, 0) = (1 - e^{-\gamma\Delta})/\gamma$ (or $\Delta$ if $\gamma = 0$) and $S_\gamma(\Delta, 0) = 0$.

For products of trigonometric functions with exponential damping, we use

$$I_{cc}(\Delta; \gamma, \lambda_1, \lambda_2) := \int_0^\Delta e^{-\gamma s} \cos(\lambda_1 s) \cos(\lambda_2 s) \, ds = \tfrac{1}{2}[C_\gamma(\Delta, \lambda_1 - \lambda_2) + C_\gamma(\Delta, \lambda_1 + \lambda_2)], \tag{54}$$

$$I_{ss}(\Delta; \gamma, \lambda_1, \lambda_2) := \int_0^\Delta e^{-\gamma s} \sin(\lambda_1 s) \sin(\lambda_2 s) \, ds = \tfrac{1}{2}[C_\gamma(\Delta, \lambda_1 - \lambda_2) - C_\gamma(\Delta, \lambda_1 + \lambda_2)], \tag{55}$$

$$I_{sc}(\Delta; \gamma, \lambda_1, \lambda_2) := \int_0^\Delta e^{-\gamma s} \sin(\lambda_1 s) \cos(\lambda_2 s) \, ds = \tfrac{1}{2}[S_\gamma(\Delta, \lambda_1 + \lambda_2) + S_\gamma(\Delta, \lambda_1 - \lambda_2)], \tag{56}$$

$$I_{cs}(\Delta; \gamma, \lambda_1, \lambda_2) := \int_0^\Delta e^{-\gamma s} \cos(\lambda_1 s) \sin(\lambda_2 s) \, ds = \tfrac{1}{2}[S_\gamma(\Delta, \lambda_1 + \lambda_2) - S_\gamma(\Delta, \lambda_1 - \lambda_2)]. \tag{57}$$

For undamped integrals (when $\gamma = 0$), we recover the standard trigonometric identities. For $a, b > 0$ and $a \neq b$:

$$I_{cc}(\Delta; 0, a, b) = \frac{\sin\big((a - b)\Delta\big)}{2(a - b)} + \frac{\sin\big((a + b)\Delta\big)}{2(a + b)}, \tag{58}$$

$$I_{ss}(\Delta; 0, a, b) = \frac{\sin\big((a - b)\Delta\big)}{2(a - b)} - \frac{\sin\big((a + b)\Delta\big)}{2(a + b)}, \tag{59}$$

$$I_{sc}(\Delta; 0, a, b) = \frac{1 - \cos\big((a + b)\Delta\big)}{2(a + b)} + \frac{1 - \cos\big((a - b)\Delta\big)}{2(a - b)}, \tag{60}$$

$$I_{cs}(\Delta; 0, a, b) = \frac{1 - \cos\big((a + b)\Delta\big)}{2(a + b)} + \frac{1 - \cos\big((b - a)\Delta\big)}{2(b - a)}. \tag{61}$$

Note that $I_{cs}(\Delta; 0, a, b) = I_{sc}(\Delta; 0, b, a)$ (frequencies swapped). For $a = b$, we use the continuous limits: $I_{cc}(\Delta; 0, a, a) = \frac{\Delta}{2} + \frac{\sin(2a\Delta)}{4a}$, $I_{ss}(\Delta; 0, a, a) = \frac{\Delta}{2} - \frac{\sin(2a\Delta)}{4a}$, and $I_{sc}(\Delta; 0, a, a) = I_{cs}(\Delta; 0, a, a) = \frac{1 - \cos(2a\Delta)}{4a}$.

### A.5.2 Driven Oscillator: Steady-State Solution

Consider the vector ODE

$$\ddot{x} + 2\gamma\dot{x} + \omega_0^2 x = f(t), \qquad t \geq t_i, \tag{62}$$

with vector forcing expanded in harmonics

$$f_i(t) = \sum_{m=1}^{M_f} \big(P_{i,m} \cos(\varpi_m t) + Q_{i,m} \sin(\varpi_m t)\big), \qquad P_{i,m}, Q_{i,m} \in \mathbb{R}^{d_k}, \ \varpi_m > 0. \tag{63}$$

For a single frequency component with coefficients $(P, Q, \varpi)$, the steady-state particular solution has the form $x_{ss}(t) = C \cos(\varpi t) + D \sin(\varpi t)$. Substituting into equation 62 and equating coefficients gives the linear system

$$\begin{bmatrix} \omega_0^2 - \varpi^2 & 2\gamma\varpi \\ -2\gamma\varpi & \omega_0^2 - \varpi^2 \end{bmatrix} \begin{pmatrix} C \\ D \end{pmatrix} = \begin{pmatrix} P \\ Q \end{pmatrix}.$$

With $\Delta_\varpi := (\omega_0^2 - \varpi^2)^2 + (2\gamma\varpi)^2$, Cramer's rule yields

$$C = \frac{P(\omega_0^2 - \varpi^2) - Q(2\gamma\varpi)}{\Delta_\varpi}, \qquad D = \frac{Q(\omega_0^2 - \varpi^2) + P(2\gamma\varpi)}{\Delta_\varpi}. \tag{64}$$

### A.5.3 UNDERDAMPED DRIVEN KEY ($\gamma < \omega_0$): FULL SOLUTION AND ATTENTION

Let $\omega_d := \sqrt{\omega_0^2 - \gamma^2}$ be the damped frequency. The complete steady-state solution is

$$x_{\mathrm{ss},i}(t) = \sum_{m=1}^{M_f} \Big( C_{i,m} \cos(\varpi_m t) + D_{i,m} \sin(\varpi_m t) \Big), \tag{65}$$

where each $(C_{i,m}, D_{i,m})$ is given by equation 64 applied to $(P_{i,m}, Q_{i,m}, \varpi_m)$.

**Transient for zero initial particular state.** To enforce clean anchoring, we require

$$x_{\mathrm{part}}(t_i) = 0, \qquad \dot{x}_{\mathrm{part}}(t_i) = 0.$$

The transient solution has the form $x_{\mathrm{tr}}(t_i + s) = e^{-\gamma s} \big( E_i \cos(\omega_d s) + F_i \sin(\omega_d s) \big)$ where

$$E_i = -x_{\mathrm{ss},i}(t_i), \tag{66}$$

$$F_i = \frac{-\gamma\, x_{\mathrm{ss},i}(t_i) \;+\; \sum_{m=1}^{M_f} C_{i,m}\varpi_m \sin(\varpi_m t_i) \;-\; \sum_{m=1}^{M_f} D_{i,m}\varpi_m \cos(\varpi_m t_i)}{\omega_d}. \tag{67}$$

**Driven key in anchor-shifted form.** Let $s = t - t_i$. The steady-state part becomes

$$x_{\mathrm{ss},i}(t_i + s) = \sum_{m=1}^{M_f} \Big( \widehat{C}_{i,m} \cos(\varpi_m s) + \widehat{D}_{i,m} \sin(\varpi_m s) \Big), \tag{68}$$

where the rotated coefficients are

$$\widehat{C}_{i,m} := C_{i,m} \cos(\varpi_m t_i) + D_{i,m} \sin(\varpi_m t_i), \quad \widehat{D}_{i,m} := -C_{i,m} \sin(\varpi_m t_i) + D_{i,m} \cos(\varpi_m t_i). \tag{69}$$

The complete particular key is

$$k_{i,\mathrm{part}}(t_i + s) = x_{\mathrm{ss},i}(t_i + s) + e^{-\gamma s} \big( E_i \cos(\omega_d s) + F_i \sin(\omega_d s) \big). \tag{70}$$

**Averaged attention: decomposition.** Using equation 51, equation 68, and equation 70 with $s \in [0, \Delta]$:

$$\int_{t_i}^{t} \langle q(\tau), k_{i,\mathrm{part}}(\tau) \rangle \, d\tau = \underbrace{\int_{0}^{\Delta} \langle q(t_i + s), x_{\mathrm{ss},i}(t_i + s) \rangle \, ds}_{\mathcal{I}_i^{(\mathrm{ss})}} +$$

$$\underbrace{\int_{0}^{\Delta} e^{-\gamma s} \langle q(t_i + s), E_i \cos(\omega_d s) + F_i \sin(\omega_d s) \rangle \, ds}_{\mathcal{I}_i^{(\mathrm{tr})}}. \tag{71}$$

**Steady-state contribution $\mathcal{I}_i^{(\mathrm{ss})}$.** Expanding the query and steady-state solutions:

$$\mathcal{I}_i^{(\mathrm{ss})} = \sum_{j=1}^{J} \sum_{m=1}^{M_f} \int_{0}^{\Delta} \langle \tilde{A}_j \cos(\omega_j s) + \tilde{B}_j \sin(\omega_j s), \widehat{C}_{i,m} \cos(\varpi_m s) + \widehat{D}_{i,m} \sin(\varpi_m s) \rangle \, ds.$$

Using the undamped kernels equation 58–equation 61:

$$\mathcal{I}_i^{(\mathrm{ss})} = \sum_{j=1}^{J} \sum_{m=1}^{M_f} \Big[ \langle \tilde{A}_j, \widehat{C}_{i,m} \rangle \, I_{cc}(\Delta; 0, \omega_j, \varpi_m) + \langle \tilde{A}_j, \widehat{D}_{i,m} \rangle \, I_{cs}(\Delta; 0, \omega_j, \varpi_m)$$

$$+ \langle \tilde{B}_j, \widehat{C}_{i,m} \rangle \, I_{sc}(\Delta; 0, \omega_j, \varpi_m) + \langle \tilde{B}_j, \widehat{D}_{i,m} \rangle \, I_{ss}(\Delta; 0, \omega_j, \varpi_m) \Big]. \tag{72}$$

**Transient contribution $\mathcal{I}_i^{(\mathrm{tr})}$.** Using the damped kernels equation 54–equation 57 with $\lambda_1 \in \{\omega_d\}$ and $\lambda_2 \in \{\omega_j\}$:

$$
\mathcal{I}_i^{(\mathrm{tr})} = \sum_{j=1}^{J} \Big[ \langle E_i, \tilde{A}_j \rangle \, I_{cc}(\Delta; \gamma, \omega_d, \omega_j) + \langle E_i, \tilde{B}_j \rangle \, I_{cs}(\Delta; \gamma, \omega_d, \omega_j)
$$
$$
+ \langle F_i, \tilde{A}_j \rangle \, I_{sc}(\Delta; \gamma, \omega_d, \omega_j) + \langle F_i, \tilde{B}_j \rangle \, I_{ss}(\Delta; \gamma, \omega_d, \omega_j) \Big]. \tag{73}
$$

**Final result.** The driven contribution to the averaged attention is

$$
\boxed{\alpha_i^{(\mathrm{driven})}(t) = \frac{1}{\Delta}\Big(\mathcal{I}_i^{(\mathrm{ss})} + \mathcal{I}_i^{(\mathrm{tr})}\Big),} \tag{74}
$$

where $\mathcal{I}_i^{(\mathrm{ss})}$ and $\mathcal{I}_i^{(\mathrm{tr})}$ are given by equation 72 and equation 73, respectively.

The complete logit is

$$
\alpha_i(t) = \alpha_i^{(\mathrm{hom})}(t) + \langle \bar{q}_{[t_i,t]}, c_i \rangle + \alpha_i^{(\mathrm{driven})}(t), \tag{75}
$$

where $\alpha_i^{(\mathrm{hom})}(t)$ is the homogeneous contribution derived in Cases I–III above, and $\bar{q}_{[t_i,t]} = \frac{1}{\Delta} \int_{t_i}^{t} q(\tau)\, d\tau$ is the average query over the interval.

### A.5.4 CRITICAL AND OVERDAMPED DRIVEN KEYS

The derivation follows the same structure with modified transient forms:

**Critical damping $(\gamma = \omega_0)$.** The transient basis is $x_{\mathrm{tr}}(t_i + s) = e^{-\gamma s}(E + Fs)$ with

$$
E = -x_{\mathrm{ss}}(t_i), \qquad F = \gamma E - \dot{x}_{\mathrm{ss}}(t_i).
$$

The transient contribution becomes

$$
\mathcal{I}_i^{(\mathrm{tr})} = \sum_{j=1}^{J} \Big[ \langle E, \tilde{A}_j \rangle \, C_\gamma(\Delta, \omega_j) + \langle E, \tilde{B}_j \rangle \, S_\gamma(\Delta, \omega_j)
$$
$$
+ \langle F, \tilde{A}_j \rangle \int_0^{\Delta} s\, e^{-\gamma s} \cos(\omega_j s)\, ds + \langle F, \tilde{B}_j \rangle \int_0^{\Delta} s\, e^{-\gamma s} \sin(\omega_j s)\, ds \Big],
$$
$$
\tag{76}
$$

where the integrals involving $s$ can be evaluated by integration by parts.

**Overdamped $(\gamma > \omega_0)$.** Let $\sigma := \sqrt{\gamma^2 - \omega_0^2} > 0$. The transient basis is

$$
x_{\mathrm{tr}}(t_i + s) = U\, e^{-(\gamma - \sigma)s} + V\, e^{-(\gamma + \sigma)s},
$$

where

$$
U = \frac{-(\gamma + \sigma)x_{\mathrm{ss}}(t_i) + \dot{x}_{\mathrm{ss}}(t_i)}{2\sigma}, \quad V = \frac{-(\gamma - \sigma)x_{\mathrm{ss}}(t_i) - \dot{x}_{\mathrm{ss}}(t_i)}{2\sigma}.
$$

The transient contribution is

$$
\mathcal{I}_i^{(\mathrm{tr})} = \sum_{j=1}^{J} \Big[ \langle U, \tilde{A}_j \rangle \, C_{\gamma-\sigma}(\Delta, \omega_j) + \langle U, \tilde{B}_j \rangle \, S_{\gamma-\sigma}(\Delta, \omega_j)
$$
$$
+ \langle V, \tilde{A}_j \rangle \, C_{\gamma+\sigma}(\Delta, \omega_j) + \langle V, \tilde{B}_j \rangle \, S_{\gamma+\sigma}(\Delta, \omega_j) \Big]. \tag{77}
$$

In both cases, the steady-state contribution $\mathcal{I}_i^{(\mathrm{ss})}$ remains as in equation 72, and the final attention coefficient is given by equation 74 with the appropriate transient contribution.

## B   Harmonic Approximation Theorem

Fix a compact interval $[a, b] \subset \mathbb{R}$, feature dimension $d_k \geq 1$, and observation times $t_1 < \cdots < t_N$ in $[a, b]$. Let $q \colon [a, b] \to \mathbb{R}^{d_k}$ be continuous. For each observation index $i \in \{1, \ldots, N\}$, let $k_i \colon [t_i, b] \to \mathbb{R}^{d_k}$ be a continuous *key trajectory*. Throughout, $\|\cdot\|_2$ denotes the Euclidean vector norm, $\|\cdot\|$ denotes the induced operator norm, and $\|q\|_\infty := \sup_{t \in [a,b]} \|q(t)\|_2$.

**Definition 1** (Averaged inner-product logit). For $t \geq t_i$ define

$$
\alpha_i(t) := \begin{cases} \dfrac{1}{t - t_i} \displaystyle\int_{t_i}^t \langle q(\tau),\, k_i(\tau) \rangle \, d\tau, & t > t_i, \\ \langle q(t_i),\, k_i(t_i) \rangle, & t = t_i. \end{cases} \tag{78}
$$

**Definition 2** (Masked pre-softmax CT attention and softmax). At an evaluation time $t$, only keys with $t_i \leq t$ contribute. The pre-softmax CT-attention matrix (rows indexed by $t_j$, columns by $i$) is

$$
\mathsf{Attn}^{\mathrm{CT}}(Q, K) \;=\; \big[\, \alpha_i(t_j) \,\big]_{\substack{j=1,\ldots,N \\ i=1,\ i \leq j}}^{N} \in (\mathbb{R} \cup \{-\infty\})^{N \times N},
$$

where entries with $j < i$ are undefined by equation 78 and are masked (set to $-\infty$ prior to softmax). The softmax attention vector at time $t$ is

$$
w_i(t) := \frac{\exp\big(\alpha_i(t)/\sqrt{d_k}\big)}{\sum_{j \,:\, t_j \leq t} \exp\big(\alpha_j(t)/\sqrt{d_k}\big)} \quad \text{(sum over valid } j\text{)}. \tag{79}
$$

We use a single shared bank of harmonic modes; only the *initial conditions* differ across keys.

**Definition 3** (Fixed oscillator bank and readout). Let $L := b - a$. Include the *zero mode* and fix the grid

$$
\omega_0 := 0, \qquad \omega_n := \frac{n\pi}{L} \quad (n \geq 1).
$$

Choose $M \in \mathbb{N}$ and use modes $n = 0, 1, \ldots, M$. For (possibly damped) per-mode parameters $\gamma_n \geq 0$, define $2 \times 2$ blocks

$$
A_n = \begin{bmatrix} 0 & 1 \\ -\omega_n^2 & -2\gamma_n \end{bmatrix},
$$

and let $A = \mathrm{diag}(A_0, \ldots, A_M) \in \mathbb{R}^{2(M+1) \times 2(M+1)}$. For feature dimension $d_k$, take $d_k$ independent copies (one per coordinate) so the full state is $z \in \mathbb{R}^{2(M+1)d_k}$ and the dynamics $\dot{z}(t) = Az(t)$ hold coordinate-wise.

For key index $i$, the system is anchored at $t_i$ with initial state $z_{i,0}$ via $z_i(t_i) = z_{i,0}$ and $z_i(t) = e^{A(t-t_i)} z_{i,0}$. Denote by $x_{\ell,n}(t)$ the *position* coordinate of the $(\ell, n)$-oscillator. The readout *sums positions across modes for each feature coordinate*:

$$
k_{i,\ell}(t) \;=\; \sum_{n=0}^{M} x_{\ell,n}(t), \qquad \ell = 1, \ldots, d_k, \tag{80}
$$

i.e., $k_i(t) = C z_i(t)$ with $C \in \mathbb{R}^{d_k \times 2(M+1)d_k}$ that puts ones on position entries and zeros elsewhere. In the main theorem we set $\gamma_n = 0$; a perturbation lemma then allows $\gamma_n > 0$.

*Remark* 1. For $\omega_0 = 0$, $x_{\ell,0}(t) = A_{\ell,0} + B_{\ell,0}(t - t_i)$. We will *always* choose $B_{\ell,0} = 0$ so the zero mode supplies constants without linear drift.

**Definition 4** (Fejér kernel and means). For $N \in \mathbb{N}$, the Fejér kernel $K_N \colon \mathbb{R} \to [0, \infty)$ is

$$
K_N(\theta) = \frac{1}{N+1} \left( \frac{\sin\big((N+1)\theta/2\big)}{\sin(\theta/2)} \right)^2 = \sum_{k=-N}^{N} \left( 1 - \frac{|k|}{N+1} \right) e^{ik\theta}.
$$

Given a $2\pi$-periodic, continuous $F : \mathbb{R} \to \mathbb{R}$, its Fejér mean is

$$
\sigma_N[F](s) = \frac{1}{2\pi} \int_{-\pi}^{\pi} F(s - \theta) \, K_N(\theta) \, d\theta.
$$

**Lemma 1** (Basic properties of $K_N$). *For every $N \in \mathbb{N}$:*

1. $K_N(\theta) \geq 0$ *for all $\theta \in \mathbb{R}$.*
2. $\dfrac{1}{2\pi} \displaystyle\int_{-\pi}^{\pi} K_N(\theta)\, d\theta = 1.$
3. *For any fixed $\delta \in (0, \pi]$,*

$$\frac{1}{2\pi} \int_{|\theta| \geq \delta} K_N(\theta)\, d\theta \;\leq\; \frac{1}{(N+1)\, \sin^2(\delta/2)}.$$

*Proof.* (1) Using the geometric sum,

$$\sum_{j=0}^{N} e^{ij\theta} = \frac{1 - e^{i(N+1)\theta}}{1 - e^{i\theta}} = e^{iN\theta/2}\, \frac{\sin\big((N+1)\theta/2\big)}{\sin(\theta/2)}.$$

Hence

$$K_N(\theta) \;=\; \frac{1}{N+1} \Big| \sum_{j=0}^{N} e^{ij\theta} \Big|^2 \;\geq\; 0.$$

(2) Integrating the Fourier series in Definition 4 term-wise over $[-\pi, \pi]$ annihilates all nonzero frequencies; the constant term is 1, so $\frac{1}{2\pi}\int_{-\pi}^{\pi} K_N(\theta)\, d\theta = 1$.

(3) For $|\theta| \geq \delta$ we have $\sin(\theta/2) \geq \sin(\delta/2) > 0$, whence

$$K_N(\theta) \;=\; \frac{1}{N+1}\, \frac{\sin^2\big((N+1)\theta/2\big)}{\sin^2(\theta/2)} \;\leq\; \frac{1}{(N+1)\, \sin^2(\delta/2)}.$$

Integrate this bound over a set of measure at most $2\pi$ to get the claim. $\qquad\square$

**Proposition 1** (Uniform convergence of Fejér means). *If $F \in C(\mathbb{T})$ (with $\mathbb{T} := \mathbb{R}/2\pi\mathbb{Z}$), then $\sigma_N[F] \to F$ uniformly on $\mathbb{R}$ as $N \to \infty$.*

*Proof.* Fix $\varepsilon > 0$. By uniform continuity on the circle, choose $\delta \in (0, \pi]$ with $|F(s) - F(s - \theta)| \leq \varepsilon/3$ when $|\theta| < \delta$. Then for any $s$,

$$|\sigma_N[F](s) - F(s)| \leq \frac{1}{2\pi} \int_{|\theta| < \delta} |F(s-\theta) - F(s)| K_N(\theta)\, d\theta + \frac{1}{2\pi} \int_{|\theta| \geq \delta} 2\|F\|_\infty K_N(\theta)\, d\theta$$

$$\leq \frac{\varepsilon}{3} \cdot 1 + \frac{2\|F\|_\infty}{(N+1)\sin^2(\delta/2)} \quad \text{by Lemma 1.}$$

Choose $N$ large so the second term is $< 2\varepsilon/3$; then $|\sigma_N[F] - F| < \varepsilon$ uniformly. $\qquad\square$

**Lemma 2** (Vector Fejér density on the half-range grid). *Let $f \in C([a,b]; \mathbb{R}^{d_k})$. For any $\varepsilon > 0$ there exist $N \in \mathbb{N}$ and coefficients $c_0 \in \mathbb{R}^{d_k}$, $c_n, s_n \in \mathbb{R}^{d_k}$ ($1 \leq n \leq N$) such that*

$$P_N(t) \;=\; c_0 + \sum_{n=1}^{N} \Big( c_n \cos \omega_n (t-a) + s_n \sin \omega_n (t-a) \Big) \tag{81}$$

*satisfies $\sup_{t \in [a,b]} \|f(t) - P_N(t)\|_2 < \varepsilon$.*

*Proof.* For each coordinate $f^\ell$ define the *even* $2L$-periodic extension

$$F^\ell(s) = \begin{cases} f^\ell(a+s), & s \in [0, L], \\ f^\ell(a-s), & s \in [-L, 0), \end{cases} \quad \text{extended } 2L\text{-periodically.}$$

Each $F^\ell \in C(\mathbb{T}_{2L})$ (where $\mathbb{T}_{2L} := \mathbb{R}/(2L\mathbb{Z})$). Applying Fejér on the circle of length $2L$ (equivalently, on $[0, 2\pi]$ after the affine map $s \mapsto 2\pi s/(2L)$) and restricting to $s \in [0, L]$ yields $\sigma_N[F^\ell](s) \to f^\ell(a+s)$ uniformly. Writing $\sigma_N[F^\ell](a+s)$ in the form $c_0^\ell + \sum_{n=1}^{N} c_n^\ell \cos(\omega_n s)$ (evenness gives only cosines; allowing $s_n^\ell = 0$ is harmless), choose a common $N$ so that for all $\ell$, $\sup_{t \in [a,b]} |f^\ell(t) - \sigma_N[F^\ell](t-a)| < \varepsilon/\sqrt{d_k}$. Assemble $c_0, c_n, s_n$ coordinate-wise to obtain equation 81 with the stated bound. $\qquad\square$

**Lemma 3** (Phase shift from $(t-a)$ to $(t-t_i)$). *For $\phi_n := \omega_n(t_i - a)$ and any $c_n, s_n \in \mathbb{R}^{d_k}$, there are unique $\tilde{c}_n, \tilde{s}_n \in \mathbb{R}^{d_k}$ such that*

$$c_n \cos \omega_n(t-a) + s_n \sin \omega_n(t-a) = \tilde{c}_n \cos \omega_n(t-t_i) + \tilde{s}_n \sin \omega_n(t-t_i),$$

*with*

$$\begin{pmatrix} \tilde{c}_n \\ \tilde{s}_n \end{pmatrix} = \begin{bmatrix} \cos \phi_n & \sin \phi_n \\ -\sin \phi_n & \cos \phi_n \end{bmatrix} \begin{pmatrix} c_n \\ s_n \end{pmatrix}.$$

**Lemma 4** (Exact realizability of vector trigonometric polynomials, $\gamma_n = 0$). *Fix $M \geq N$ and the undamped bank ($\gamma_n = 0$). For a vector trigonometric polynomial*

$$P_N(t) = c_0 + \sum_{n=1}^{N} \left( \tilde{c}_n \cos \omega_n(t - t_i) + \tilde{s}_n \sin \omega_n(t - t_i) \right),$$

*there exist initial conditions $z_{i,0}$ such that the readout equation 80 satisfies $k_i(t) \equiv P_N(t)$ for all $t \geq t_i$.*

*Proof.* For the $(\ell, n)$ oscillator ($n \geq 1$) with $\ddot{x}_{\ell,n} + \omega_n^2 x_{\ell,n} = 0$, the solution is $x_{\ell,n}(t) = A_{\ell,n} \cos \omega_n(t - t_i) + \frac{B_{\ell,n}}{\omega_n} \sin \omega_n(t - t_i)$. Choose $A_{\ell,n} = (\tilde{c}_n)^\ell$ and $B_{\ell,n} = \omega_n(\tilde{s}_n)^\ell$. For $n = 0$, set $x_{\ell,0}(t) \equiv (c_0)^\ell$ (initial velocity zero). Summing positions across $n$ gives $k_{i,\ell}(t) = P_N^\ell(t)$. $\square$

**Lemma 5** (Matrix-exponential perturbation bound). *Let $A_0$ be the undamped bank matrix and $A_\gamma = A_0 + \Delta$ with $\Delta = \mathrm{diag}(\Delta_0, \ldots, \Delta_M)$, $\Delta_n = \begin{pmatrix} 0 & 0 \\ 0 & -2\gamma_n \end{pmatrix}$. Fix $T := b - a$ and a bound $\bar{\gamma} \geq 0$. If $0 \leq \gamma_n \leq \bar{\gamma}$ for all $n$, then there exists a constant $K = K(T, \{\omega_n\}, C, \bar{\gamma})$ such that, for all $t \in [0, T]$,*

$$\left\| C\left( e^{A_\gamma t} - e^{A_0 t} \right) \right\| \leq K \max_{0 \leq n \leq M} \gamma_n.$$

*Proof.* By Duhamel/variation-of-constants, $e^{A_\gamma t} - e^{A_0 t} = \int_0^t e^{A_\gamma(t-s)} \Delta \, e^{A_0 s} \, ds$. Hence

$$\left\| C(e^{A_\gamma t} - e^{A_0 t}) \right\| \leq \|C\| \, \|\Delta\| \int_0^t \|e^{A_\gamma(t-s)}\| \, \|e^{A_0 s}\| \, ds.$$

Define

$$M_{\bar{\gamma}} := \sup_{\substack{0 \leq \gamma_n \leq \bar{\gamma} \\ u \in [0,T]}} \left\| e^{A_\gamma u} \right\| \quad \text{and} \quad M_0 := \sup_{u \in [0,T]} \left\| e^{A_0 u} \right\|.$$

The map $(\gamma, u) \mapsto e^{A_\gamma u}$ is continuous, and the set $\{\gamma : 0 \leq \gamma_n \leq \bar{\gamma}\} \times [0, T]$ is compact, so $M_{\bar{\gamma}} < \infty$. Therefore,

$$\left\| C(e^{A_\gamma t} - e^{A_0 t}) \right\| \leq \|C\| \, \|\Delta\| \, M_{\bar{\gamma}} M_0 \, t \leq 2\|C\| \, M_{\bar{\gamma}} M_0 \, T \, \max_n \gamma_n.$$

Taking $K := 2\|C\| \, M_{\bar{\gamma}} M_0 \, T$ yields the claim. $\square$

*Remark* 2. Thus, after constructing exact undamped realizations via Lemma 4, turning on small damping changes the readout by at most $O(\max \gamma_n)$ uniformly on $[t_i, b]$. This addresses both amplitude decay and the frequency shift $\sqrt{\omega_n^2 - \gamma_n^2}$.

**Theorem 2.** *Let $q \in C([a, b]; \mathbb{R}^{d_k})$ and continuous keys $\{k_i\}_{i=1}^N$ with $k_i : [t_i, b] \to \mathbb{R}^{d_k}$. For any $\varepsilon > 0$ there exists an integer $M$ (depending on $\varepsilon$ and the keys) and a single shared oscillator bank on the fixed grid $\{\omega_n\}_{n=0}^M$ with $\gamma_n = 0$ such that one can choose initial states $\{z_{i,0}\}_{i=1}^N$ with the property*

$$\sup_{t \in [t_i, b]} \|k_i(t) - \tilde{k}_i(t)\|_2 < \varepsilon \quad \text{for all } i,$$

*where $\tilde{k}_i(t) := C e^{A(t - t_i)} z_{i,0}$ is the bank-generated key. Consequently, for all $j \geq i$,*

$$\left| \alpha_i(t_j; q, k_i) - \alpha_i(t_j; q, \tilde{k}_i) \right| \leq \|q\|_\infty \varepsilon, \qquad \|w(t_j) - \tilde{w}(t_j)\|_1 \leq \frac{\|q\|_\infty}{\sqrt{d_k}} \varepsilon.$$

*Proof.* Fix $\varepsilon > 0$. For each $i$, extend $k_i$ continuously from $[t_i, b]$ to $[a, b]$ (e.g., set $k_i(t) = k_i(t_i)$ for $t \in [a, t_i]$). Apply Lemma 2 to this extension to obtain a vector trigonometric polynomial

$$P_i(t) = c_{i,0} + \sum_{n=1}^{N_i} \big( c_{i,n} \cos \omega_n(t - a) + s_{i,n} \sin \omega_n(t - a) \big)$$

with $\sup_{t \in [a,b]} \|k_i(t) - P_i(t)\|_2 < \varepsilon/2$. Use Lemma 3 to rewrite $P_i$ as

$$P_i(t) = c_{i,0} + \sum_{n=1}^{N_i} \big( \tilde{c}_{i,n} \cos \omega_n(t - t_i) + \tilde{s}_{i,n} \sin \omega_n(t - t_i) \big).$$

Let $N := \max_i N_i$ and take $M \geq N$. By Lemma 4 (with $\gamma_n = 0$), choose $z_{i,0}$ so that the shared bank realizes $P_i$ *exactly*: $\tilde{k}_i(t) \equiv P_i(t)$ on $[t_i, b]$. Therefore $\sup_{t \in [t_i, b]} \|k_i(t) - \tilde{k}_i(t)\|_2 < \varepsilon/2 < \varepsilon$. For $t > t_i$,

$$|\alpha_i(t) - \tilde{\alpha}_i(t)| \leq \frac{1}{t - t_i} \int_{t_i}^{t} \|q(\tau)\|_2 \, \|k_i(\tau) - \tilde{k}_i(\tau)\|_2 \, d\tau \leq \|q\|_\infty \, \varepsilon.$$

At $t = t_i$ the bound $\big| \big\langle q(t_i), k_i(t_i) - \tilde{k}_i(t_i) \big\rangle \big| \leq \|q\|_\infty \, \varepsilon$ is immediate. Applying the softmax Lipschitz Lemma 6 to the logits scaled by $1/\sqrt{d_k}$ yields the stated $\ell_1$ bound. $\qquad\square$

**Corollary 2.** *Under the hypotheses of Theorem 2, fix $\varepsilon > 0$ and construct the undamped realization above. Then there exists $\bar{\gamma} > 0$ such that, for any damped bank with $0 \leq \gamma_n \leq \bar{\gamma}$, one can reuse the same initial states $\{z_{i,0}\}$ and obtain*

$$\sup_{t \in [t_i, b]} \|k_i(t) - \tilde{k}_i^{(\gamma)}(t)\|_2 < \varepsilon, \qquad \|w^{(\gamma)}(t_j) - w(t_j)\|_1 \leq \frac{\|q\|_\infty}{\sqrt{d_k}} \, \varepsilon,$$

*where the superscript $(\gamma)$ denotes readouts from the damped bank. In particular, a small amount of damping does not affect universality.*

*Proof.* By Lemma 5, for $T = b - a$ we have $\sup_{t \in [0,T]} \|C(e^{A_\gamma t} - e^{A_0 t})\| \leq K \max \gamma_n$, hence for each $i$

$$\sup_{t \in [t_i, b]} \|\tilde{k}_i^{(\gamma)}(t) - \tilde{k}_i(t)\|_2 \leq \Big( \sup_{u \in [0,T]} \|C(e^{A_\gamma u} - e^{A_0 u})\| \Big) \|z_{i,0}\| \leq K \max \gamma_n \|z_{i,0}\|.$$

Let $Z_* := \max_i \|z_{i,0}\|$. Choose $\bar{\gamma} > 0$ so that $K \bar{\gamma} Z_* \leq \varepsilon/2$. (Since the family $\{A_\gamma : 0 \leq \gamma_n \leq \bar{\gamma}\}$ is compact and $u \mapsto e^{A_\gamma u}$ is continuous on $[0, T]$, $K$ can be taken uniformly on $[0, \bar{\gamma}]$.) Combine this with the $\varepsilon/2$ approximation from Theorem 2. $\qquad\square$

**Lemma 6** (Softmax $\ell_\infty \to \ell_1$ bound). *For any $x, y \in \mathbb{R}^m$,*

$$\|\mathrm{softmax}(x) - \mathrm{softmax}(y)\|_1 \leq \|x - y\|_\infty.$$

*Consequently, with logits scaled by $1/\sqrt{d_k}$ as in equation 79, the Lipschitz constant becomes $1/\sqrt{d_k}$.*

*Proof.* Let $s = \mathrm{softmax}(u)$. For any $v$ with $\|v\|_\infty \leq 1$, the softmax Jacobian satisfies

$$J_u v = \mathrm{diag}(s)v - s(s^\top v) = s \odot \big( v - (s^\top v)\mathbf{1} \big).$$

Hence

$$\|J_u v\|_1 = \sum_i s_i |v_i - t| \quad \text{with } t := s^\top v \in [-1, 1].$$

Maximizing over $\|v\|_\infty \leq 1$ is attained at $v_i \in \{\pm 1\}$. A direct calculation then gives $\sum_i s_i |v_i - t| = 1 - t^2 \leq 1$, so $\|J_u\|_{\infty \to 1} \leq 1$. By the mean value theorem along the segment $y + t(x - y)$,

$$\|\mathrm{softmax}(x) - \mathrm{softmax}(y)\|_1 \leq \int_0^1 \|J_{y+t(x-y)}(x - y)\|_1 \, dt \leq \|x - y\|_\infty.$$

For logits scaled by $1/\sqrt{d_k}$, the bound acquires the factor $1/\sqrt{d_k}$. $\qquad\square$

## C  $\mathbb{E}(3)$-EQUIVARIANCE

### C.1  GROUP ACTIONS, REPRESENTATIONS, AND $\mathbb{E}(3)$

A group action of $G$ on a set $X$ is a function $f : G \times X \rightarrow X$ such that:

1. $f(e, x) = x \quad \forall\, x \in X$
2. $f\big(g, f(h, x)\big) = f(gh, x) \quad \forall\, g, h \in G,\ x \in X$

$$\boxed{g \cdot x \equiv f(g, x)}$$

Eg - SO(3) acts on $\mathbb{R}^3$ by rotation, $R \cdot v = Rv$;    Translation group $\rightarrow t \cdot x = x + t$.

A representation of a group $G$ is a homomorphism $\varphi : G \rightarrow \mathrm{GL}(V)$ where $V$ is a vector space and $\mathrm{GL}(V)$ is the group of invertible linear transformations of $V$, i.e., for each group element $g$, we get a matrix $\varphi(g)$ such that

$$\varphi(gh) = \varphi(g)\, \varphi(h).$$

**Euclidean Group - $\mathbb{E}(3)$**

$$\mathbb{E}(3) = \mathrm{SO}(3) \ltimes \mathbb{R}^3 \quad \text{(semiproduct)}$$

An element $g \in \mathbb{E}(3)$ is a pair $(R, t)$ where $R \in \mathrm{SO}(3)$ is a rotation matrix, $t \in \mathbb{R}^3$ is a translation vector.

Group operation: $(R_1, t_1) \cdot (R_2, t_2) = (R_1 R_2,\ R_1 t_2 + t_1)$

Proof:  Given 2 transformations

$$(R_1, t_1)\ ;\ (R_2, t_2)$$

Their composition means: first apply $(R_2, t_2)$ then apply $(R_1, t_1)$.

A point $x \in \mathbb{R}^3$ transforms as,

$$(R_2, t_2) \cdot x = R_2 x + t_2$$

then applying $(R_1, t_1)$

$$(R_1, t_1) \circ (R_2 x + t_2) = R_1(R_2 x + t_2) + t_1 = (R_1 R_2)x + (R_1 t_2 + t_1).$$

So the combined transformation is:

$$(R_1, t_1) \cdot (R_2, t_2) = (R_1 R_2,\ R_1 t_2 + t_1).$$

Finally, we get the action on $\mathbb{R}^3$ as $(R, t) \cdot x = Rx + t$.

### C.2  SPHERICAL HARMONICS

Any point $r \in \mathbb{R}^3$ can be written as:

$$r = r\,(\sin\theta\cos\phi,\ \sin\theta\sin\phi,\ \cos\theta)$$

where $r \geq 0,\ 0 \leq \theta \leq \pi,\ 0 \leq \phi \leq 2\pi$.

**Laplacian in Spherical Coordinates**:

$$\nabla^2 = \frac{1}{r^2}\frac{\partial}{\partial r}\left(r^2\frac{\partial}{\partial r}\right) + \frac{1}{r^2\sin\theta}\frac{\partial}{\partial\theta}\left(\sin\theta\frac{\partial}{\partial\theta}\right) + \frac{1}{r^2\sin^2\theta}\frac{\partial^2}{\partial\phi^2}.$$

Solutions to the Laplace Eqn. using separation of variables can be written as

$$\{\nabla^2 f = 0\} \quad \Rightarrow \quad f(r, \theta, \phi) = R(r)\, Y(\theta, \phi).$$

The angular part $Y(\theta, \phi)$ gives spherical harmonics,

$$Y_\ell^m(\theta, \phi) = \sqrt{\frac{(2\ell+1)\,(\ell-|m|)!}{4\pi\,(\ell+|m|)!}}\ P_\ell^{|m|}(\cos\theta)\, e^{im\phi},$$

where $P_\ell^{|m|}$ are associated Legendre polynomials.

**Key Properties**

1) **Orthonormality**:

$$\int_0^\pi \int_0^{2\pi} Y_\ell^m(\Omega)\, Y_{\ell'}^{m'}(\Omega)^*\, \mathrm{d}\Omega = \delta_{\ell\ell'}\, \delta_{mm'}\,, \qquad \mathrm{d}\Omega = \sin\theta\, \mathrm{d}\theta\, \mathrm{d}\phi. \qquad (82)$$

2) **Completeness**: Any $f(\hat{r})$ on the sphere can be expanded in spherical harmonics.

3) **Rotation**:

$$Y_\ell^m(R^{-1}\hat{r}) = \sum_{m'} D_{mm'}^{(\ell)}(R)\, Y_\ell^{m'}(\hat{r})$$

or

$$Y_\ell^m(\hat{r}') = \sum_{m'} \left[D^{(\ell)}(R)\right]_{mm'}^* Y_\ell^{m'}(\hat{r});\, (\hat{r}' = R\,\hat{r})$$

## C.3  WIGNER $D$-MATRICES

$D^{(\ell)}(R)$ are the matrix representations of rotations in the $\ell^{\text{th}}$ irreducible representation (irrep).

A 3D rotation operator can be written as

$$R(\alpha, \beta, \gamma) = e^{-i\alpha\hat{J}_z}\, e^{-i\beta\hat{J}_y}\, e^{-i\gamma\hat{J}_z}\,, \qquad (83)$$

where $\alpha, \beta, \gamma$ are Euler angles and $\hat{J}_x, \hat{J}_y, \hat{J}_z$ are the components of angular momentum.

The Wigner $D$-matrix is a unitary square matrix of dimension $2j + 1$ in the spherical basis with elements

$$D_{mm'}^j(\alpha, \beta, \gamma) \equiv \langle jm|R(\alpha, \beta, \gamma)|jm'\rangle$$

$$= e^{-im\alpha}\, d_{mm'}^j(\beta)\, e^{-im'\gamma}$$

$$d_{mm'}^j(\beta) = \langle jm|e^{-i\beta\hat{J}_y}|jm'\rangle = D_{mm'}^j(0, \beta, 0)$$

Here $d_{mm'}^j$ is an element of the reduced Wigner $d$-matrix.

**Key Properties**

1) **Unitarity**:  $D^{(\ell)}(R)^\dagger = D^{(\ell)}(R^{-1})$.
2) **Group homomorphism**:  $D^{(\ell)}(R_1 R_2) = D^{(\ell)}(R_1)\, D^{(\ell)}(R_2)$.
3) **Orthogonality**:

$$\int_0^{2\pi} \mathrm{d}\alpha \int_0^\pi \mathrm{d}\beta\, \sin\beta \int_0^{2\pi} \mathrm{d}\gamma\, D_{m'k'}^{j'}(\alpha, \beta, \gamma)^*\, D_{mk}^j(\alpha, \beta, \gamma) = \frac{8\pi^2}{2j+1}\, \delta_{mm'}\, \delta_{kk'}\, \delta_{j'j}\,. \tag{84}$$

## C.4  TENSORS

The tensor product decomposes as

$$V_{\ell_1} \otimes V_{\ell_2} = \bigoplus_{\ell=|\ell_1-\ell_2|}^{\ell_1+\ell_2} V_\ell \qquad \text{(Direct Sum)}. \qquad (85)$$

### C.4.1  CLEBSCH-GORDON COEFFICIENTS AND QUANTUM MECHANICAL ADDITION OF ANGULAR MOMENTUM

The Clebsch-Gordan coefficients are the expansion coefficients:

$$|j_1 m_1\rangle \otimes |j_2 m_2\rangle = \sum_{j,m} \langle j_1 m_1, j_2 m_2|jm\rangle\, |jm\rangle\,. \qquad (86)$$

**Key Properties**

    1) **Selection rules**:

$$\langle j_1 m_1, j_2 m_2 | j' m' \rangle = 0 \quad \text{unless} \quad |j_1 - j_2| \leq j' \leq j_1 + j_2 \text{ and } m' = m_1 + m_2. \quad (87)$$

    2) **Orthogonality**: ($\langle jm | j_1 m_1, j_2 m_2 \rangle \equiv \langle j_1 m_1, j_2 m_2 | jm \rangle$):

$$\sum_{j=|j_1-j_2|}^{j_1+j_2} \sum_{m=-j}^{j} \langle j_1 m_1, j_2 m_2 \,|\, jm \rangle \, \langle jm \,|\, j_1 m_1', j_2 m_2' \rangle$$

$$= \langle j_1 m_1, j_2 m_2 \,|\, j_1 m_1', j_2 m_2' \rangle = \delta_{m_1 m_1'} \, \delta_{m_2 m_2'} \quad \text{(i)}$$

$$\sum_{m_1, m_2} \langle j' m' \,|\, j_1 m_1, j_2 m_2 \rangle \, \langle j_1 m_1, j_2 m_2 \,|\, jm \rangle = \langle j' m' \,|\, jm \rangle = \delta_{jj'} \, \delta_{mm'} . \quad \text{(ii)}$$

    3) Equivalence Relation to Wigner (D)-matrices

$$\int_0^{2\pi} \mathrm{d}\alpha \int_0^{\pi} \mathrm{d}\beta \, \sin\beta \int_0^{2\pi} \mathrm{d}\gamma \, D_{MK}^{J}(\alpha,\beta,\gamma)^* \, D_{m_1 k_1}^{j_1}(\alpha,\beta,\gamma) \, D_{m_2 k_2}^{j_2}(\alpha,\beta,\gamma)$$

$$= \frac{8\pi^2}{2J+1} \, \langle j_1 m_1 j_2 m_2 | JM \rangle \, \langle j_1 k_1 j_2 k_2 | JK \rangle \, .$$

    4) **Relation to spherical harmonics**

$$\int_{S^2} Y_{\ell_1}^{m_1}(\Omega)^* \, Y_{\ell_2}^{m_2}(\Omega)^* \, Y_L^M(\Omega) \, \mathrm{d}\Omega =$$

$$\sqrt{\frac{(2\ell_1+1)(2\ell_2+1)}{4\pi(2L+1)}} \, \langle \ell_1 0 \, \ell_2 0 \,|\, L0 \rangle \, \langle \ell_1 m_1 \, \ell_2 m_2 \,|\, LM \rangle \quad (88)$$

$$\implies Y_{\ell_1}^{m_1}(\Omega) \, Y_{\ell_2}^{m_2}(\Omega) =$$

$$\sum_{L,M} \sqrt{\frac{(2\ell_1+1)(2\ell_2+1)}{4\pi(2L+1)}} \, \langle \ell_1 0 \, \ell_2 0 \,|\, L0 \rangle \, \langle \ell_1 m_1 \, \ell_2 m_2 \,|\, LM \rangle \, Y_L^M(\Omega)$$

$$(89)$$

## C.5   Equivariance

A function $f : X \to Y$ is equivariant w.r.t. group actions $f_X$ on $X$ and $f_Y$ on $Y$ if

$$\boxed{f(f_X(g,x)) = f_Y(g, f(x)) \quad \forall g \in G, \, x \in X} \quad (90)$$

A geometric tensor of type $(\ell)$ is a $(2\ell+1)$-component object

$$T^{(\ell)} = \left( T_{-\ell}, \, T_{-\ell+1}, \, \ldots, \, T_{\ell} \right)^{\mathsf{T}} \quad (91)$$

that transforms under rotations $R \in \mathrm{SO}(3)$ as

$$T^{(\ell)'} = D^{(\ell)}(R) \, T^{(\ell)} . \quad (92)$$

$\ell = 0 \Rightarrow$ scalars,      $\ell = 1 \Rightarrow$ vectors.

Geometric tensors can be represented using spherical harmonics and radial basis functions:

$$T^{(\ell)}(\mathbf{r}, t) = \sum_{n=1}^{\infty} \sum_{m=-\ell}^{\ell} T_{nm}^{(\ell)}(t) \, R_n^{(\ell)}(r) \, Y_m^{\ell}(\hat{\mathbf{r}}), \qquad \hat{\mathbf{r}} = \frac{\mathbf{r}}{\|\mathbf{r}\|}. \quad (93)$$

where

    • $T_{nm}^{(\ell)}(t) \in \mathbb{C}$ are time-dependent coefficients,

- $R_n^{(\ell)}(r)$ are radial basis functions,
- $Y_m^\ell(\hat{\mathbf{r}})$ are the (complex) spherical harmonics.

This works because spherical harmonics are precisely the basis functions for irreducible representations of $SO(3)$.

We can use Peter-Weyl theorem to show that spherical harmonics form a complete orthonormal basis for $L^2(S^2)$. Combined with the completeness of an appropriate radial basis on $L^2(\mathbb{R}^+)$, the tensor product gives completeness on $L^2(\mathbb{R}^3)$. To start, Peter–Weyl theorem states: for a compact group $G$ (e.g. $SO(3)$),

$$L^2(G) = \bigoplus_{\ell \in \widehat{G}} V_\ell \otimes V_\ell^*, \tag{94}$$

i.e. every square-integrable function on the group decomposes into finite-dimensional irreducible representations of $G$.

$L^2(S^2)$: **Square-Integrable Functions on the Sphere**

$S^2$ is the unit sphere in $\mathbb{R}^3$, i.e. the set of all directions:

$$S^2 = \left\{ \hat{\mathbf{r}} \in \mathbb{R}^3 : \|\hat{\mathbf{r}}\| = 1 \right\}.$$

$L^2(S^2)$ is the space of all $f : S^2 \to \mathbb{C}$ such that

$$\int_{S^2} |f(\theta, \phi)|^2 \, d\Omega < \infty, \qquad d\Omega = \sin\theta \, d\theta \, d\phi.$$

The spherical harmonics $Y_\ell^m(\theta, \phi)$ form a complete orthonormal basis for $L^2(S^2)$. Hence any $f \in L^2(S^2)$ can be written as

$$f(\theta, \phi) = \sum_{\ell=0}^{\infty} \sum_{m=-\ell}^{\ell} a_{\ell m} Y_\ell^m(\theta, \phi). \tag{95}$$

$L^2(\mathbb{R}^+)$: **Radial Part**

Let $\mathbb{R}^+ = [0, \infty)$. Then

$$L^2(\mathbb{R}^+) = \left\{ f : [0, \infty) \to \mathbb{C} \;:\; \int_0^\infty |f(r)|^2 \, r^2 \, dr < \infty \right\}.$$

$L^2(\mathbb{R}^3)$: **Full 3-Dimensional Space**

This is the space of all square-integrable functions on $\mathbb{R}^3$, $f : \mathbb{R}^3 \to \mathbb{C}$, with

$$\int_{\mathbb{R}^3} |f(\mathbf{r})|^2 \, d^3\mathbf{r} < \infty.$$

In spherical coordinates $\mathbf{r} = (r, \theta, \phi)$, one naturally has the factorization

$$L^2(\mathbb{R}^3) \cong L^2(\mathbb{R}^+) \otimes L^2(S^2).$$

Therefore, the tensor product of a radial basis $R_n^{(\ell)}(r)$ and spherical harmonics $Y_m^\ell(\theta, \phi)$ gives a complete basis on $L^2(\mathbb{R}^3)$:

$$f(\mathbf{r}) = \sum_{n,\ell,m} a_{n\ell m} R_n^{(\ell)}(r) Y_m^\ell(\theta, \phi), \tag{96}$$

which is a complete representation for all square-integrable functions in $\mathbb{R}^3$.

**Radial Basis Functions:**

- **Gaussian-type orbitals (should work for our case):**

$$R_n^{(\ell)}(r) \ = \ N_n^{(\ell)} \, r^\ell \, e^{-\beta_n r^2}, \qquad \int_0^\infty \left| R_n^{(\ell)}(r) \right|^2 r^2 \, dr = 1. \tag{97}$$

- **Bessel functions (for problems with radial boundaries):**
  A convenient finite radial basis on a ball of radius $R$ is given by spherical Bessel functions:

$$R_n^{(\ell)}(r) \ = \ \sqrt{\frac{2}{R^3}} \, \frac{1}{\left| j_{\ell+1}(z_{n,\ell}) \right|} \, j_\ell\!\left( \frac{z_{n,\ell}\, r}{R} \right), \qquad j_\ell(z_{n,\ell}) = 0, \ z_{n,\ell} \text{ the } n\text{-th zero.} \tag{98}$$

With this thorough background, let us now tackle the bull by its horns: building $\mathbb{E}(3)$-equivariant neural networks. A standard layer $y = \sigma(Wx + b)$ is *not* equivariant.

The most general $\mathbb{E}(3)$-equivariant linear operation between geometric tensors is

$$T^{(\ell_{\text{out}})} = \sum_{\ell_{\text{in}}} \sum_\ell W^{(\ell_{\text{out}}, \ell_{\text{in}}, \ell)} \left[ T_{\text{in}}^{(\ell_{\text{in}})} \otimes Y^{(\ell)} \right]^{(\ell_{\text{out}})}, \tag{99}$$

where

- $T_{\text{in}}^{(\ell_{\text{in}})}$ is a tensor of type $(\ell_{\text{in}})$;
- $Y^{(\ell)}$ provides geometric information about relative positions;
- $[T_{\text{in}}^{(\ell_{\text{in}})} \otimes Y^{(\ell)}]^{(\ell_{\text{out}})}$ combines them using Clebsch-Gordan coefficients.;
- $W^{(\ell_{\text{out}}, \ell_{\text{in}}, \ell)}$ are scalar weights.

1) The tensor product $\left[ T^{(\ell_1)} \otimes T^{(\ell_2)} \right]^{(L)}$ is computed as

$$\left[ T^{(\ell_1)} \otimes Y^{(\ell_2)} \right]_m^{(L)} = \sum_{m_1=-\ell_1}^{\ell_1} \sum_{m_2=-\ell_2}^{\ell_2} \langle \ell_1 m_1, \ell_2 m_2 | L m \rangle \ T_{m_1}^{(\ell_1)} \, Y_{m_2}^{(\ell_2)}. \tag{100}$$

2) For a relative position vector $\mathbf{r}_{ij} = \mathbf{r}_j - \mathbf{r}_i$,

$$Y_\ell^m(\hat{\mathbf{r}}_{ij}) = Y_\ell^m(\theta_{ij}, \phi_{ij}), \qquad (\theta_{ij}, \phi_{ij}) \text{ are the spherical angles of } \hat{\mathbf{r}}_{ij} = \frac{\mathbf{r}_{ij}}{\|\mathbf{r}_{ij}\|}. \tag{101}$$

3) For a node $i$ with neighbours $N(i)$,

$$\overline{T}_i^{(\ell_{\text{out}})} = \sum_{j \in N(i)} \sum_{\ell_{\text{in}}} \sum_\ell W^{(\ell_{\text{out}}, \ell_{\text{in}}, \ell)} \left[ T_j^{(\ell_{\text{in}})} \otimes Y^{(\ell)}(\hat{\mathbf{r}}_{ij}) \right]^{(\ell_{\text{out}})}. \tag{102}$$

We claim that the above operation is $\mathbb{E}(3)$-equivariant.

**Proof:**
Consider a transformation $g = (R, t) \in \mathbb{E}(3)$

Under the transformation:

$$\mathbf{r}_i' = R\,\mathbf{r}_i + \mathbf{t},$$
$$\mathbf{r}_{ij}' = \mathbf{r}_i' - \mathbf{r}_j' = R(\mathbf{r}_i - \mathbf{r}_j) = R\,\mathbf{r}_{ij},$$
$$\hat{\mathbf{r}}_{ij}' = R\hat{\mathbf{r}}_{ij}.$$

Spherical harmonics transform as:

$$y^{(\ell)}(\hat{\mathbf{r}}_{ij}') = y^{(\ell)}(R\hat{\mathbf{r}}_{ij}) = D^{(\ell)}(R)\, y^{(\ell)}(\hat{\mathbf{r}}_{ij}).$$

Input tensors transform as:

$$T_j^{(\ell_{\mathrm{in}})\,\prime} \;=\; D^{(\ell_{\mathrm{in}})}(R)\,T_j^{(\ell_{\mathrm{in}})}.$$

The tensor product preserves equivariance,

$$\big[\,T_j^{(\ell_{\mathrm{in}})} \otimes y^{(\ell)}(\widehat{\mathbf{r}}'_{ij})\,\big]^{(\ell_{\mathrm{out}})} = D^{(\ell_{\mathrm{out}})}(R)\,\big[\,T_j^{(\ell_{\mathrm{in}})} \otimes y^{(\ell)}(\widehat{\mathbf{r}}_{ij})\,\big]^{(\ell_{\mathrm{out}})}.$$

Since weights are scalars, the output is:

$$T_i^{(\ell_{\mathrm{out}})\,\prime} \;=\; D^{(\ell_{\mathrm{out}})}(R)\,T_i^{(\ell_{\mathrm{out}})}.$$

This proves $\mathbb{E}(3)$-equivariance.

Finally, we look at the continuous-time generalization for ContiFormer.

Consider the architecture of the ContiFormer, described in the original paper Chen et al. (2023).

**Now instead of scalars $q, k, v \in \mathbb{R}^d$, we promote these to irreducible representations of $SO(3)$, written as:**

$$T^{(\ell)}(\mathbf{r}, t) \;\in\; \mathbb{R}^{2\ell+1}.$$

Each $T^{(\ell)}$ is a feature that transforms under rotation as:

For $(R, \mathbf{t}) \in \mathbb{E}(3)$,

$$T^{(\ell)\,\prime}(\mathbf{r}, t) \;=\; D^{(\ell)}(R)\,T^{(\ell)}\big(R^{-1}(\mathbf{r} - \mathbf{t}), t\big),$$

where $R^{-1}(\mathbf{r} - \mathbf{t})$ denotes the transformed coordinate.

Query, key, value Tensors:

$$Q^{(\ell_q)}(\mathbf{r}, t) = W_Q^{(\ell_q)}\,T^{(\ell_q)}(\mathbf{r}, t),$$
$$K^{(\ell_k)}(\mathbf{r}, t) = W_K^{(\ell_k)}\,T^{(\ell_k)}(\mathbf{r}, t),$$
$$V^{(\ell_v)}(\mathbf{r}, t) = W_V^{(\ell_v)}\,T^{(\ell_v)}(\mathbf{r}, t).$$

To allow for *rotational equivariance*, instead of using a dot product, we define a geometric inner product via tensor contraction:

$$\alpha(\mathbf{r}, t; \mathbf{r}_i, t_i) \;=\; \frac{1}{t - t_i} \int_{t_i}^{t} \sum_{\ell_2, m_2} Q^{(\ell_q)}(\mathbf{r}, \tau) \cdot \big[\,K^{(\ell_k)}(\mathbf{r}_i, \tau) \otimes Y^{(\ell)}(\widehat{\mathbf{r} - \mathbf{r}_i})\,\big]^{(\ell_q)}\, d\tau. \tag{103}$$

- $K \otimes Y$ is the combined key with spherical harmonics.
- Projection to type $\ell_q$ ensures match with $Q$.

This respects equivariance because $Y^{(\ell)}(\widehat{\mathbf{r}})$ transform under $SO(3)$ as irreducible representations, providing angular information.

The tensor product and Clebsch-Gordan decomposition ensures results transform predictably.

$\mathbb{E}(3)$-equivariant expected values:

$$V_{\exp}^{(\ell_v)}(\mathbf{r}, t; \mathbf{r}_i\, t_i) \;=\; \frac{1}{t - t_i} \int_{t_i}^{t} V^{(\ell_v)}(\mathbf{r}_i, \tau)\, d\tau. \tag{104}$$

Full attention update:

$$T_{\text{out}}^{(\ell_{\text{out}})}(\mathbf{r}, t) = \sum_{i=1}^{N} \sum_{\ell_v, \ell_{\text{mix}}} W_{\text{out}}^{(\ell_{\text{out}}, \ell_v, \ell_{\text{mix}})} \left[ \alpha(\mathbf{r}, t; \mathbf{r}_i, t_i) \cdot V_{\exp}^{(\ell_v)}(\mathbf{r}, t; \mathbf{r}_i, t_i) \otimes Y^{(\ell_{\text{mix}})}(\widehat{\mathbf{r} - \mathbf{r}_i}) \right]^{(\ell_{\text{out}})}.$$

$$(105)$$

The weights $W_{\text{out}}^{(\cdot)}$ are learnable scalar coeffecients over radial basis functions.

$\mathbb{E}(3)$-Equivariant Neural ODE:

$$\frac{\partial T^{(\ell)}(\mathbf{r}, t)}{\partial t} = f_{\text{contiformer}}^{(\ell)} \left[ \left\{ T^{(\ell')}(\cdot, t) \right\}_{\ell'} \right](\mathbf{r}) =$$
$$\underbrace{\text{CTAttn}^{(\ell)}(\mathbf{r}, t)}_{\text{modelling interaction b/w neighbouring nodes}} + \underbrace{\text{FFN}^{(\ell)}(\mathbf{r}, t)}_{\text{acting on each node independently}} \qquad (106)$$

Continuous-time attention (CTAttn):

$$\text{CTAttn}^{(\ell)}(\mathbf{r}, t) =$$
$$\int_{-\infty}^{t} \int_{\mathbb{R}^3} \rho(t - s) \sum_{\ell', \ell''} W_{\text{attn}}^{(\ell, \ell', \ell'')} \left[ \alpha(\mathbf{r}, t; \mathbf{r}', s) \, V_{\exp}^{(\ell')}(\mathbf{r}, t; \mathbf{r}', s) \otimes Y^{(\ell'')}(\widehat{\mathbf{r} - \mathbf{r}'}) \right]^{(\ell)} d\mathbf{r}' \, ds$$

$$(107)$$

where $\rho(t - s)$ is a temporal weighting function.

Finite temporal window for practical implementation:

$$\text{CTAttn}^{(\ell)}(\mathbf{r}, t) =$$
$$\int_{t-\Delta t}^{t} \int_{\|\mathbf{r}' - \mathbf{r}\| < \Delta r} \rho(t - s) \sum_{\ell', \ell''} W_{\text{attn}}^{(\ell, \ell', \ell'')} \left[ \alpha(\mathbf{r}, t; \mathbf{r}', s) \, V_{\exp}^{(\ell')}(\mathbf{r}, t; \mathbf{r}', s) \otimes Y^{(\ell'')}(\widehat{\mathbf{r} - \mathbf{r}'}) \right]^{(\ell)} d\mathbf{r}' \, ds$$

$$(108)$$

Let us check whether this is $\mathbb{E}(3)$-equivariant:

Under $(R, \mathbf{t}) \in \mathbb{E}(3)$,

$$T^{(\ell)}(\mathbf{r}, t) = D^{(\ell)}(R) \, T^{(\ell)}\big(R^{-1}(\mathbf{r} - \mathbf{t}), t\big).$$

Attention weight invariance:

$$\alpha'(\mathbf{r}, t; \mathbf{r}', s) = \alpha\big(R^{-1}(\mathbf{r} - \mathbf{t}), t; \; R^{-1}(\mathbf{r}' - \mathbf{t}), s\big).$$

Since the attention weights depend only on $\|\mathbf{r} - \mathbf{r}'\|$ and temporal differences, this property holds.

- The attention function $\alpha(\mathbf{r}, t; \mathbf{r}_i; t_i)$ is continuous in $t$ by construction of the continuity condition.
- The spherical harmonics $Y^{(\ell)}$ ensures smooth spatial variations.

## D Results Continued

| Model | Test accuracy (%) |
|---|---|
| † LMU (39) | 87.7 ± 0.1 |
| † LSTM (20) | 87.3 ± 0.4 |
| † GRU (30) | 86.2 ± n/a |
| † expRNN (41) | 84.3 ± 0.3 |
| † Vanilla RNN (49) | 67.4 ± 7.7 |
| *coRNN (42) | 86.7 ± 0.3 |
| LTC (1) | 61.8 ± 6.1 |
| **OsciFormer** | **93.3 ± 0.2** |

Table 5: Test accuracy comparison across different models

## E Attention Visualisation and Ablation

### E.1 Ablation Studies

| J Modes | Synthetic (Acc↑) | MIMIC (Acc↑) | Traffic (LL↑) | HR (RMSE↓) | MI (UCR) (Acc↑) |
|---|---|---|---|---|---|
| 1 | 0.752 ± 0.042 | 0.801 ± 0.008 | -0.892 ± 0.031 | 4.12 ± 0.35 | 48.2 ± 5.3 |
| 2 | 0.793 ± 0.038 | 0.816 ± 0.007 | -0.718 ± 0.028 | 3.45 ± 0.28 | 62.4 ± 4.1 |
| 4 | 0.828 ± 0.025 | 0.828 ± 0.006 | -0.612 ± 0.024 | 2.89 ± 0.22 | 78.7 ± 2.8 |
| 6 | 0.839 ± 0.014 | 0.833 ± 0.007 | -0.578 ± 0.021 | 2.67 ± 0.19 | 89.5 ± 0.8 |
| **8** | **0.841 ± 0.00** | **0.834 ± 0.007** | **-0.558 ± 0.025** | **2.56 ± 0.18** | **91.8 ± 0.2** |
| 12 | 0.841 ± 0.00 | 0.834 ± 0.007 | -0.557 ± 0.024 | 2.55 ± 0.18 | 91.7 ± 0.3 |
| 16 | 0.841 ± 0.01 | 0.834 ± 0.008 | -0.558 ± 0.025 | 2.56 ± 0.19 | 91.7 ± 0.3 |

Table 6: Effect of oscillator mode count (J) on downstream performance.

| J Modes | Synthetic (min) | MIMIC (min) | Traffic (min) | HR (min) | MI (min) |
|---|---|---|---|---|---|
| 1 | 0.18 ± 0.02 | 0.34 ± 0.03 | 0.41 ± 0.03 | 0.28 ± 0.02 | 0.52 ± 0.04 |
| 2 | 0.22 ± 0.02 | 0.42 ± 0.04 | 0.51 ± 0.04 | 0.35 ± 0.03 | 0.65 ± 0.05 |
| 4 | 0.31 ± 0.03 | 0.58 ± 0.05 | 0.71 ± 0.05 | 0.48 ± 0.04 | 0.91 ± 0.07 |
| 6 | 0.42 ± 0.03 | 0.79 ± 0.06 | 0.96 ± 0.07 | 0.65 ± 0.05 | 1.23 ± 0.09 |
| **8** | **0.56 ± 0.04** | **1.05 ± 0.08** | **1.28 ± 0.09** | **0.86 ± 0.06** | **1.64 ± 0.12** |
| 12 | 0.83 ± 0.06 | 1.56 ± 0.11 | 1.89 ± 0.13 | 1.27 ± 0.09 | 2.42 ± 0.18 |
| 16 | 1.11 ± 0.08 | 2.08 ± 0.15 | 2.51 ± 0.18 | 1.69 ± 0.12 | 3.21 ± 0.24 |

Table 7: Per-epoch training time as a function of oscillator modes (J).

| Damping Range | Synthetic (Acc↑) | MIMIC (Acc↑) | Traffic (LL↑) | HR (RMSE↓) | MI (Acc↑) |
|---|---|---|---|---|---|
| [0.00, 0.00] | 0.834 ± 0.02 | 0.829 ± 0.008 | -0.572 ± 0.026 | 2.68 ± 0.20 | 89.1 ± 0.8 |
| [0.01, 0.10] | 0.839 ± 0.01 | 0.832 ± 0.007 | -0.562 ± 0.025 | 2.61 ± 0.19 | 90.8 ± 0.5 |
| **[0.05, 0.40]** | **0.841 ± 0.00** | **0.834 ± 0.007** | **-0.558 ± 0.025** | **2.56 ± 0.18** | **91.8 ± 0.2** |
| [0.10, 0.60] | 0.840 ± 0.01 | 0.833 ± 0.007 | -0.559 ± 0.025 | 2.58 ± 0.18 | 91.5 ± 0.3 |
| [0.20, 0.80] | 0.837 ± 0.01 | 0.831 ± 0.008 | -0.564 ± 0.026 | 2.63 ± 0.19 | 90.7 ± 0.4 |
| [0.50, 1.00] | 0.828 ± 0.02 | 0.825 ± 0.009 | -0.581 ± 0.028 | 2.75 ± 0.21 | 88.9 ± 0.7 |

Table 8: Ablation over the initial damping range ($\zeta \sim \mathcal{U}[\zeta_{\min}, \zeta_{\max}]$).

| Grid Type | Synthetic (Acc↑) | Traffic (LL↑) | MI (Acc↑) | Time/epoch (min) |
|---|---|---|---|---|
| Linear [0.1, 10] | $0.836 \pm 0.01$ | $-0.565 \pm 0.025$ | $90.2 \pm 0.6$ | $0.62 \pm 0.05$ |
| Log-Uniform $[10^{-2}, 10^1]$ | $\mathbf{0.841 \pm 0.00}$ | $\mathbf{-0.558 \pm 0.025}$ | $\mathbf{91.8 \pm 0.2}$ | $\mathbf{0.56 \pm 0.04}$ |
| Random Uniform | $0.838 \pm 0.01$ | $-0.561 \pm 0.025$ | $91.1 \pm 0.4$ | $0.58 \pm 0.04$ |
| Geometric (sparse) | $0.834 \pm 0.02$ | $-0.567 \pm 0.026$ | $89.7 \pm 0.8$ | $0.54 \pm 0.04$ |
| Fixed Harmonics ($\omega_n = n\pi/L$) | $0.792 \pm 0.03$ | $-0.623 \pm 0.030$ | $82.4 \pm 1.2$ | $0.53 \pm 0.04$ |

Table 9: Impact of frequency grid parameterization.

| dataset | UD% | NearCrit% | OD% | median $\zeta$ | median $\omega_d$ (UD only) |
|---|---|---|---|---|---|
| neonate | 79.78 | 5.05 | 15.18 | 0.746 | 0.648 |
| traffic | 77.05 | 4.73 | 18.22 | 0.771 | 0.610 |
| mimic | 78.20 | 4.66 | 17.14 | 0.753 | 0.646 |
| stackoverflow | 78.25 | 5.21 | 16.54 | 0.759 | 0.668 |
| **bookorder** | **74.05** | 4.83 | **21.11** | **0.793** | **0.699** |

Table 10: Distribution of learned damping regimes by dataset.

| dataset | P($\zeta \geq 1.05$) | P($\zeta \geq 1.10$) | median $\zeta$ (after) |
|---|---|---|---|
| neonate | 0.1176 | 0.0690 | 0.7385 |
| traffic | 0.1465 | 0.0914 | 0.7641 |
| mimic | 0.1385 | 0.0832 | 0.7605 |
| stackoverflow | 0.1350 | 0.0777 | 0.7589 |
| **bookorder** | **0.1844** | **0.1191** | **0.8020** |

Table 11: Tail of the damping distribution across datasets.

## E.2 EXPERIMENTS- CLASSIFICATION

To make the resonance interpretation of our oscillator attention concrete, we construct a small, fully trainable experiment on synthetic irregular time series. The goal is to show that, after standard backpropagation on a simple prediction task, the learned attention weights follow the same resonance filter as that of a damped driven harmonic oscillator.

**Synthetic data:**  We consider a bank of $M = 41$ angular frequencies

$$\Omega = \{\omega_1, \ldots, \omega_M\}, \qquad \omega_m = \omega_{\min} + (m-1)\Delta\omega,$$

with $\omega_{\min} = 2\pi \cdot 0.5$ and $\Delta\omega = 2\pi \cdot 0.1$. Each training example is a short irregularly sampled trajectory of a *single* sinusoid with frequency $\omega_\star \in \Omega$ and random phase.

For each example:

1. We sample a label index $m_\star \sim \mathrm{Unif}\{1, \ldots, M\}$ and $\omega_\star = \omega_{m_\star}$.
2. We sample $L = 32$ time stamps $0 \leq t_1 < \cdots < t_L \leq T$ with $T = 5$ from a homogeneous Poisson process with rate $\lambda = 6$ and then re-normalize to $[0, T]$.
3. We sample an amplitude $A \sim \mathrm{Unif}[0.8, 1.2]$ and phase $\phi \sim \mathrm{Unif}[0, 2\pi]$. For each $t_\ell$, form the two–dimensional observation

$$x_\ell = \begin{bmatrix} A\cos(\omega_\star t_\ell + \phi) \\ A\sin(\omega_\star t_\ell + \phi) \end{bmatrix} + \varepsilon_\ell, \quad \varepsilon_\ell \sim \mathcal{N}(0, 0.05^2 I_2).$$

The target is the class index $m_\star$, i.e. the model must recover which frequency generated the sequence from irregular samples and additive noise. We generate $50,000$ sequences for training, $10,000$ for validation, and $10,000$ for testing.

**Model:**  We use a single head oscillator attention layer followed by a small classifier. Each input pair $(x_\ell, t_\ell)$ is first embedded to $d = 32$ dimensions via a linear map $E : \mathbb{R}^2 \to \mathbb{R}^d$; this produces token embeddings $h_\ell = Ex_\ell$.

For each token $h_\ell$ we instantiate a key and value oscillator with independent frequencies and damping per hidden coordinate:

$$\ddot{k}_c(t) + 2\gamma_c^{(k)}\dot{k}_c(t) + \left(\omega_c^{(k)}\right)^2 k_c(t) = F_c^{(k)}(t), \qquad \ddot{v}_c(t) + 2\gamma_c^{(v)}\dot{v}_c(t) + \left(\omega_c^{(v)}\right)^2 v_c(t) = F_c^{(v)}(t),$$

with closed-form solutions derived in Appendix A. The driving terms $F^{(k)}(t)$ and $F^{(v)}(t)$ are sinusoidal functions of time whose amplitudes are linear functions of $h_\ell$; in particular, each coordinate sees a weighted sum of $\cos(\cdot)$ and $\sin(\cdot)$ terms evaluated at $t_\ell$. We anchor the oscillator state at $t_\ell$ and evaluate the trajectories on $[t_\ell, T]$ using the analytic expressions.

A single query $q(t)$ is defined for the final prediction time $T$. We parameterise $q$ as a truncated sinusoidal basis,

$$q(t) = \sum_{j=1}^{J} \left(A_j \cos(\tilde{\omega}_j t) + B_j \sin(\tilde{\omega}_j t)\right),$$

with $J = 8$ and learnable coefficients $A_j, B_j \in \mathbb{R}^d$ and fixed frequencies $\tilde{\omega}_j$ on the same grid as $\Omega$. The continuous-time attention logit from token $i$ to the query at $T$ is

$$\alpha_i(T) = \frac{1}{T - t_i} \int_{t_i}^{T} \langle q(\tau), k_i(\tau) \rangle \, d\tau,$$

which we evaluate in closed form using the oscillator formulas from Appendix A.5. The attention weights are

$$w_i(T) = \frac{\exp(\alpha_i(T)/\sqrt{d})}{\sum_{j=1}^{L} \exp(\alpha_j(T)/\sqrt{d})}.$$

The attended value is $\bar{v}(T) = \sum_{i=1}^{L} w_i(T)v_i(T)$, followed by a two-layer MLP with hidden width 64 and ReLU nonlinearity that maps $\bar{v}(T)$ to $M$ logits. We train all parameters end-to-end with cross-entropy loss.

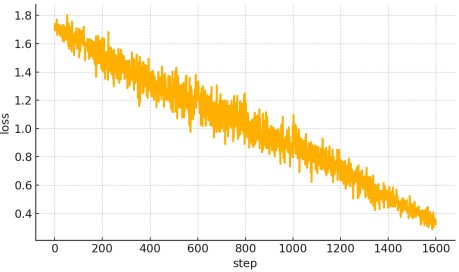

(a) Training loss for the Classification on synthetic irregular task.

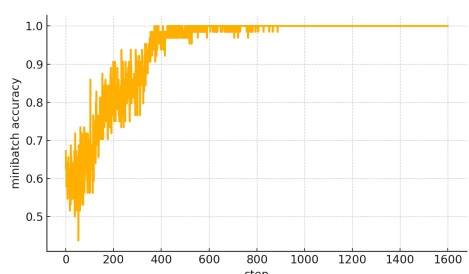

(b) Classification accuracy vs Time Steps

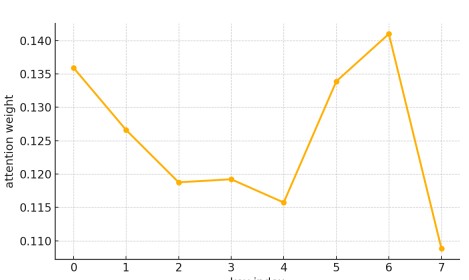

(c) Average attention weights over the eight oscillator keys at random Initialisation.

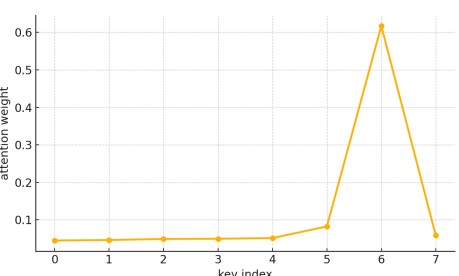

(d) Average attention weights after training.

Figure 3

**Training Method:** We optimise with Adam (learning rate $10^{-3}$, weight decay $10^{-2}$) for 200 epochs, batch size 128, and early stopping on validation accuracy. All oscillator frequencies are initialised by sampling $\omega_c^{(k)}, \omega_c^{(v)}$ log-uniformly from $[10^{-2}, 10^1]$ on the rescaled interval $[0, 1]$; damping factors are initialised in $[0.05, 0.4]$. The query basis frequencies $\tilde{\omega}_j$ are fixed to a subset of $\Omega$ and only their amplitudes are learned.

**Visualisations:** To relate the learned attention to resonance, we inspect the model after training and compute the following quantities:

1. The resonance amplitude profile $|H_i(\omega)| = \frac{1}{\sqrt{(\omega_{0,i}^2 - \omega^2)^2 + (2\gamma_i\omega)^2}}$ for each learned key $i$ using its trained parameters $(\omega_{0,i}, \gamma_i)$.
2. The phase-dependent attention map $\alpha(\omega, \varphi)$ across the frequency-phase plane for individual keys.
3. The maximum achievable attention $\alpha_{\max}(\omega) = \max_\varphi[\alpha(\omega, \varphi)]$ and the optimal phase $\varphi^*(\omega) = \arg H(\omega)$ that yields this maximum.
4. The attention weight distribution across keys for validation examples, both before and after training.
5. The confusion matrix of average attention weights (rows = true class, columns = keys) to verify that attention concentrates on keys whose natural frequencies match the signal's dominant frequency.

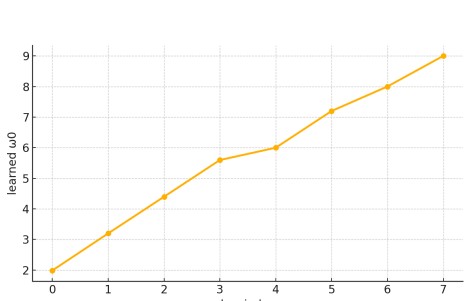

(a) Learned natural frequencies for the eight oscillator keys

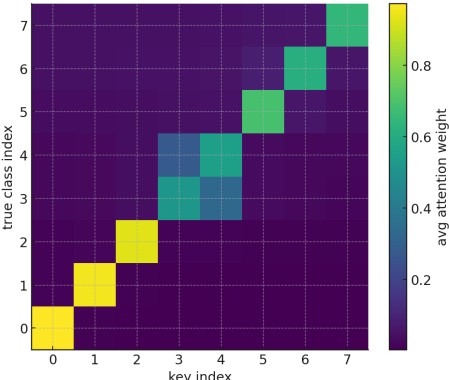

(b) Confusion matrix of mean attention weights

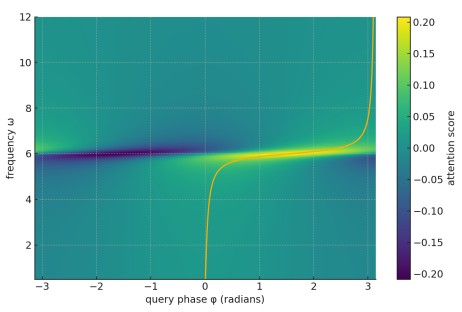

(c) Phase–frequency attention $\alpha(\omega, \varphi)$ for a representative key. The bright ridge in the $(\omega, \varphi)$ plane indicates the resonance region.

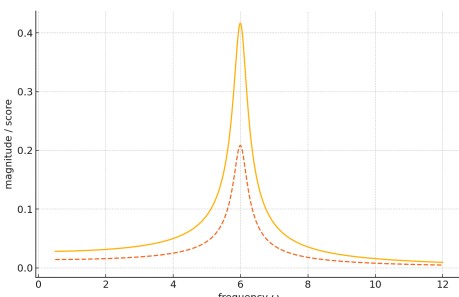

(d) Magnitude of the analytical transfer function $|H(\omega)|$ and the corresponding maximal learned attention response $\alpha_{\max}(\omega)$ as functions of driving frequency.

Figure 4

### E.3 EXPERIMENTS- REGRESSION

We consider a small 1D forecasting task designed to expose the internal behaviour of the oscillator-based attention model.

**Task:** Each sequence is generated as a sum of 1–3 cosine components

$$y(t) = \sum_k a_k \cos(\omega_k t), \qquad \omega_k \in \{2.0, 3.2, 4.4, 5.6, 6.0, 7.2, 8.0, 9.0\},$$

with random amplitudes $a_k$. The process is observed on an irregular time grid $0 < t_1 < \cdots < t_N < T_{\text{future}}$. The gaps $t_{n+1} - t_n$ are i.i.d. draws from a Gamma distribution, so both the number of points and their locations vary from sequence to sequence. Each observation is corrupted with independent Gaussian noise,

$$y_n^{\text{obs}} = y(t_n) + \varepsilon_n, \qquad \varepsilon_n \sim \mathcal{N}(0, \sigma^2).$$

The prediction target is a single future value

$$y_{\text{target}} = y(T_{\text{future}}), \qquad T_{\text{future}} = 7.0.$$

Thus, the model must forecast a future point of a multi-frequency signal from noisy, irregularly sampled observations.

**Features:** For each sequence we compute trigonometric features on the irregular grid that approximate the cosine and sine coefficients of the trajectory. For a fixed set of analysis frequencies $(\omega_j)_j$ (the same grid as above), we form

$$A_j \approx \frac{2}{T} \int_0^T y(t) \cos(\omega_j t)\, dt, \qquad B_j \approx \frac{2}{T} \int_0^T y(t) \sin(\omega_j t)\, dt,$$

using the trapezoidal rule on $\{(t_n, y_n^{\text{obs}})\}_n$. We then define the energy $E_j = A_j^2 + B_j^2$ and use stabilized, normalized features

$$Z_j = \frac{\log(1 + E_j) - \mu_j}{\sigma_j},$$

where $(\mu_j, \sigma_j)$ are the empirical mean and standard deviation of $\log(1 + E_j)$ over the training set. This provides a data-driven approximation to a sinusoidal expansion of the query.

**Model:** The attention mechanism mirrors the oscillator-based formulation in the main text. We use $K = 8$ keys. Key $i$ is parameterised by a natural frequency $\omega_{0,i}$ and a damping coefficient $\gamma_i$, and is associated with the standard second-order transfer function magnitude

$$H_i(\omega) = \frac{1}{\sqrt{(\omega_{0,i}^2 - \omega^2)^2 + (2\gamma_i \omega)^2}}.$$

Given the feature vector $Z$, we form a non-negative "query spectrum"

$$Q_j = \text{softplus}(w_j Z_j + b_j),$$

with learned scalars $w_j$ and $b_j$. The attention logit for key $i$ is then

$$\alpha_i = \sum_j Q_j \left| H_i(\omega_j) \right|.$$

Applying a softmax over $(\alpha_i)_i$ yields attention weights

$$\tilde{w}_i = \frac{\exp(\alpha_i)}{\sum_{k=1}^K \exp(\alpha_k)}.$$

The model predicts the target as a convex combination of learned values $v_i$,

$$\hat{y} = \sum_{i=1}^K \tilde{w}_i v_i.$$

All quantities $(\omega_{0,i}, \gamma_i, w_j, b_j, v_i)$ are trained end-to-end with backpropagation.

**Training setup:** We generate $2000$ training sequences and $400$ validation sequences. The network is trained with mean-squared error loss, using Adam as the optimiser. As a simple baseline we also evaluate a constant predictor $\hat{y} = \mathbb{E}[y_{\text{target}}]$ estimated on the training set.

On the validation set the constant baseline attains an MSE of $\approx 0.78$ with $\text{std}(y_{\text{target}}) \approx 0.88$. The learned oscillator model reaches a validation MSE of $\approx 0.10$, corresponding to an RMSE of $\approx 0.31$ and a correlation of $\approx 0.94$ between $\hat{y}$ and $y_{\text{target}}$. Thus the model reduces the error by roughly $65\%$ relative to the constant predictor while keeping the setting small enough that we can inspect the learned resonance structure.

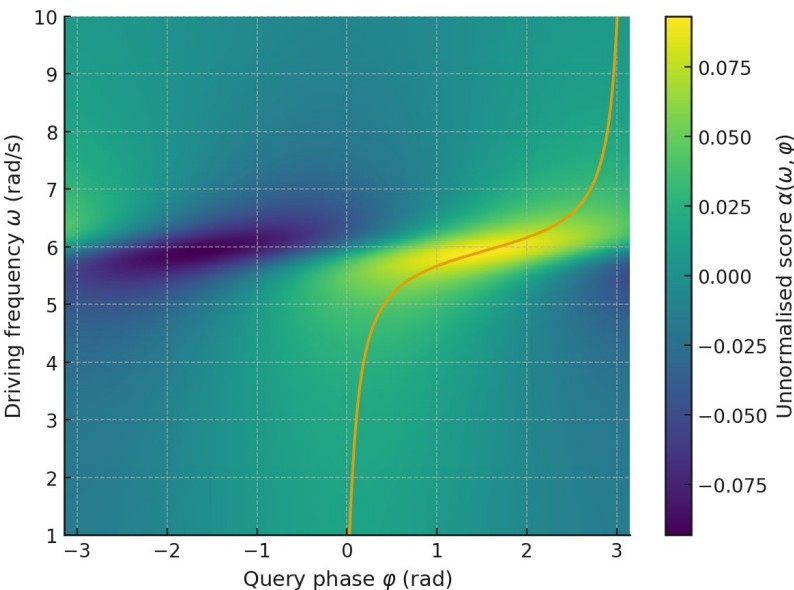

Figure 5: Phase–frequency attention $\alpha(\omega, \varphi)$ for a representative key. The bright ridge in the $(\omega, \varphi)$ plane indicates the resonance region.

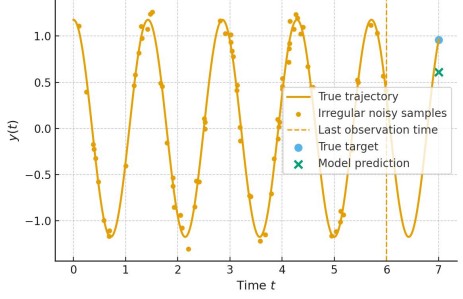

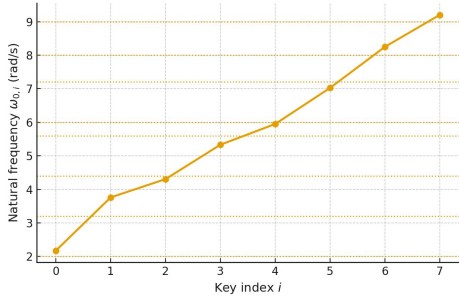

(a) Sequence from the 1-D regression task: true underlying trajectory (line), irregular noisy observations (dots), final observation time, and the true versus predicted future target at $T_{\text{future}} = 7$.

(b) Learned natural frequencies of the eight oscillator keys

Figure 6

# F    CHAOTIC SYSTEMS AND FAIL CASES

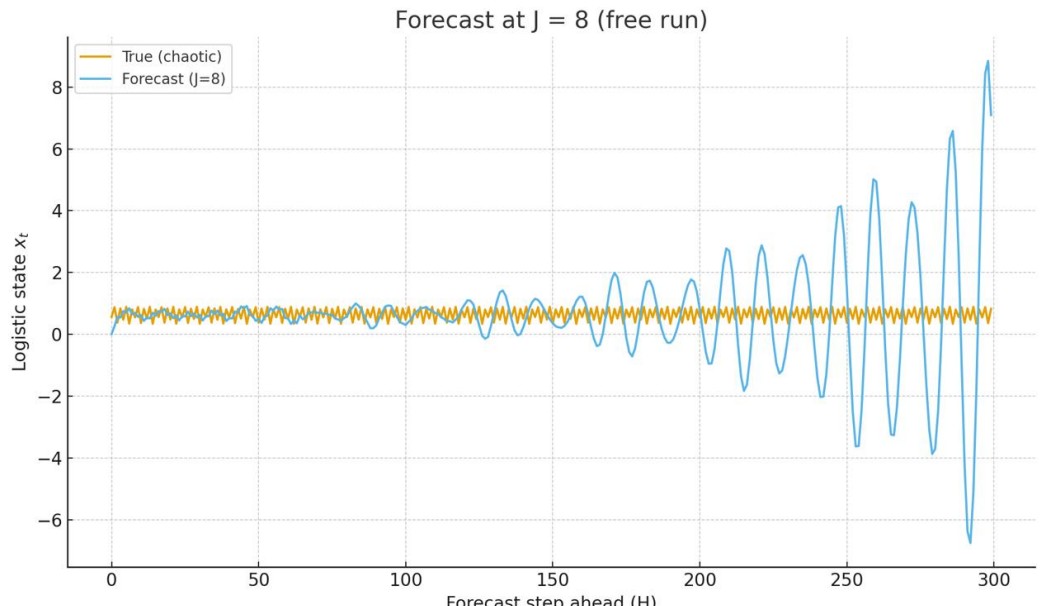

Figure 7: Forecast on the chaotic logistic map with $J = 8$ oscillator modes.

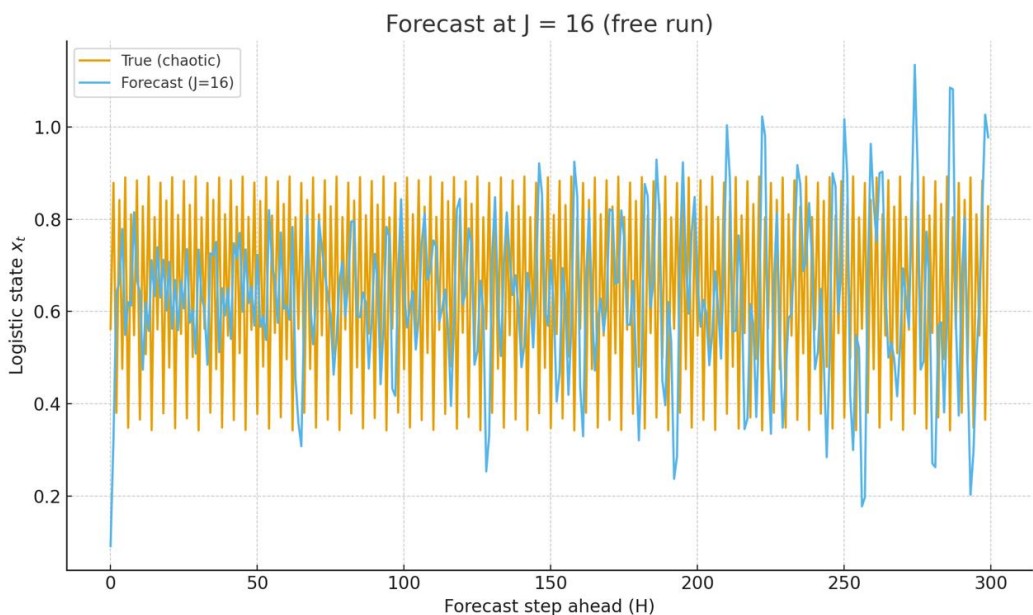

Figure 8: Forecast on the chaotic logistic map with $J = 16$ oscillator modes.

To illustrate a clear failure case, we run a small chaos experiment on the logistic map. The system is one–dimensional and is defined by

$$x_{t+1} = r\,x_t(1 - x_t), \qquad r = 3.57,\ x_0 = 0.6. \tag{109}$$

For this choice of $r$ the map is chaotic and has a positive Lyapunov exponent. Small errors in $x_t$ grow exponentially over time, so long-horizon prediction is intrinsically hard.

We generate a long sequence from the map and train our model in a one-step-ahead fashion. The model sees a short window of past values and is asked to predict $x_{t+1}$. At test time we perform a *free run*: we seed the model with a short true window and then feed back its own predictions for $H$ steps.

Figures 7 and 8 show free-running forecasts for two oscillator-bank sizes. With $J = 8$ modes, the model quickly leaves the attractor and produces oscillations with unrealistic amplitude. Increasing to $J = 16$ keeps the forecast bounded in the right range, but the trajectory still decorrelates from the true chaotic path after a few steps.

