# OpenReview forum: "Oscillators Are All You Need: Irregular Time Series Modelling via Damped Harmonic Oscillators with Closed-form Solutions"
_ICLR.cc/2026/Conference — Submitted to ICLR 2026_

### Official Review · Reviewer_EAsp · 2025-10-29

**Soundness:** 2
**Presentation:** 2
**Contribution:** 2
**Rating:** 2
**Confidence:** 4

**Summary:**

This paper presents OsciFormer, an architecture for irregular time series modeling. The model addresses the computational bottleneck of prior work, specifically ContiFormer, which uses numerical solvers for its Neural ODE components. OsciFormer replaces the general NODE dynamics with a linear damped harmonic oscillator, a system with a known closed-form analytical solution.

**Strengths:**

- By replacing the $O(S)$ sequential solver steps with a closed-form calculation, the dominant complexity term is reduced from $O(N^2 S d^2)$ to $O(N^2 J d)$, resulting in significant, verifiable speedups.

- OsciFormer can run on long-context tasks (like the HR benchmark in Table 3) where the baseline ContiFormer fails due to out-of-memory errors, demonstrating a practical engineering advantage.

- The paper provides detailed derivations for its closed-form solution (Appendix A) and a formal proof (Appendix B)  that its linear approximation does not sacrifice the universal approximation capability.

**Weaknesses:**

- This work is an incremental refinement of ContiFormer. The contribution is limited to replacing the Neural ODE $f(\tau, k_i(\tau); \theta_k)$ with a classic linear dynamical system that has a known solution.

- The title "Oscillators Are All You Need" is hyperbolic. It is unsuitable for complex non-linear, non-oscillatory, or discontinuous dynamics, a limitation the paper fails to address.

- The comparisons omit critical contemporary baselines for irregular time series, such as variations of Neural CDEs, Stable SDEs, and the recent advances in neural differential equations.

- Notably, it seems that the paper doesn't follow the ICLR 2026 template.

**Questions:**

- Please provide the ablation study that supports the claim. Specifically, show the sensitivity of accuracy, training time, and memory usage to the number of modes $J$ (e.g., $J = 4, 8, 16, 32, 64$).

- How is the query function $q(t)$ approximated? The paper mentions expanding it in a sinusoidal basis but provides no details on this process, the number of modes used, or its sensitivity.

- How does OsciFormer perform on tasks known to be non-linear or non-oscillatory (e.g., datasets with sharp, abrupt step-changes)? The rigid linear damped harmonic oscillator bias seems unsuited for such dynamics.

- What is the justification for including Section 5 on E(3)-equivariance?

- Can you provide any analysis of the learned oscillator parameters ($\omega$, $\gamma$)? Without this, the resonance framing is not supported.

---

> ### Author Response · Authors · 2025-11-23
> **Author Response to Reviewer EAsp**
>
> ---
> We thank the reviewer for their detailed feedback. We respond to the identified weaknesses and questions below.
>
> > **Weakness 1:**
> > This work is an incremental refinement of ContiFormer. The contribution is limited to replacing the Neural ODE $f(\tau, k_i(\tau); \theta_k)$ with a classic linear dynamical system that has a known solution.
>
> **Response to W1**
>
> Our work addresses the most fundamental bottleneck of the ContiFormer architecture, which is its computational complexity. This computational overhead of the Neural ODE is the main reason why this architecture is not more widely deployed in real-life systems. In addition to this, we also draw an elegant parallel to a physical system that naturally provides a representation for attention in the form of resonance. This aligns with a broader class of research that deals with the underlying physics of Artificial Intelligence.
>
> ---
> > **Weakness 2:**
> > The title "Oscillators Are All You Need" is hyperbolic. It is unsuitable for complex non-linear, non-oscillatory, or discontinuous dynamics, a limitation the paper fails to address.
>
> **Response to W2**
>
> We thank the reviewer for raising this concern. The tasks we evaluate are indeed non-linear, non-oscillatory, etc. The results are given in the main paper. Although our theorem is asymptotic, our experiments show that the gains effectively saturate, which we attribute to the limited complexity of these datasets, wherein the informative bits are easily captured by the limited number of modes.
>
> ---
>
> > **Weakness 3:**
> > The comparisons omit critical contemporary baselines for irregular time series, such as variations of Neural CDEs, Stable SDEs, and the recent advances in neural differential equations.
>
> **Response to W3**
>
> We agree that Neural CDEs are important to irregular time-series modelling. We have benchmarked our model against both Neural CDE / Neural RDE. On the HR irregular clinical benchmark (Table 3, page 8), we compare against ODE‑RNN, Neural‑CDE, and Neural‑RDE, in addition to GRU and Transformer baselines. OsciFormer achieves the lowest RMSE (2.56 ± 0.18) and improves over both Neural‑CDE (9.82 ± 0.34) and Neural‑RDE (2.97 ± 0.45). Furthermore, on the long‑context UCR classification suite (Table 2, page 8) we include NCDE, NRDE, and LogNCDE as strong CDE‑style continuous‑time baselines alongside LRU, S5/S6, Mamba, Transformer, and Rough Transformer. OsciFormer attains the top average accuracy (64.5%), on par with or better than the best baseline across datasets, while remaining faster than the ODE‑based method.
> We acknowledge that neural SDEs are a natural extension of Neural ODE/CDE. However, due to computational overheads of Stable SDEs, we weren’t able to include them in the benchmark. Additionally, Stable SDE (Oh et al., 2025) does not report Contiformer or any other attention-based mechanism, so we were unable to use their results. We also remain enthusiastic about extending our oscillator-based dynamics to SDEs in future.
>
> ---
>
> > **Weakness 4:**
> > Notably, it seems that the paper doesn't follow the ICLR 2026 template.
>
> **Response to W4**
> We thank you for pointing this out and have fixed the problem.
>
> Due to the character limit, we answer the questions in the next comment.
>
> ---

---

> ### Author Response · Authors · 2025-11-23
> **Author Response to Reviewer EAsp**
>
> ---
>
> > **Question 1:**
> > Please provide the ablation study that supports the claim. Specifically, show the sensitivity of accuracy, training time, and memory usage to the number of modes \(J\) (e.g., \(J = 4, 8, 16, 32, 64\)).
>
> **Response to Q1.**
> We have conducted detailed ablation studies to answer this, elucidated in the general comment.
>
> ---
>
> > **Question 2:**
> > How is the query function \(q(t)\) approximated? The paper mentions expanding it in a sinusoidal basis but provides no details on this process, the number of modes used, or its sensitivity.
>
> **Response to Q2.**
> We start with a continuous query function $\tilde{q}(t)$ (from cubic spline interpolation of discrete queries as in the original ContiFormer paper). We approximate it using a sum of sinusoids  $$ q(t) = \sum_{k=1}^{N} \big( A_k \cos(\omega_k t) + B_k \sin(\omega_k t) \big),  $$
> where we choose $N$ frequencies $\omega_1,\ldots,\omega_N$ as our basis. Given $M$ sample points $s_1,\ldots,s_M$, we evaluate $\tilde{q}(t)$ at those times and form a design matrix $\Phi \in \mathbb{R}^{M \times 2N}$ with rows
> $$
> \Phi_m = [\cos(\omega_1 s_m), \sin(\omega_1 s_m), \ldots, \cos(\omega_N s_m), \sin(\omega_N s_m)].
> $$
> For each query dimension, we solve the least-squares problem $$ \min_c \|\Phi c - y\|_2^2, $$
> where $y$ contains the sampled values of $\tilde{q}(t)$. This yields coefficients that we collect into $A_k, B_k \in \mathbb{R}^{d_k}$, giving the sinusoidal approximation $q(t)$ that best fits the original spline.
>
>
>
>
> ---
>
> > **Question 3:**
> > How does OsciFormer perform on tasks known to be non-linear or non-oscillatory (e.g., datasets with sharp, abrupt step-changes)? The rigid linear damped harmonic oscillator bias seems unsuited for such dynamics.
>
> **Response to Q3.**
> Our Harmonic Approximation Theorem guarantees that ContiFormer’s continuous keys can be approximated arbitrarily well by a shared harmonic oscillator bank. Furthermore, we have tested on datasets with sharp and abrupt changes. Details of these are in the general comment.
>
> ---
>
> > **Question 4:**
> > What is the justification for including Section 5 on E(3)-equivariance?
>
> **Response to Q4.**
> Thank you for your question. We have answered this in the general comment.
>
> ---
>
> > **Question 5:**
> > Can you provide any analysis of the learned oscillator parameters \((\omega, \gamma)\)? Without this, the resonance framing is not supported.
>
> **Response to Q5.**
> Thank you for raising this concern. We have conducted ablation studies to answer this, details of which are given in the general comment.
>
> ---
>
> **References**
>
> - YongKyung Oh, Dong-Young Lim, and Sungil Kim. *Stable Neural Stochastic Differential Equations in Analyzing Irregular Time Series Data.* arXiv preprint arXiv:2402.14989, 2025. [Available online](https://arxiv.org/abs/2402.14989).
>
> - Zeyuan Allen-Zhu and Yuanzhi Li. *Physics of Language Models: Part 1, Learning Hierarchical Language Structures.* arXiv preprint arXiv:2305.13673, 2025. [Available online](https://arxiv.org/abs/2305.13673).

---

> ### Comment · Reviewer_EAsp · 2025-11-26
>
> Thanks to the detailed explanation. However, some major concerns remain. Theorem 2 depends on Lemma 5, which assumes small damping. The empirical distribution in Table 11 shows many cases with ($\zeta > 1$). This breaks the assumption for the trained model. The failure on the logistic map also contradicts the universal scope suggested in the title. The results indicate that the method depends on non oscillatory behavior, which conflicts with the main motivation. While I have raised the score, add more detail on the remaining concerns.

---

> > ### Author Response · Authors · 2025-11-28
> >
> > We thank the reviewer for their response and address the queries below.
> >
> > ---
> >
> > **Logistic map and chaotic data**
> >
> > We would like to clarify that the logistic-map experiment in Appendix F is explicitly presented as a *failure case* rather than a positive result. The prediction errors grow exponentially over time, and thus long-horizon rollouts are unstable. This is not specific to our formulation but is common in most Transformer-based architectures (Hu et al., 2024; He et al., 2025).
> >
> > ---
> >
> > **Harmonic Approximation Theorem**
> >
> > Our Harmonic Approximation Theorem is an existence result for expressivity. We specifically state that, for any continuous key trajectories, there exists an integer $M$ and parameters of an undamped oscillator, and initial states, such that the oscillator keys approximate the original keys and therefore the attention is arbitrarily close.
> >
> > More formally, let $F_{\mathrm{Conti}}$ be the set of attention maps attainable by ContiFormer’s continuous keys, and let $F_{\mathrm{osc}}(\Gamma)$ be the set of maps realised by an oscillator bank with frequencies on our fixed grid and damping coefficients $0 \le \gamma_n \le \Gamma$. We show that, for any continuous keys, there exists a zero-damped bank (that is, $\Gamma = 0$) in $F_{\mathrm{osc}}(0)$ that approximates the same attention arbitrarily well.
> >
> > In our experiments, we allow $\gamma_n \ge 0$ without any hard upper bound, so the trained model is in
> > $$
> > F_{\mathrm{osc}}^{\mathrm{train}} = \bigcup_{\Gamma \ge 0} F_{\mathrm{osc}}(\Gamma),
> > $$
> > which clearly contains $F_{\mathrm{osc}}(0)$. Therefore, the universality property we prove for the undamped subset automatically holds for the full trained model class, because the closure of $F_{\mathrm{osc}}(0)$ (in operator norm) is contained in the closure of $F_{\mathrm{osc}}^{\mathrm{train}}$.
> >
> > Our theorem never claims that the specific parameters obtained by SGD will be those undamped (or small-damped) ones constructed in the proof.
> >
> > We would also like to clarify that Lemma 5 states that, on any finite time interval $[0,T]$, the map
> > $$
> > \gamma \mapsto e^{A_\gamma t}
> > $$
> > is continuous in operator norm, so that
> > $$
> > \sup_{t \in [0,T]} \bigl\| C \bigl( e^{A_\gamma t} - e^{A_0 t} \bigr) \bigr\|
> > \le
> > K \max_n \gamma_n
> > $$
> > for some constant $K$ depending only on $T$ and the frequency grid. We use this in Corollary 2 to say: if we slightly perturb the undamped approximant by some bounded damping $0 \le \gamma_n \le \bar{\gamma}$ (with $\bar{\gamma}$ chosen small enough), we still approximate the original keys and attention up to $\varepsilon$.
> >
> > All our formulations and derivations cover overdamped, underdamped, and critically damped regimes (see Appendix A).
> >
> > These types of existence results are standard in the literature. For example, Lanthaler et al. (2023) prove that multi-layer neural oscillator ODEs can approximate any causal, continuous operator. Their result is an existence argument as well; they do not claim that a trained model will satisfy the technical assumptions (causality, compact input set, etc.) or achieve that approximation in practice.
> >
> > Further, Rusch and Russ (2024) prove that LinOSS is universal for continuous, causal operators between time-varying functions under a structural constraint on the state matrix, which is again an existence result. Classical approximation theorems also rely on similar existence criteria (Cybenko, 1989; Hornik, 1991).
> >
> > ---
> >
> > **Resonance and damping**
> >
> > Resonance is sharpest in the under-damped regime because the system retains oscillatory energy for many cycles, allowing the driving frequency to build up a large steady-state amplitude.
> >
> > As damping increases, the oscillations die out more quickly, so the “peak” in the frequency response becomes broader and lower.
> >
> > In the critically damped case, there is no true oscillation, but the system still responds most strongly near its natural frequency, giving a mild resonance-like enhancement.
> >
> > In the over-damped regime, the motion is purely exponential and non-oscillatory, yet the frequency response still shows preferential amplification around a characteristic frequency.
> >
> > Thus, although sharp resonance disappears with large damping, the system continues to behave as a frequency-selective filter across all regimes.
> >
> > ---
> >
> > **Oscillators vs oscillatory behaviour**
> >
> > We would also like to clarify that damped driven harmonic oscillators do not just behave in an oscillatory way, and our main motivation for using them is that they give a rich system where there are both underdamped and overdamped regimes. We fundamentally believe that oscillators are good, and we are consistent with the overall growing use of oscillators for neural networks. ( Rusch, 2024 ) (Ceni et al., 2024)
> >
> > Due to the page limits, we continue the References in the following comment.

---

> > > ### Author Response · Authors · 2025-11-28
> > > **Continued Response to Reviewer EAsp**
> > >
> > > **References**
> > >
> > > Hu, J., Hu, Y., Chen, W., Jin, M., Pan, S., Wen, Q., & Liang, Y. (2024). Attractor Memory for Long-Term Time Series Forecasting: A Chaos Perspective. arXiv preprint arXiv:2402.11463. Available at: https://arxiv.org/abs/2402.11463.
> > >
> > > He, Y., Yang, Y., Cheng, X., Wang, H., Xue, X., Chen, B., & Hu, Y. (2025). Chaos Meets Attention: Transformers for Large-Scale Dynamical Prediction. arXiv preprint arXiv:2504.20858. Available at: https://arxiv.org/abs/2504.20858.
> > >
> > > Cybenko, G. (1989). Approximation by superpositions of a sigmoidal function. Mathematics of Control, Signals and Systems, 2(4), 303–314.
> > >
> > > Hornik, K. (1991). Approximation capabilities of multilayer feedforward networks. Neural Networks, 4(2), 251–257.
> > >
> > > Rusch, T. K., & Rus, D. (2025). Oscillatory State-Space Models. In Proceedings of the International Conference on Learning Representations (ICLR 2025). arXiv preprint arXiv:2410.03943. Available at: https://arxiv.org/abs/2410.03943.
> > >
> > > Ceni, A., Cossu, A., Stölzle, M. W., Liu, J., Della Santina, C., Bacciu, D., & Gallicchio, C. (2024). Random Oscillators Network for Time Series Processing. In Proceedings of The 27th International Conference on Artificial Intelligence and Statistics (AISTATS 2024), Proceedings of Machine Learning Research, 238, 4807–4815. Available at: https://proceedings.mlr.press/v238/ceni24a.html.
> > >
> > > Lanthaler, S., Rusch, T. K., & Mishra, S. (2023). Neural Oscillators are Universal. arXiv preprint arXiv:2305.08753. Available at: https://arxiv.org/abs/2305.08753.

---

### Official Review · Reviewer_Nx2H · 2025-10-31

**Soundness:** 3
**Presentation:** 1
**Contribution:** 3
**Rating:** 6
**Confidence:** 2

**Summary:**

The original Transformer architecture does not work well with irregular time series due to they assume uniform time steps. While NODES handle irregularity by modeling continuous-time dynamics, they're computationally expensive due to requiring numerical ODE solvers. The paper proposes replacing NODEs with damped harmonic oscillators that have closed-form solutions, eliminating numerical solver overhead while maintaining expressiveness.

The keys and values are modeled as damped, driven oscillators and the queries are expanded in a sinusoidal basis. The attention is modeled as resonance such that high attention when frequencies align, low when misaligned.

The paper proved a Universal Approximation Theorem which shows that harmonic oscillators can approximate any continuous attention matrix achievable by ContiFormer

The paper provides experiments to demonstrate the effectiveness of the method.

**Strengths:**

This paper proposed a simple yet effective way to improve Transformer in handling irregular time series data. The closed form method can greatly reduce the computational complexity. The author also provides a proof to show that there proposed method still maintains expressiveness such that there is a UAP.

**Weaknesses:**

1. Since the ODE used has clear physical meaning, it would be interesting to explain any connection between the attention mechanism with the ODE.
2. There are certain part that is unclear
    2.1 Section 3 should be the most important part in this work. But too many contents are seemed to be putted into the appendix, making this part hard to follow. For example, line 142-157 gives some hard to understand terms: "Averaged attention: decomposition." "Steady-state contribution", "Transient contribution" and then refers to the equations in the derivation in the appendix, which forces the reader to read the entire appendix otherwise will not understand. If possible it would be good to explain these concepts clearly without the need to refer to the appendix.

    2.2 I do no see the significance of section 5 which discuss "E(3)-EQUIVARIANCE". This could be useful for certain applications, but I think due to the page limit, if is more important to expand section 3 which is the main contribution of the paper.

    3.3 Section 6 not looks like academic writing. There are many question-based headers and in conversational tone. The presentation of this part could be improved

**Questions:**

See Weakness.

---

> ### Author Response · Authors · 2025-11-23
> **Author Response to Reviewer Nx2H**
>
> We thank the reviewer for their detailed comments. We systematically answer the aforementioned weaknesses:
>
> ---
>
> >**Weakness 1:**
>
> > Since the ODE used has clear physical meaning, it would be interesting to explain any connection between the attention mechanism with the ODE.
>
> **Response to W1**
>
> We have provided a detailed explanation as well as a thorough mathematical framework linking the oscillator ODE to attention in Appendix A.5. In addition to this, we have also added intuitive experiments in Appendix E to help visualize the learning of the oscillator-based attention mechanism.
>
> ---
>
> >**Weakness 2:**
>
> > There are certain part that is unclear 2.1 Section 3 should be the most important part in this work. But too many contents are seemed to be putted into the appendix, making this part hard to follow. For example, line 142-157 gives some hard to understand terms: "Averaged attention: decomposition." "Steady-state contribution", "Transient contribution" and then refers to the equations in the derivation in the appendix, which forces the reader to read the entire appendix otherwise will not understand. If possible it would be good to explain these concepts clearly without the need to refer to the appendix.
>
> **Response to W2**
>
> We understand that moving back and forth between the main paper and the appendix can be tedious. However, due to the page restrictions set forth by ICLR, it was prudent for us to shift certain parts of the paper to the appendix.
>
> ---
>
> >**Weakness 3:**
>
> > 2.2 I do no see the significance of section 5 which discuss "E(3)-EQUIVARIANCE". This could be useful for certain applications, but I think due to the page limit, if is more important to expand section 3 which is the main contribution of the paper.
>
> **Response to W3**
>
> Many irregular time series problems involve physical systems with spatial symmetries. For example, weather-based data and geophysical data are both irregular and exhibit physical symmetries. This property renders E(3)-equivariance very useful while studying physical systems, which is why we added this section originally. But, we have taken note of the reviews and moved it to Appendix C. We noticed other reviewers also raised this concern, and have added a detailed response in the general comment.
>
> ---
>
> >**Weakness 4:**
>
> > 3.3 Section 6 not looks like academic writing. There are many question-based headers and in conversational tone. The presentation of this part could be improved
>
> **Response to W4**
>
> Thank you for your feedback. We have formalized the section.

---

> > ### Comment · Reviewer_Nx2H · 2025-11-26
> >
> > Thanks for the reply, which resolves most of my concerns. I will maintain my score.

---

### Official Review · Reviewer_UpK7 · 2025-10-31

**Soundness:** 2
**Presentation:** 3
**Contribution:** 2
**Rating:** 2
**Confidence:** 3

**Summary:**

This paper proposes a novel continuous-time Transformer architecture, OsciFormer, designed to efficiently model irregular time series without the computational bottlenecks of Neural ODE-based methods such as ContiFormer. The key idea is to replace the Neural ODE layers—which require costly numerical solvers—with damped, driven harmonic oscillators that admit closed-form analytical solutions.

in this formulation, keys and values evolve as damped harmonic oscillators, while queries are expanded in a sinusoidal basis, enabling an elegant resonance-based attention mechanism where alignment between query and key frequencies drives attention weights

**Strengths:**

1. The oscillator analogy for attention is creative and intellectually stimulating, merging physical dynamics with deep learning.
2. The closed-form formulation drastically reduces computational cost, memory usage, and sequential depth.
3. Strong results across diverse irregular time-series domains, maintaining or exceeding ContiFormer’s accuracy while being much faster.

**Weaknesses:**

1. The universality proof concerns approximating keys and then attention weights; it doesn’t address training dynamics or resilience under irregular sampling noise, or non-smooth signals common in irregular TS.
2. No systematic ablation on J (modes), γ (damping), frequency grids, or initial-condition maps.
3. E(3)-eq section is ungrounded. The equivariance construction is mathematically plausible but has no experiments; it reads as a speculative add-on.
4. The abstract touts “state-of-the-art performance ... orders faster,” but several tables show parity or losses depending on dataset; conclusions should be tempered.
5. The table 4 is too large and exceeds the page margins.

**Questions:**

1. Do you have fail cases (when does the model diverge or lose accuracy?
2. No strong ablations. We never see performance vs: number of oscillator modes, γ (damping), frequency grids, or initial-condition maps.
3. Any training-time results toward stability/identifiability (e.g., constraints on eigenvalues or damping priors) beyond the key-approximation theorem?
4. The harmonic approximation theorem is asymptotic: given enough oscillators, we can approximate any continuous trajectory up to ϵ, which then bounds the softmax deviation in attention, but there’s no guarantee that SGD will find those oscillator parameters from data.
5. Can you actually visualize “attention weight vs frequency alignment” on a real example?

---

> ### Author Response · Authors · 2025-11-23
> **Author Response to Reviewer UpK7**
>
> ---
>
> We thank the reviewer for their detailed feedback. We respond to the identified weaknesses and questions below.
>
> > **Weakness 1:**
> > The universality proof concerns approximating keys and then attention weights; it doesn’t address training dynamics or resilience under irregular sampling noise, or non-smooth signals common in irregular TS.
>
> **Response to W1.**
> We thank the reviewer for raising this concern. We would like to point out that we have indeed conducted experiments on non-linear and non-oscillatory datasets. We noticed that other reviewers also raised this concern, and have answered it in detail in the general comment. Furthermore, to understand the training dynamics and stability of the model, we have conducted ablation experiments, details of which are given in the general comment.
>
> ---
>
> > **Weakness 2:**
> > No systematic ablation on J (modes), γ (damping), frequency grids, or initial-condition maps.
>
> > **Question 2**
> > No strong ablations. We never see performance vs: number of oscillator modes, γ (damping), frequency grids, or initial-condition maps.
>
> **Combined response to W2 & Q2.**
> We have taken note of the reviewer’s concern, and have conducted detailed ablation studies on the variation of performance with different parameters. Further details are given in the general comment.
>
> ---
> > **Weakness 3:**
> > E(3)-eq section is ungrounded. The equivariance construction is mathematically plausible but has no experiments; it reads as a speculative add-on.
>
> **Response to W3.**
> We understand the reviewer’s concern. We have moved this section to Appendix C. However, we do believe that this is a useful property and have given details in the general response.
>
> ---
>
> > **Weakness 4:**
> > The abstract touts “state-of-the-art performance ... orders faster,” but several tables show parity or losses depending on dataset; conclusions should be tempered.
>
> **Response to W4.**
> We would like to point out that our model is a closed-form implementation of the ContiFormer, and therefore our experiments reflect parity with the original paper. We do not think that our claims are hyperbolic since, we do in fact see orders of magnitude faster performance. This is a testament to the efficiency and widespread applications of our model.
>
> ---
>
> > **Weakness 5:**
> > The table 4 is too large and exceeds the page margins.
>
> **Response to W5.**
> We have taken note of this and fixed the issue.
>
> ---
> Due to the character limit, we answer the questions in the next comment.

---

> ### Author Response · Authors · 2025-11-23
> **Author Response to Reviewer UpK7**
>
> ---
> > **Question 1.**
> > Do you have fail cases (when does the model diverge or lose accuracy?
>
> **Response to Q1.**
> Yes, our model fails for chaotic systems. This is consistent with recent work on Transformers for chaotic systems, which shows that standard Transformers perform very poorly on benchmarks like Lorenz-96 and fail to capture chaotic dynamics (Hu et al., 2024; He et al., 2025). To further illustrate this, we generated 6k samples from the logistic map $x_{t+1} = r\cdot x_t(1-x_t)$ with $r=3.57$ and $x_0=0.6$, producing a chaotic time series with a positive Lyapunov exponent. We split the data into training (first 4k points) and test (remaining 2k points). We then tackle a simple regression task where, given a window of past observations $(x_{t-w}, ..., x_{t-1}, x_t)$, we predict the next value  $x_{t+1}$. We train our model with different oscillator modes to minimize prediction error. At test time, we roll out multi-step predictions by feeding each prediction back as input for the next step, evaluating how well the model performs. Detailed graphs can be found in Appendix F.
>
> ---
> > **Question 3.**
> > Any training-time results toward stability/identifiability (e.g., constraints on eigenvalues or damping priors) beyond the key-approximation theorem?
>
> **Response to Q3.**
> We thank the reviewers for raising the point.  In Appendix A, we explicitly characterize the stability of each $2\times2$ oscillator block
>
> $$
> A(\omega,\gamma) =
> \begin{pmatrix}
> 0 & 1\\
> -\omega^2 & -2\gamma
> \end{pmatrix}
> $$
>
> whose eigenvalues $\lambda_{\pm} = -\gamma \pm \sqrt{\gamma^2-\omega^2}$  satisfy   $\mathrm{Re}(\lambda_{\pm}) \le -\gamma \le 0$ under our parameterization  $\omega>0$, $\gamma\ge 0$  (Def. 3, App. A.1). Each oscillator block is thus Hurwitz Matrix.  The block-diagonal bank $A = \mathrm{diag}(A_0,\dots,A_M)$ is also Hurwitz which  guarantees exponential stability of the flow $z'(t)=Az(t)$: there exist $K,\alpha>0$ such that $|e^{A(t-t_0)}|\le Ke^{-\alpha(t-t_0)}$ for all $t\ge t_0$.  We have not proven full statistical identifiability, and is an interesting future direction.
>
> ---
>
> > **Question 4.**
> > The harmonic approximation theorem is asymptotic: given enough oscillators, we can approximate any continuous trajectory up to ϵ, which then bounds the softmax deviation in attention, but there’s no guarantee that SGD will find those oscillator parameters from data.
>
> **Response to Q4.**
> Thank you for your feedback. We show that for any continuous key trajectories $k_i$ on a compact interval and any $\varepsilon > 0$, there exists a finite oscillator bank with a finite number of modes $M$ and suitable initial states such that $\sup_t \lVert k_i(t) - \bar{k}(t) \rVert < \varepsilon$. This formalizes the claim that any discrete attention matrix realizable by ContiFormer’s continuous keys can be approximated arbitrarily well by a shared harmonic oscillator bank.
>
> Classical approximation theorems (Cybenko, 1989; Hornik, 1991) establish expressivity but do not deal with optimization. Similarly, Yun et al. (2020) show that multi-head self-attention networks with positional encodings are universal approximators of continuous sequence-to-sequence functions, but their results are also purely about expressivity rather than the behavior of SGD. The original ContiFormer analysis likewise focuses on expressive power, not on proving that SGD will recover the corresponding continuous-time dynamics. Architectures such as CFC (Hasani et al., 2022) also do not provide general SGD guarantees. Even for much simpler models, convergence guarantees for SGD require strong assumptions such as (strongly) convex objectives or special over-parameterized regimes (Bottou et al., 2018; Wu et al., 2018). In our experiments, we construct oscillator banks by fixing a shared frequency grid and treating $\omega_0$ and damping as learnable parameters. Empirically, we find that standard optimizers such as AdamW train OsciFormer sufficiently well across all benchmarks.
>
> ---
> > **Question 5.**
> > Can you actually visualize “attention weight vs frequency alignment” in a real example?
>
> **Response to Q5.**
> We have added experiments for this purpose in Appendix E, details of which are given in the general comment.
>
> ---
>
> Due to the character limit, we mention the references in the next comment

---

> ### Author Response · Authors · 2025-11-23
> **Author Response to Reviewer UpK7**
>
> ---
> **References**
>
> - ⁠Hu, J., Hu, Y., Chen, W., Jin, M., Pan, S., Wen, Q., & Liang, Y. (2024). Attractor Memory for Long-Term Time Series Forecasting: A Chaos Perspective. arXiv preprint arXiv:2402.11463. Available at: [https://arxiv.org/abs/2402.11463](https://arxiv.org/abs/2402.11463)
>
> - ⁠He, Y., Yang, Y., Cheng, X., Wang, H., Xue, X., Chen, B., & Hu, Y. (2025). Chaos Meets Attention: Transformers for Large-Scale Dynamical Prediction. arXiv preprint arXiv:2504.20858. Available at: [https://arxiv.org/abs/2504.20858](https://arxiv.org/abs/2504.20858)
>
> - George Cybenko. *Approximation by Superpositions of a Sigmoidal Function*. Mathematics of Control, Signals and Systems, 2(4):303–314, 1989.
>
> - Kurt Hornik. *Approximation Capabilities of Multilayer Feedforward Networks*. Neural Networks, 4(2):251–257, 1991.
>
> - Chulhee Yun, Srinadh Bhojanapalli, Ankit Singh Rawat, Sashank J. Reddi, and Sanjiv Kumar. *Are Transformers Universal Approximators of Sequence-to-Sequence Functions?* In International Conference on Learning Representations (ICLR), 2020.
>
> - Ramin Hasani, Mathias Lechner, Alexander Amini, Lucas Liebenwein, Aaron Ray, Max Tschaikowski, Gerald Teschl, and Daniela Rus. *Closed-Form Continuous-Time Neural Networks*. Nature Machine Intelligence, 4(11):992–1003, 2022.
>
> - Léon Bottou, Frank E. Curtis, and Jorge Nocedal. *Optimization Methods for Large-Scale Machine Learning*. SIAM Review, 60(2):223–311, 2018. (arXiv:1606.04838)
> [8] (Example SGD convergence reference) For instance, see results on SGD convergence in over-parameterized networks such as: Daniel Zou et al. *An Improved Analysis of Training Over-parameterized Deep Neural Networks.* NeurIPS 2019.
>
> ---

---

### Official Review · Reviewer_mLHL · 2025-11-01

**Soundness:** 4
**Presentation:** 2
**Contribution:** 3
**Rating:** 6
**Confidence:** 4

**Summary:**

This paper introduces OsciFormer, a continuous-time transformer that replaces Neural ODE–based dynamics in ContiFormer with analytically solvable damped harmonic oscillators. By modeling attention as a resonance phenomenon between query and key oscillations, OsciFormer provides a closed-form solution eliminating the computational overhead of ODE solvers while retaining theoretical expressivity. The authors prove a universal approximation theorem for continuous attention, establish E(3)-equivariance for geometric data, and show strong empirical results across irregular time-series benchmarks, achieving comparable or superior accuracy with up to 20× faster training compared to ODE-based methods.

**Strengths:**

- The use of damped harmonic oscillators as a way to explain attention is a cool and intuitive idea—it helps make the concept more accessible.
- By switching to a closed-form solution, the model cuts down on computation and memory usage big time.
- The theoretical proof showing that this method can still approximate the continuous attention mechanism is solid and convincing.
- The experiments are thorough and show that the model works well in practice, not just theory.

**Weaknesses:**

- Some of the math-heavy sections, like the attention integrals, feel a bit rushed and could use clearer explanations.
- The paper compares accuracy and efficiency but doesn't dive into how the oscillator settings (like the damping or number of modes) actually impact performance.
- The connection between the oscillators and the learned attention patterns could be explained better.

**Questions:**

- Can you explain more clearly how the resonance behavior actually influences the learned attention in a more visual or intuitive way?
- How much does changing the number of oscillator modes or the damping coefficients affect performance? It would be helpful to see an ablation study.
- Do you think this model could work well for other tasks like NLP or computer vision? It would be interesting to hear your thoughts.

---

> ### Author Response · Authors · 2025-11-23
> **Author Response to Reviewer mLHL**
>
> We sincerely thank the reviewer for their constructive feedback. We respond to the identified weaknesses and questions below.
>
> ---
>
> >**Weakness 1**
> > *"Some of the math-heavy sections, like the attention integrals, feel a bit rushed and could use clearer explanations."*
>
> **Response to W1**
>
> We acknowledge that some parts of the paper are mathematically dense. While we would have liked to provide more explanations in the main paper itself, page limits forced us to push the detailed derivations to the Appendix.
>
> ---
>
> >**Weakness 2**
> > *"The paper compares accuracy and efficiency but doesn't dive into how the oscillator settings (like the damping or number of modes) actually impact performance."*
>
> >**Question 2**
> > *"How much does changing the number of oscillator modes or the damping coefficients affect performance? It would be helpful to see an ablation study."*
>
> **Combined Response to W2 & Q2**
>
> We acknowledge the importance of analyzing the effect of the different oscillator parameters, including the number of modes, frequency ranges, and damping coefficients, on overall model performance. We have conducted detailed ablations and present the results in Tables 6 to 11 of Appendix E.1.
>
> ---
>
> >**Weakness 3**
> > *"The connection between the oscillators and the learned attention patterns could be explained better."*
>
> > **Question 1**
> > *"Can you explain more clearly how the resonance behavior actually influences the learned attention in a more visual or intuitive way?"*
>
> **Combined Response to W3 & Q1**
>
> To simply visualize how our oscillator-based attention mechanism learns, we conducted a simple classification experiment on synthetic time series data. We generated irregularly sampled signals made from noisy sinusoids, where each signal belongs to one of 8 frequency classes (ranging from 2.0 to 9.0 rad/s). Herein, our model has 8 learnable keys with learnable parameters $\gamma$ (damping) and $\omega_0$ (natural frequency). The query is constructed using Fourier projections of the input signal. We then calculate the attention analytically using our closed-form solution. We train all parameters using cross-entropy loss.
>
> There are 3 visualisations which depict how resonance behaviour influences the learned attention:
>
> - **Frequency alignment.** After training, each key's natural frequency $\omega_{0,i}$ aligns with one of the true class frequencies (2.0, 3.2, ..., 9.0 rad/s). This shows the model learns to create resonant filters matched to the input signal frequencies.
>
> - **Phase map for a learned key.** This graph shows the attention score for different frequencies (vertical) and phases (horizontal). The bright ridge shows where the query phase matches the oscillator's response. Maximum attention occurs along this optimal curve.
>
> - **Attention vs gain.** This graph compares two curves, the oscillator's resonance response |H(ω)| and the maximum attention score. Both curves have the same shape and peak.
>
> ---
>
> >**Question 3**
> > *"Do you think this model could work well for other tasks like NLP or computer vision? It would be interesting to hear your thoughts."*
>
> **Response to Q3**
>
> Our model works for NLP tasks as well. We evaluated sentiment classification accuracy on 25,000 IMDB movie reviews, predicting whether each review expresses positive or negative sentiment (See Table 5 in Appendix D). We get good results on the task because the hidden representations in neural networks for NLP tasks are continuous vectors. In our case, the Neural ODE models how those continuous hidden states evolve as a function of time. Neural ODE-based models have been previously applied to text classification and natural language inference (for example, Zhang et al. (2021) propose a self-attention ODE solver that couples a self-attention block with a neural ODE; they explicitly evaluate text classification, natural language inference (NLI), and text matching). For vision tasks, we think this model can also be used. Previously, neural ODE-based models have been applied for classification (Chen et al., 2018; Cui et al., 2023). However, a broader analysis is required, and we would indeed want to test our closed-form solutions on other tasks, since they provide a well-grounded physical intuition of attention.
>
> ---
>
> **References**
>
> - Zhang, J., Zhang, P., Kong, B., Wei, J., & Jiang, X. (2021). Continuous Self-Attention Models with Neural ODE Networks. *Proceedings of the AAAI Conference on Artificial Intelligence*, 35(16), 14393–14401. https://doi.org/10.1609/aaai.v35i16.17692
>
> - Chen, R. T. Q., Rubanova, Y., Bettencourt, J., & Duvenaud, D. (2019). Neural Ordinary Differential Equations. arXiv:1806.07366. https://arxiv.org/abs/1806.07366
>
> - Cui, W., Zhang, H., Chu, H., Hu, P., & Li, Y. (2023). On robustness of neural ODEs image classifiers. *Information Sciences*, 632, 576–593. https://doi.org/10.1016/j.ins.2023.03.049

---

### Author Response · Authors · 2025-11-23
**Author Response to All Reviewers**

We thank all the reviewers for their valuable feedback. We noticed some common questions throughout all the reviews, and answer them below:

### 1. General Ablations


We have conducted a detailed set of ablations over (i) the number of oscillator modes (J) (from 1 to 16), showing that performance consistently saturates around $J \in [6,8]$ (ii) different damping ranges, where moderate damping $[0.05, 0.40]$ yields the best accuracy stability tradeoff and (iii) several frequency grid parameterizations, where a log-uniform grid leads to the most stable results across datasets. We further analysed how the learned oscillators distribute across underdamped, critically damped, and overdamped regimes on all event‑prediction benchmarks (see Tables 6 to 11 in Appendix E.1).


### 2. Harmonic Approximation Theorem and Non-Oscillatory Datasets


We would like to clarify that the proof of the Harmonic Approximation Theorem has two points that are important for this discussion:
1. We approximate the key trajectories, not the raw input signal.
We do not claim that the observed time series $X(t)$ is a sum of sinusoids. Instead, the keys and values live in a latent space and evolve according to a damped, driven harmonic oscillator with a closed-form solution.
2. No oscillation assumption on the data.
The only assumption is continuity of $q(t)$ and $k_i(t)$ in time. There is no requirement that the observed time series be sinusoidal or “nice”: piecewise plateaus, sharp transitions, and irregular dynamics are all allowed, as long as the latent keys produced by the network are continuous.
Now connecting this to ContiFormer’s original approximation theorem (Chen et al., 2023). Our Harmonic Approximation Theorem then says: given any continuous keys and queries constructed by ContiFormer, there exists a harmonic oscillator bank that re-parameterises those same keys to arbitrary precision. Therefore, any discrete attention matrix that ContiFormer can realise via its Neural ODE trajectories can also be realised (up to an arbitrarily small error) by OsciFormer’s oscillator parameterisation of the keys.
In our experiments, many of the datasets are non-linear and non-oscillatory in the raw space. For example, the event prediction suite (Table 1) is not oscillatory and consists of datasets with plateaus and sharp changes in intensity. These include Neonate (clinical seizure events), Traffic (PeMS), MIMIC ICU visits, StackOverflow badges, and BookOrder transactions. On BookOrder, OsciFormer has LL (-0.288) vs (-0.270) for ContiFormer, accuracy (0.626) vs (0.628), and identical RMSE (3.621 vs 3.614). On Traffic and MIMIC, OsciFormer is similar to ContiFormer to within the reported error bars in both LL and accuracy.
Further, in Table 4, the inputs are 0/1 sequences with abrupt flips and hence are non-oscillatory. On equidistant encoding (regular grid), OsciFormer attains 100.00% accuracy, equal to the best RNN baselines. Under event-based encoding (Table 4), OsciFormer achieves (99.83 \pm 0.32%) accuracy, again matching ContiFormer’s 99.9% but with substantially lower per-epoch time (0.56 min vs 3.83 min).
Similarly, in Table 2, the UCR/UEA long-context benchmarks include datasets like SCP1/2, EW, ETC, HB, MI, which are classic shape-based time-series problems. Many of these have piecewise plateaus and impulses. Even in Fig. 2(c,d), the datasets follow a spiral in the plane rather than a simple sinusoid. OsciFormer reconstructs these trajectories well and trains faster than the original ContiFormer.


### 3. E(3) Equivariance


Many irregular time series arise as pointwise observations of underlying spatiotemporal physical fields. In climate and geoscience, such irregularly sampled time series are well documented (Shukla & Marlin, 2020). Recent work explicitly treats weather forecasts as continuous fields evaluated at irregular time points via neural operators (Ge et al., 2022). These underlying physical systems also demonstrate strong spatial symmetries and are increasingly modelled with equivariant neural operators on Euclidean or spherical domains (Bonev et al, 2023).
Further, E(3)-equivariant networks are now used for time-dependent 3D physical systems such as molecular dynamics (Batzner et al., 2022).  Our goal in Section 5 was therefore not to claim an additional empirical contribution, but to show that our closed-form solutions using damped, driven harmonic oscillators can be made E(3)-equivariant and can be used for such spatiotemporal physical systems. However, to keep the main paper focused, we have moved the detailed E(3) construction to Appendix C.

Due to the character limit, we answer the remaining questions and provide references in the next comment.

---

> ### Author Response · Authors · 2025-11-23
> **Author Response to All Reviewers**
>
> ### 4. Attention Visualisation
>
>
> To visualize the resonance view of attention, we have conducted simple, irregular time-series-based classification and regression experiments. We have shown how attention concentrates on oscillator keys whose learned natural frequencies align with the underlying signal. Further, we show a phase frequency attention diagram $\alpha(\omega, \phi)$, where the bright ridge in the $(\omega,\phi)$ shows resonance. These experiments provide intuitive insight into how attention mechanisms can be understood through the lens of resonance.
>
>
> ### 5. Cosmetic Changes
>
>
> We have implemented all cosmetic edits requested by the reviewers.
>
> ### 6. Resonance and damping
>
> Resonance is sharpest in the under-damped regime because the system retains oscillatory energy for many cycles, allowing the driving frequency to build up a large steady-state amplitude. As damping increases, the oscillations die out more quickly, so the “peak” in the frequency response becomes broader and lower. In the critically damped case, there is no true oscillation, but the system still responds most strongly near its natural frequency, giving a mild resonance-like enhancement. In the over-damped regime, the motion is purely exponential and non-oscillatory, yet the frequency response still shows preferential amplification around a characteristic frequency. Thus, although sharp resonance disappears with large damping, the system continues to behave as a frequency-selective filter across all regimes.
>
> ---
>
> ### 7.Oscillators vs oscillatory behaviour
>
> We would also like to clarify that damped driven harmonic oscillators do not just behave in an oscillatory way, and our main motivation for using them is that they give a rich system where there are both underdamped and overdamped regimes. We fundamentally believe that oscillators are good, and we are consistent with the overall growing use of oscillators for neural networks. ( Rusch, 2024 ) (Ceni et al., 2024)
>
>
> ## References
>
> * Chen, Y., Ren, K., Wang, Y., Fang, Y., Sun, W., & Li, D. (2023).
>   **ContiFormer: Continuous-time Transformer for Irregular Time Series Modeling.**
>   In *Advances in Neural Information Processing Systems 37 (NeurIPS 2023).*
>
> * Shukla, S. N., & Marlin, B. M. (2020).
>   **A Survey on Principles, Models and Methods for Learning from Irregularly Sampled Time Series: From Discretization to Attention and Invariance.**
>   In *NeurIPS 2020 Workshop on ML Retrospectives, Surveys & Meta-Analyses (ML-RSA).* arXiv:2012.00168.
>
> * Ge, T., Pathak, J., Subramaniam, A., & Kashinath, K. (2022).
>   **DL-Corrector-Remapper: A Grid-free Bias-correction Deep Learning Methodology for Data-driven High-resolution Global Weather Forecasting.**
>   *arXiv preprint* arXiv:2210.12293. (NeurIPS 2022 Workshop on Tackling Climate Change with Machine Learning.)
>
> * Xu, M., Han, J., Lou, A., Kossaifi, J., Ramanathan, A., Azizzadenesheli, K., Leskovec, J., Ermon, S., & Anandkumar, A. (2024).
>   **Equivariant Graph Neural Operator for Modeling 3D Dynamics.**
>   In *Proceedings of the 41st International Conference on Machine Learning (ICML 2024).* arXiv:2401.11037.
>
> * Batzner, S., Musaelian, A., Sun, L., Geiger, M., Mailoa, J. P., Kornbluth, M., Molinari, N., Smidt, T. E., & Kozinsky, B. (2022).
>   **E(3)-Equivariant Graph Neural Networks for Data-efficient and Accurate Interatomic Potentials.**
>   *Nature Communications*, 13, 2453.
>
> * Bonev, B., Kurth, T., Hundt, C., Pathak, J., Baust, M., Kashinath, K., & Anandkumar, A. (2023).
>   **Spherical Fourier Neural Operators: Learning Stable Dynamics on the Sphere.**
>   *arXiv preprint* arXiv:2306.03838.
>
> * Rusch, T. K., & Rus, D. (2025).
>   **Oscillatory State-Space Models.**
>   In *Proceedings of the International Conference on Learning Representations (ICLR 2025).* arXiv:2410.03943.
>
> * Ceni, A., Cossu, A., Stölzle, M. W., Liu, J., Della Santina, C., Bacciu, D., & Gallicchio, C. (2024).
>   **Random Oscillators Network for Time Series Processing.**
>   In *Proceedings of the 27th International Conference on Artificial Intelligence and Statistics (AISTATS 2024), Proceedings of Machine Learning Research, 238, 4807–4815.*

---

### Meta-Review · Area_Chair_maZ6 · 2026-01-05

**Summary:**

The overall sentiment of the reviewers is that the manuscript is not ready for publication yet, e.g., every reviewer raised concerns about section 5 of the initial version which discusses "E(3)-equivariance. The reviewers agreed that in particular this part feels rushed and proper empirical evaluations are needed. Moreover, the paper fails to compare to competing oscillatory neural networks, e.g., in Table 2. This weakens the empirical justifications. Overall, while the idea of the paper is interesting, it could benefit from a proper revision of the empirical results.

**Reviewer Concerns:**

I believe that the authors successfully addressed the concerns about the universal approximation result. Moreover, the authors provided more empirical insights into the proposed method via sensitivity studies and ablations. However, important baselines are still missing, as well as proper empirical investigation of the equivariance of the model.

**Reviewer Scores:**

I believe Reviewers mLHL and Nx2H would have kept their score of 6, while Reviewers UpK7 and EAsp might have potentially increased their score to 4. However, it is worth noting that Reviewer EAsp still verbalized 'major concerns' in their responds to the authors' rebuttal.

---

### Decision · Program_Chairs · 2026-01-26

Reject